# The maintenance of oocytes in the mammalian ovary involves extreme protein longevity

Katarina Harasimov [1,9,13], Rebecca L. Gorry[1,13], Luisa M. Welp[2,3,13], Sarah Mae Penir [1,13], Yehor Horokhovskyi [4,13], Shiya Cheng [1], Katsuyoshi Takaoka [1,10], Alexandra Stützer[2], Ann-Sophie Frombach[1], Ana Lisa Taylor Tavares[5,11], Monika Raabe[2], Sara Haag [1,12], Debojit Saha[1], Katharina Grewe[6,7], Vera Schipper[1], Silvio O. Rizzoli [6,7], Henning Urlaub [2,3,8 ✉], Juliane Liepe [4 ✉] & Melina Schuh [1,8 ✉]

Women are born with all of their oocytes. The oocyte proteome must be maintained with minimal damage throughout the woman's reproductive life, and hence for decades. Here we report that oocyte and ovarian proteostasis involves extreme protein longevity. Mouse ovaries had more extremely long-lived proteins than other tissues, including brain. These long-lived proteins had diverse functions, including in mitochondria, the cytoskeleton, chromatin and proteostasis. The stable proteins resided not only in oocytes but also in long-lived ovarian somatic cells. Our data suggest that mammals increase protein longevity and enhance proteostasis by chaperones and cellular antioxidants to maintain the female germline for long periods. Indeed, protein aggregation in oocytes did not increase with age and proteasome activity did not decay. However, increasing protein longevity cannot fully block female germline senescence. Large-scale proteome profiling of ~8,890 proteins revealed a decline in many long-lived proteins of the proteostasis network in the aging ovary, accompanied by massive proteome remodeling, which eventually leads to female fertility decline.

The female ovary is critical for reproduction. It stores the primordial follicles, which contain the oocytes and their associated somatic cells[1,2]. The primordial follicles are generated in the female fetus and do not appear to be replenished after birth. Females are therefore thought to be born with a finite pool of oocytes, called the ovarian reserve[3,4].

The proteome of oocytes must be maintained in a healthy state throughout a woman's reproductive life to ensure the success of the next generation. Whether germ cells have adapted their proteostasis to maintain and propagate a healthy proteome is unknown[5]. The identification of such adaptations could reveal the principles involved in resetting the aging clock and inform new therapeutic strategies to delay age-related diseases.

In this study, we analysed proteostasis in mammalian oocytes and ovaries by combining quantitative mass spectrometry (MS), pulse–chase labelling, single-cell RNA-seq and nanoscale secondary ion MS (NanoSIMS). We found that the maintenance of oocytes in the mammalian ovary involves exceptional protein longevity. Many of the extremely long-lived proteins decline as the ovary ages. We propose that extreme protein longevity is required to propagate a healthy germline across generations, yet protein depletion may promote the rapid age-related decline in female fertility.

**a** Identification of long-lived proteins in oocytes

PULSE: $^{13}C_6$-Lys until birth | CHASE: $^{12}C_6$-Lys from birth

>98.8% $^{13}C_6$-Lys labelled mother · Newborn pups · 8 weeks · 4,948 oocytes · MS analysis

**b** Pathways enriched in long-lived proteins in oocytes

**c**

Actin cytoskeleton

| Gene name | Protein (%H) |
|---|---|
| Carmil2 | 70.7 |
| Filip1 | 95.6 |
| Fmn1 | 87.5 |
| Xirp2 | 86.0 |

Chromosome organization

| Gene name | Protein (%H) |
|---|---|
| Cbfa2t2 | 90.4 |
| H3f3a | 72.2 |
| Hira | 94.5 |
| Kat2b | 20.9 |
| Rbl1 | 50.0 |
| Sap30l | 26.5 |
| Smc3 | 17.7 |
| Tshz3 | 100.0 |
| Wrap53 | 64.8 |

DNA repair

| Gene name | Protein (%H) |
|---|---|
| Ascc3 | 32.8 |
| Rif1 | 16.3 |

Microtubule cytoskeleton

| Gene name | Protein (%H) |
|---|---|
| Cep250 | 18.6 |
| Cep290 | 50.0 |
| Kif1b | 41.2 |
| Kif3b | 33.0 |
| Mtcl1 | 75.3 |
| Sass6 | 78.7 |

Mitochondrial proteins

| Gene name | Protein (%H) |
|---|---|
| Acsl3 | 40.5 |
| Clybl | 38.7 |
| Pptc7 | 42.9 |
| Ttc19 | 100.0 |
| Twnk | 100.0 |

Others

| Gene name | Protein (%H) |
|---|---|
| Cdk16 | 19.7 |
| Cdk18 | 37.0 |
| Cyp1b1 | 34.6 |
| Dnajc21 | 17.4 |
| Ecd | 17.0 |
| Foxo1 | 21.9 |
| Fsip2 | 48.3 |
| Itga11 | 35.9 |
| Myof | 50.0 |
| Rabl2 | 16.7 |
| Rif1 | 16.3 |
| Tcf3 | 100 |
| Zfyve1 | 48.6 |

Ribosome

| Gene name | Protein (%H) |
|---|---|
| Bop1 | 95.1 |
| Rsl24d1 | 24.8 |

**Fig. 1 | Oocytes contain a large number of very long-lived proteins. a**, Fully labelled $^{13}C_6$-Lys females were mated and fed $^{13}C_6$-Lys chow until they gave birth (pulse). Upon birth, the pups were transferred to unlabelled foster mothers (chase) and continued to receive $^{12}C_6$-Lys chow after weaning (Fig. 1a, Extended Data Fig. 1a–e). In total, 4,948 oocytes were collected from 92 eight-week-old pubertal female progeny and processed for bottom-up MS. **b**, Selected pathways enriched with long-lived proteins in oocytes. All $^{13}C_6$-Lys-labelled proteins detected after 8 weeks were subjected to over-representation analysis. Shown are the percentage of genes of the gene set detected as enriched with their respective adjusted $P$ values for selected pathways. The $P$ values are based on a hypergeometric test and have been adjusted using Benjamini–Hochberg multiple hypothesis testing. For the complete list of enriched pathways with their corresponding exact $P$ values, see Supplementary Table 2. **c**, Selected biological processes and protein complexes with long-lived proteins. Shown are selected genes with corresponding protein %H values (inferred fraction of $^{13}C_6$-Lys, mean value over biological replicates). For a complete list of proteins and their corresponding %H values, see Supplementary Table 1.

## Results

### Oocytes contain many extremely long-lived proteins

To study protein stability in oocytes, we performed a pulse–chase experiment in which female mice were fully labelled with $^{13}C_6$-lysine ($^{13}C_6$-Lys)[6] during their development in utero, transferred to unlabelled ($^{12}C_6$-Lys) foster mothers after birth and fed unlabelled chow after weaning (Fig. 1a, Extended Data Fig. 1a–e). We collected 4,948 fully grown oocytes from 92 eight-week old mice and processed them for bottom-up MS (Fig. 1a, Extended Data Fig. 2a).

Despite the 8-week chase period and dilution by oocyte growth, we detected many $^{13}C_6$-Lys-positive proteins (768 out of 7,263 proteins),

which must have been present since birth (Extended Data Fig. 2c,d and Supplementary Table 1). Notably, 39 of the proteins were still 90–100% positive for $^{13}C_6$-Lys, 27 proteins between 50% and <90%, 219 proteins between 1 and <50%, and 483 proteins <1% (Extended Data Fig. 2e). These values represent raw $^{13}C_6$-Lys over total Lys measurements without correction for oocyte growth. The 768 long-lived proteins were significantly enriched in diverse components (for example, mitochondria, ribosome, spliceosome, proteasome, chromatin, kinetochore and cytoskeleton) and functions (for example, metabolism, chaperones, DNA repair and antioxidants; Fig. 1b,c and Supplementary Table 2).

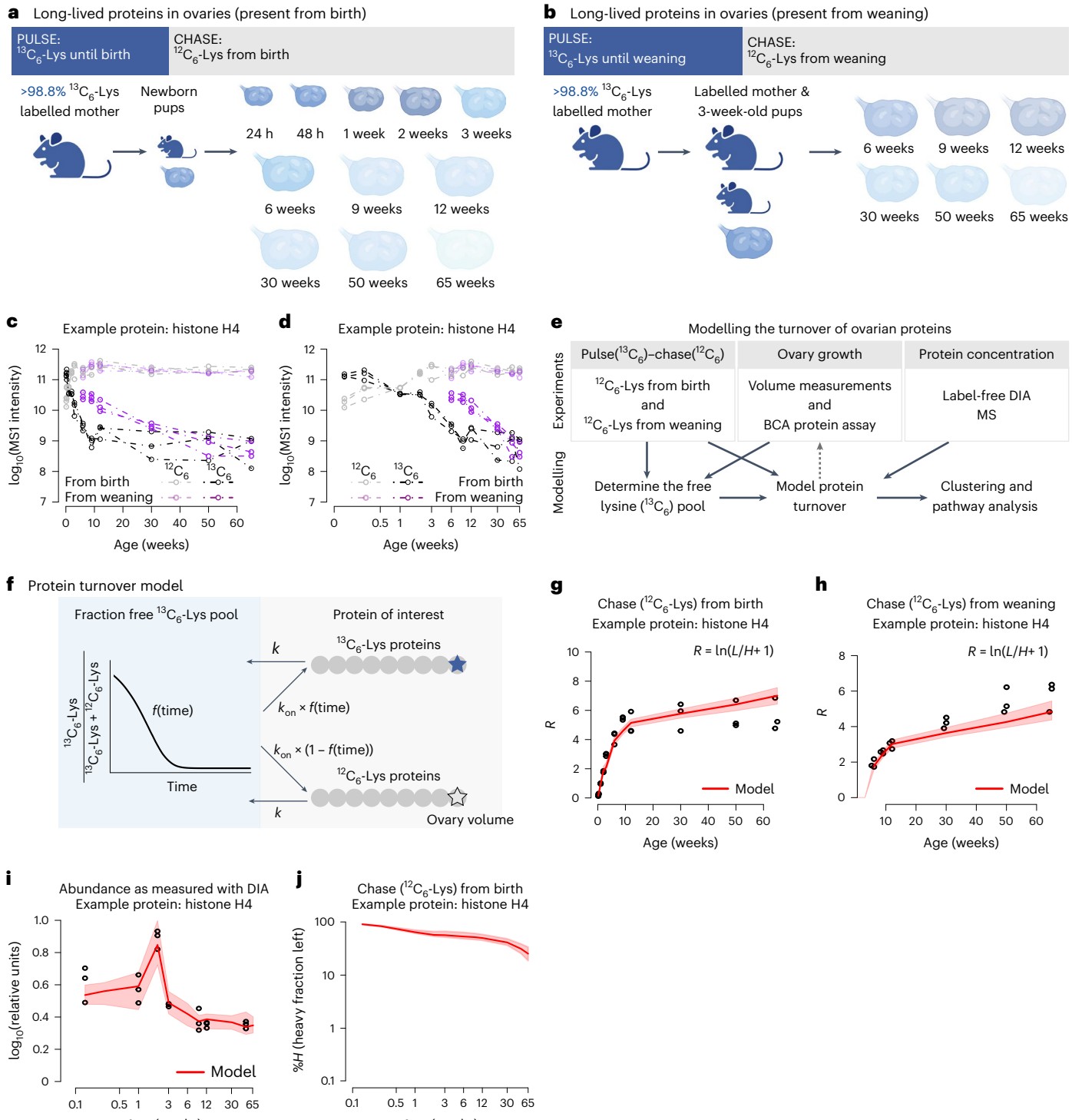

**Fig. 2 | Modelling protein turnover throughout ovarian development.**
**a**, Fully $^{13}C_6$-Lys labelled pregnant mice were fed with $^{13}C_6$-Lys chow until they gave birth (pulse). The pups were subsequently raised on $^{12}C_6$-Lys (chase). Ovaries were collected from the female progeny at eleven time points (three animals per time point) and processed for DDA MS. **b**, Progeny from fully $^{13}C_6$-Lys labelled pregnant mice were fed with $^{13}C_6$-Lys until weaning. After the weaning period (3 weeks after birth), progeny were fed $^{12}C_6$-Lys chow. Ovaries were collected from female progeny at six time points (three animals per time point) and processed for DDA MS. **c,d**, Example data used for modelling protein turnover. Shown are MS1 intensities over time (linear **c**; $\log_{10}$, **d**) scale for $^{13}C_6$-Lys (black and purple) and $^{12}C_6$-Lys (grey and pink) labelled histone H4, for two pulse lengths (pulse until birth and until weaning). Circles indicate experimental data points for three biological replicates, dashed lines are only for visual aid. **e**, Schematic of the experimental design and mathematical modelling. Pulse–chase protein

concentration data were used to inform the model and estimate protein turnover rates. Dilution factors due to ovary growth were estimated by the model and compared to experimental ovary growth measurements for validation (indicated by a grey dashed arrow). **f**, Graphical illustration of the employed protein turnover model to estimate $H_{1/2}$ values in ovaries. **g,j**, Example (histone H4) of protein turnover model fitting. Experimental data are indicated as black dots, median and confidence ranges of model fit are indicated by the red line and shaded area, respectively. **g,h**, The ratio ($R$) between heavy ($H$) and light ($L$) labelled protein intensities in the chase ($^{12}C_6$-Lys) from birth (**g**) and from weaning (**h**) experiment datasets are shown over time. **i**, Normalized protein abundance derived from DIA-MS measurements is shown over time for histone H4. **j**, Proportion of heavy labelled histone H4 compared with all proteins as derived from the estimated protein turnover posterior parameter distribution.

The 39 proteins that were still >90% positive for $^{13}C_6$-Lys included many proteins with key cellular functions, such as TWINKLE, a mitochondrial DNA helicase that is essential for mitochondrial DNA replication[7]; TTC19, which is essential for the assembly and function of complex III of the mitochondrial respiratory chain[8]; BOP1, a protein that is essential for ribosome assembly[9]; and HIRA, a histone chaperone (Fig. 1c). We also detected cohesin-related proteins as long-lived, including the cohesin subunits SMC3 and SMC1A, as well as PDS5B (Fig. 1c and Supplementary Table 1), which regulates the association of the cohesin complex with chromatin[10–14]. The known long-lived cohesin subunit REC8 (refs. 15–17) was not detected by MS, likely because it is present at very low levels in oocytes.

Together, these data establish that oocytes contain many extremely long-lived proteins. Mitochondrial proteins and proteostasis-promoting proteins were prominently enriched in this group, and hence persist with little turnover in oocytes from birth onwards.

## Extreme longevity of ovarian proteins

To examine protein longevity in the whole ovary, which contains the oocytes as well as somatic cells, such as granulosa, thecal and stromal cells, we performed two different pulse–chase experiments. First, we repeated our approach above, in which females were labelled with $^{13}C_6$-Lys during development in utero, and collected 33 ovaries at 11 time points, from 24 hours to 65 weeks of age (Fig. 2a)[18]. Second, fully labelled $^{13}C_6$-Lys pups were kept with fully labelled mothers until weaning and then fed $^{12}C_6$-Lys chow. Ovaries were collected at six time points from 6 weeks to 65 weeks of age (Fig. 2b). The second approach labels all proteins synthesized both in utero and after birth until the end of the first wave of oocyte follicle growth[2,4]. This dual strategy enabled a comprehensive analysis of the ovary proteome, which changes with age.

All samples were analysed by MS using data-dependent acquisition (DDA; Fig. 2c,d). In addition, we isolated ovaries from unlabelled mice of eight different ages and analysed the relative abundances of the proteins throughout ovarian development using data-independent acquisition (DIA) MS (Supplementary Data 1).

The turnover rates were determined by mathematical modelling (Fig. 2e,f, Methods and Extended Data Fig. 3a). Briefly, for each protein, we calculated the $^{12}C_6$-Lys ($L$) to $^{13}C_6$-Lys ($H$) ratio as $R = \ln(L/H + 1)$ at the different time points (Fig. 2g,h). These experimentally determined ratios were then compared to ratios obtained from a protein turnover model (Fig. 2f). The protein turnover model describes the abundance of $^{13}C_6$-Lys-labelled proteins over the chase period, taking into account protein degradation (with rate $k$) and protein synthesis (with rate $k_{on}$) due to recycling of free (unincorporated) $^{13}C_6$-Lys. The rate $k_{on}$ is defined as a function of the free $^{13}C_6$-Lys pool, such that the higher the remaining free $^{13}C_6$-Lys, the higher the rate of $^{13}C_6$-Lys protein synthesis rate

due to $^{13}C_6$-Lys recycling, and, hence, the slower the decay of $^{13}C_6$-Lys labelled protein.

We determined the free $^{13}C_6$-Lys and $^{12}C_6$-Lys pools throughout ovarian development by analysing the concentration of different species of uncleaved peptides containing two lysines[19,20] (Extended Data Fig. 3b–e). 2Lys peptides can either be composed of two $^{13}C_6$-Lys, two $^{12}C_6$-Lys or a mixture, allowing for the inference of the free $^{13}C_6$-Lys pool[19,20] (Extended Data Fig. 3c–e). As additional parameters, ovarian growth and changes in individual protein concentrations (DIA-MS data) were included in the modelling to account for changes in ovarian protein composition (Extended Data Fig. 3a; Methods). The resulting modelled turnover rates reflect the average behaviour of all proteins in the ovary, and cannot discern whether protein turnover differs depending on interactions or cell types.

To challenge our approach, we modified our model in various ways and compared the resulting half-lives to the full modelling approach (Extended Data Fig. 4a–d). Specifically, we compared our protein-centric turnover model including re-incorporation of free $^{13}C_6$-Lys into newly synthesized proteins (Fig. 2f) to (1) a peptide-centric 2Lys-peptide model (Extended Data Fig. 4a) and (2) a protein-centric model not including re-incorporation of free $^{13}C_6$-Lys into newly synthesized proteins (Fig. 2f with $k_{on} = 0$) using either all chase time points (Extended Data Fig. 4b), or only chase time points larger than 3 weeks or 6 weeks, respectively (Extended Data Fig. 4c,d). The 2Lys-peptide approach takes advantage of the fact that some very abundant proteins have enough 2Lys peptides resulting from missed tryptic cleavages during sample preparation, so that only peptides with two $^{13}C_6$-Lys can be taken into consideration for modelling, which are highly unlikely to be generated by reincorporation of $^{13}C_6$-Lys, and the free $^{13}C_6$-Lys pool can then be neglected. The half life ($H_{1/2}$) values calculated in this way were well consistent with our protein-centric turnover model. In addition, we also calculated the $H_{1/2}$ values from late time points only (larger than 3 weeks or 6 weeks). Growth has substantially slowed down at this stage and the free $^{13}C_6$-Lys pool is strongly depleted. In further support of our model, modelling of these late time points only (allowing for exclusion of the free $^{13}C_6$-Lys pool) gave very similar results to our full protein-centric model using all collected data and time points (Extended Data Fig. 4c,d).

Finally, we employed our protein-centric model to pulse–chase data derived from mouse liver, cartilage and skeletal muscle published previously[20] to estimate protein half-lives, and to compare them to the original published values (Extended Data Fig. 4e–g). For all three datasets, we obtained good agreement of inferred half-lives with correlation coefficients of 0.85, 0.96 and 0.88 for liver, cartilage and skeletal muscle, respectively, increasing the confidence in our modelling approach.

We summarized all the data in an atlas of 3,078 ovarian proteins (Supplementary Data 2), which shows the raw $^{12}C_6$-Lys and $^{13}C_6$-Lys

**Fig. 3 | Mouse ovaries have a >10-fold higher fraction of extremely long-lived proteins than other post-mitotic tissues. a**, Distribution of the modelled $H_{1/2}$ values in the ovary. **b**, Fraction of proteins with $H_{1/2}$ > 100 days in the mouse ovary samples in this study and various mouse and rat tissues across three studies[19–21]. **c**, Distributions of estimated $H_{1/2}$ values for proteins located in indicated subcellular compartments. Red and blue stars indicate significantly larger and smaller $H_{1/2}$ distributions compared with the whole modelled proteome, respectively (two-sided Kolmogorov–Smirnov test; $P$ values: nucleus 0.0093; mitochondrion $8.84 \times 10^{-6}$). Number of proteins in each compartment indicated in parentheses. Boxplots indicate median, first quartile and third quartile, as well as minimum and maximum after outlier removal. **d**, Cluster analysis of inferred $^{13}C_6$-Lys levels in the proteins of the aging ovaries. The dendrogram on the left corresponds to the clustering of inferred percentage $^{13}C_6$-Lys. The leftmost bar shows the medians of inferred proteins half-lives, colouring on $\log_{10}$ scale. The second and third bars indicate the latest time point at which the $^{13}C_6$-Lys pulse was detected in the data corresponding to chase from birth and weaning. The rightmost bar labels the three identified protein longevity clusters corresponding

to proteins with high, intermediate and low amounts of inferred $^{13}C_6$-Lys content. The leftmost heatmap shows the inferred percentage $^{13}C_6$-Lys; time points from 6 weeks onwards were used for clustering. Central and rightmost heatmaps show experimental data of the chase $^{12}C_6$-Lys from birth and weaning, respectively. **e**, Modelled percentage of $^{13}C_6$-Lys labelled proteins compared to $^{12}C_6$-Lys labelled proteins was used to derive three protein longevity clusters (low, intermediate and high inferred $^{13}C_6$-Lys content). Bar plot showing number of proteins in the low, intermediate and high protein longevity clusters. **f**, Violin plot showing distributions of inferred half-lives of proteins in the low, intermediate and high protein longevity clusters. Boxplots indicate median, first quartile and third quartile, as well as minimum and maximum after outlier removal. Number of proteins in each cluster as indicated in **e**. **g–i**, Dot plots comparing the $H_{1/2}$ values of different proteins in the ovary ($x$ axis) with organs and tissues ($y$ axis) measured in ref. 19 (**g**), ref. 21 (**h**) and ref. 20 (**i**, liver; **j**, cricoid cartilage). Test for association between paired samples using Spearman correlation coefficient ($C$) was performed with $P$ values estimated using algorithm AS 89. Proteins are colour-coded according to the protein longevity cluster they belong to, as in **d**.

intensities for each of the quantified proteins as determined by MS over time, as well as the modelled data (considering free $^{13}C_6$-Lys, ovarian growth and changes in protein abundance), $H_{1/2}$ values (the number of days it takes the $^{13}C_6$-Lys protein fraction to decrease by half; summarized in Extended Data Fig. 4h) and the changes in the abundance

of ovarian proteins throughout development (Fig. 2g–j and Supplementary Table 3).

A subset of 958 $^{13}C_6$-Lys proteins rapidly disappeared from the ovary during the chase ($^{12}C_6$-Lys) in both experiments (Supplementary Table 4). These proteins could not be modelled due to their extremely

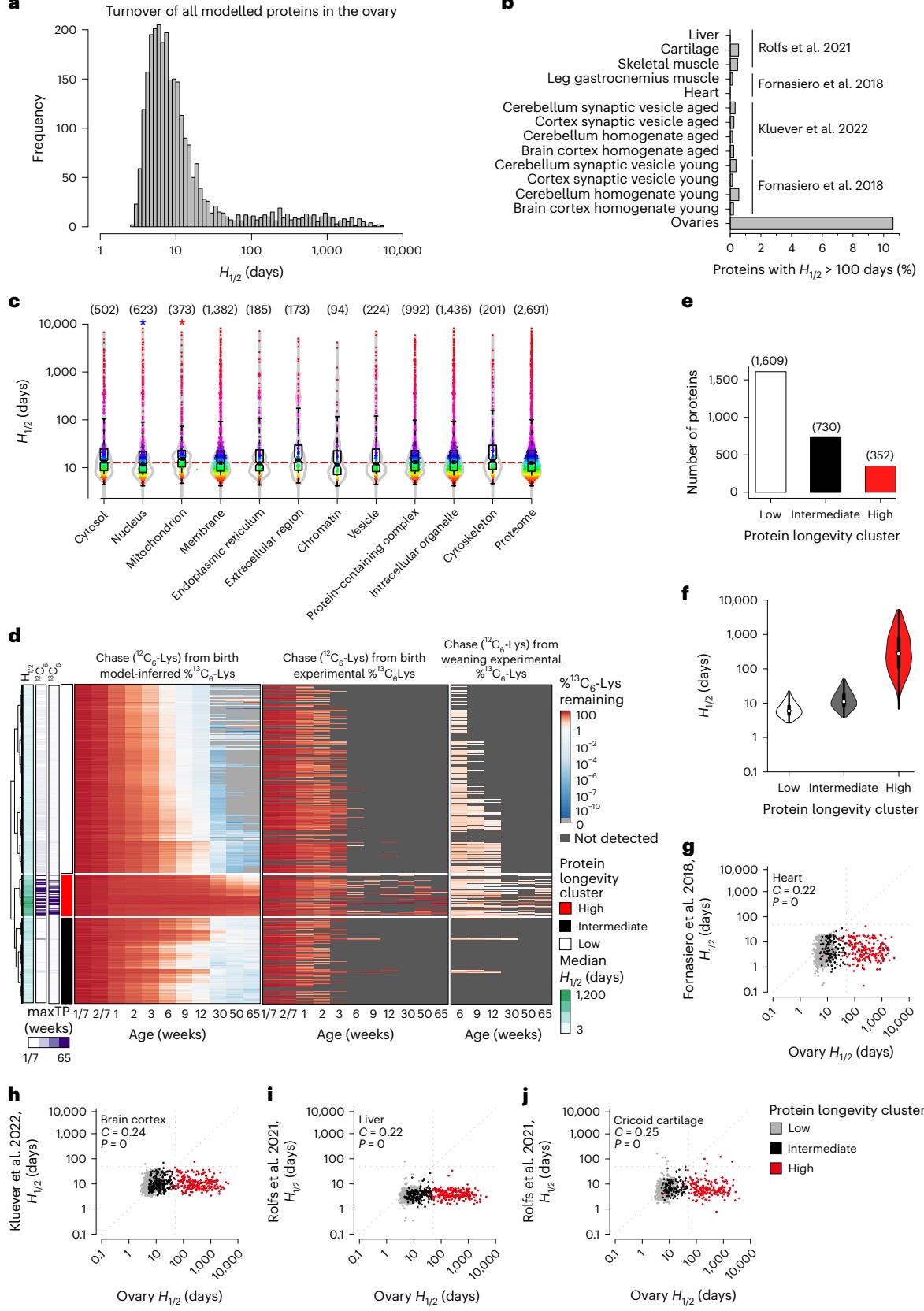

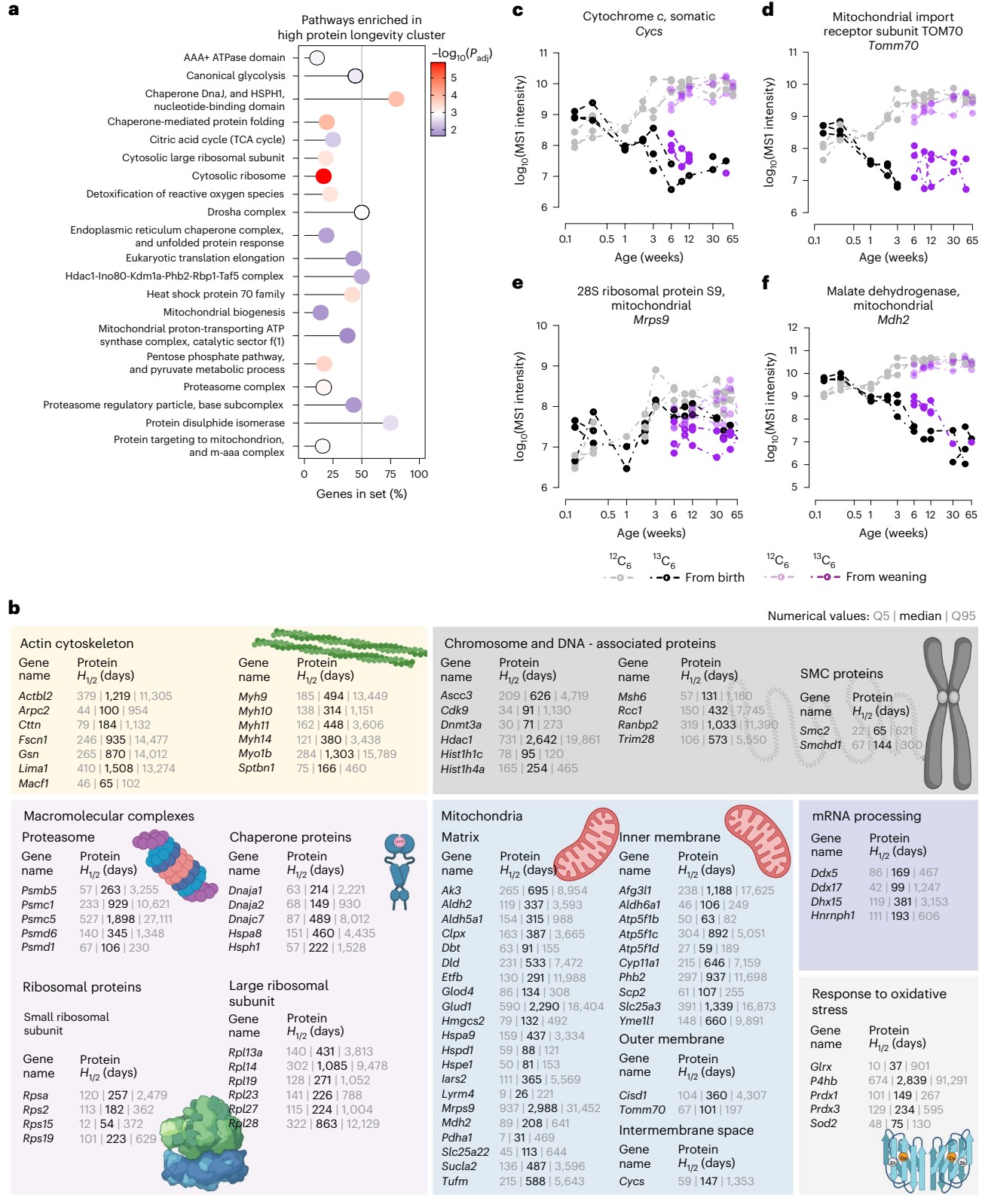

**Fig. 4 | The ovary contains long-lived proteins that persist throughout the lifetime of a mouse. a**, Selected pathways enriched in the high protein longevity cluster. Shown are the proportions of genes of the selected gene set detected in the high protein longevity cluster with their respective adjusted *P* values for selected pathways. The *P* values are based on the hypergeometric test and have been adjusted with Benjamini–Hochberg multiple hypothesis testing. For the complete list of enriched pathways with their corresponding exact *P* values, see Supplementary Table 5. **b**, Selected biological processes and protein complexes with long-lived proteins in ovaries. Shown are gene names with corresponding protein $H_{1/2}$ values (medians, 5% and 95% quantiles; designated as Q5 and Q95, respectively) in days. For the complete list of proteins and their corresponding $H_{1/2}$ values, see Supplementary Table 3 and Supplementary Data 2. **c–f**, MS1 intensities over time in $\log_{10}$ scale for $^{13}C_6$-Lys (black and purple) and $^{12}C_6$-Lys (grey and pink) labelled cytochrome *c*, somatic (**c**); mitochondrial import receptor subunit TOMM70 (**d**); 28S ribosomal protein S9, mitochondrial (**e**); and malate dehydrogenase, mitochondrial (**f**) for two pulse lengths (pulse with $^{13}C_6$-Lys until birth and pulse with $^{13}C_6$-Lys until weaning). Circles indicate experimental data points for three biological replicates, dashed lines are only for visual aid.

rapid degradation. This group was enriched in proteins involved in cell cycle regulation, cell division and DNA replication, among others, consistent with the short lifetime of proteins involved in these processes (Extended Data Fig. 4i).

The distribution of $H_{1/2}$ values in the ovary had a peak at around ~8–10 days, similar to protein turnover in other organs (Fig. 3a and Extended Data Fig. 5a)[19–21]. However, over 10% of the proteins in the ovary had $H_{1/2}$ values above 100 days, whereas fewer than 1% of proteins in other organs, including brain, muscle and cartilage, had such high $H_{1/2}$ values (Fig. 3b and Supplementary Data 3)[19–21]. Although these extremely long-lived proteins were not restricted to specific cellular locations, mitochondrial proteins had longer $H_{1/2}$ values, on average, when compared to the total ovary proteome, whereas nuclear proteins had shorter $H_{1/2}$ values (Fig. 3c). Many mitochondrial proteins (for example, cytochrome $c$, ATP synthase subunits, malate dehydrogenase and mitochondrial chaperone HSPA9) had $H_{1/2}$ values that exceeded those reported in brain tissues (Supplementary Data 3).

Using the CORUM database of protein complexes[22], we assessed the distribution of the $H_{1/2}$ values of a wider range of complexes in the ovary (Extended Data Fig. 5b). The cytoplasmic ribosome, proteasome and spliceosome contained many long-lived proteins with $H_{1/2}$ values of more than 50 days. Other long-lived complexes included the CCT complex, which is involved in the folding of actin and tubulin, and the transcription integrator complex. Shorter-lived proteins were observed in the MCM complex, the SWI/SNF chromatin remodelling complex and the transcription mediator complex (Extended Data Fig. 5b).

We clustered the proteins according to their longevity (Fig. 3d), based on the modelled $^{13}C_6$-Lys protein fractions from 6 weeks until 65 weeks of age. This resulted in three distinct protein longevity clusters, corresponding to proteins with high, intermediate and low relative $^{13}C_6$-Lys levels (Fig. 3d,e and Supplementary Table 3). These clusters comprised 352 extremely long-lived proteins with a mean $H_{1/2}$ value of 681.0 days, 730 long-lived proteins with a mean $H_{1/2}$ value of 14.6 days and 1,609 proteins with a mean $H_{1/2}$ value of 7.0 days (Fig. 3f). Proteins in the high and intermediate protein longevity clusters had significantly higher $H_{1/2}$ values in the ovary than in other tissues (Fig. 3g–j)[19–21].

The 352 extremely long-lived proteins in the ovary contained the long-lived proteins detected in oocytes (Extended Data Fig. 5c) and showed the same over-representation of mitochondria-related and proteostasis-related proteins as mouse oocytes (Fig. 4a,b, Supplementary Table 5 and Supplementary Data 3). Many of these long-lived proteins were $^{13}C_6$-Lys-positive from birth until the last analysed time point at 65 weeks (Fig. 4c–f, Extended Data Fig. 5d–f and Supplementary Data 2), and have thus persisted in the mouse ovary for 15 months, almost the entire lifespan of FVB/N females.

To give some specific examples, the mitochondrial proteins cytochrome $c$ (CYC; Fig. 4c), the mitochondrial import receptor subunit TOMM70 (Fig. 4d), the mitochondrial 28S ribosomal protein S9 (MRPS9; Fig. 4e), mitochondrial malate dehydrogenase (MDH2; Fig. 4f) and glutamate dehydrogenase 1 (GLUD1; Extended Data Fig. 5d), as well as the heat shock proteins and chaperones DNAJC7, DNAJA1, DNAJA2, HSPH1, HSPA2, HSPA8, HSPA9 and HYOU1, were extremely long-lived (Fig. 4b and Supplementary Data 3). Notably, DNAJA1 protects cells

against apoptosis by negatively regulating the translocation of BAX from the cytosol to mitochondria[23,24] (Fig. 4b). Another extremely long-lived protein was SMCHD1, which is a non-canonical member of the structural maintenance of chromosomes (SMC) protein family that plays a key role in epigenetic silencing, and is required for X-inactivation, by mediating XIST spreading (Fig. 4b)[25,26]. Other proteins involved in epigenetic regulation (including DNMT3A and HDAC1; Fig. 4b, Extended Data Fig. 5e and Supplementary Data 3), as well as proteins involved in nuclear import and export, including RAN, RANBP2, RCC1 and CSE1L (Fig. 4b) and myosins (MYH9, MYH10, MYH11, MYH14 and MYO1B; Extended Data Fig. 5f) were in the high longevity cluster. The extremely long-lived proteins in the ovary thus have essential functions in mitochondria, proteostasis, chromatin maintenance and the cytoskeleton.

## Discovery of long-lived somatic cell types in the ovary

We used single-cell RNA-seq to identify the ovarian cell types that express long-lived proteins (Supplementary Data 4). We examined ovaries from newborn mice (postnatal day 2), as this age refers to the end of the first pulse period, and to the time when the ovarian reserve has been established. Interestingly, some proteins from the high longevity cluster mapped to distinct cell types based on a greater than twofold enrichment of their transcripts (Fig. 5a,b). These cell types included not only oocytes (germ cells), which are generally assumed to reside in the ovary from birth onwards[3], but also subsets of somatic cells (Fig. 5b,c). These data suggest that the ovary contains long-lived somatic cells.

To test this hypothesis, we used NanoSIMS, which is suitable to assess the local $^{13}C/^{12}C$ ratio in a tissue[27]. We collected ovaries from mice that had obtained $^{13}C_6$-Lys feed until weaning (3 weeks of age), and had subsequently obtained $^{12}C_6$-Lys feed until 4 (Fig. 5d–g) or 8 weeks of age (Fig. 5h–k). We then measured the ratio of $^{13}C/^{12}C$ in oocytes, granulosa cells (surround the oocyte inside follicles), theca cells (line the surface of follicles), luteal cells (form the corpus luteum that develops from the follicle upon ovulation) and stromal cells (cells that reside between follicles in the ovary).

Cells of the short-lived corpus luteum had the lowest $^{13}C/^{12}C$ ratio of all cell types (Fig. 5e,i). Interestingly, the highest $^{13}C/^{12}C$ ratios were observed for subsets of stromal cells, granulosa cells, theca cells and oocytes (Fig. 5e). Then we examined the $^{13}C/^{12}C$ ratio for granulosa cells from follicles of different sizes (Fig. 5f,j). The ratios were highest for granulosa cells in small primordial and primary follicles, suggesting that the follicle cells that surround the oocytes during storage are particularly long-lived, and that their long-lived proteins are diluted with newly synthesized proteins as the follicles grow. We cannot currently determine which exact proteins are long-lived in each of these cell types, but this may be possible with future developments in single-cell MS[28]. We conclude that not only oocytes and their proteins are long-lived in the ovary, but also subsets of somatic cells and their proteins, including granulosa and stromal cells, as well as individual cells in the theca.

## Proteostasis is maintained in aged oocytes

Impaired proteostasis can lead to the accumulation of misfolded proteins, resulting in protein aggregation, which is a major pathological

**Fig. 5 | Subsets of granulosa, stromal and theca cells are long-lived in the ovary. a**, Proportion of the proteins from the high protein longevity cluster with greater or less than twofold enrichment of corresponding transcripts in specific cell types in the ovaries of postnatal day 2 mice. **b**, Distribution of the transcripts of the proteins from the high protein longevity cluster with greater than twofold enrichment in specific cell types of the ovaries of postnatal day 2 mice. **c**, Dot plot showing the expression patterns of the transcripts of the proteins from the high protein longevity cluster. Size of the dot represents the proportion of the cells expressing the gene, colour denotes $\log_2$(fold change) in the one versus all cell types differential gene expression test. **d–k**, NanoSIMS imaging of ovaries from 4-week-old (**d–g**) and 8-week-old (**h–k**) mice that were pulsed with $^{13}C_6$-Lys until weaning (3 weeks after birth), followed by a chase period with $^{12}C_6$-Lys. **d,h**, Histological sections of ovarian tissue (first column on the left). Insets show magnified areas highlighted in the histological sections (middle section of **d** and **h**). $^{13}C_6/^{12}C_6$ ratio image for the indicated histological sections and corresponding values are given on the right-hand side of the magnified insets. **e,i**, Dot plots showing $^{13}C_6/^{12}C_6$ signal in different ovarian cell types. **f,j**, Dot plots showing $^{13}C_6$ $/^{12}C_6$ signal of granulosa cells in the primary, secondary and antral follicles. **g,k**, Dot plots showing $^{13}C_6/^{12}C_6$ signal of theca cells in the primary, secondary and antral follicles. Numbers of analysed cells are shown in parentheses. $P$ values were calculated using unpaired two-tailed Student's $t$-test. n.s., not significant; *$P \le 0.05$; **$P \le 0.01$; ***$P \le 0.001$; ****$P \le 0.0001$. Scale bars, 10 μm.

hallmark of various age-related disorders such as Alzheimer's and Parkinson's diseases[29]. To investigate the vulnerability of oocytes to age-related protein aggregation, we stained aggresomes in fully grown

oocytes from young (9 weeks) and aged (65 weeks) mice with the ProteoStat dye (Fig. 6a). Strikingly, aggresomes were not increased in aged oocytes compared to young oocytes (Fig. 6a–c). By contrast,

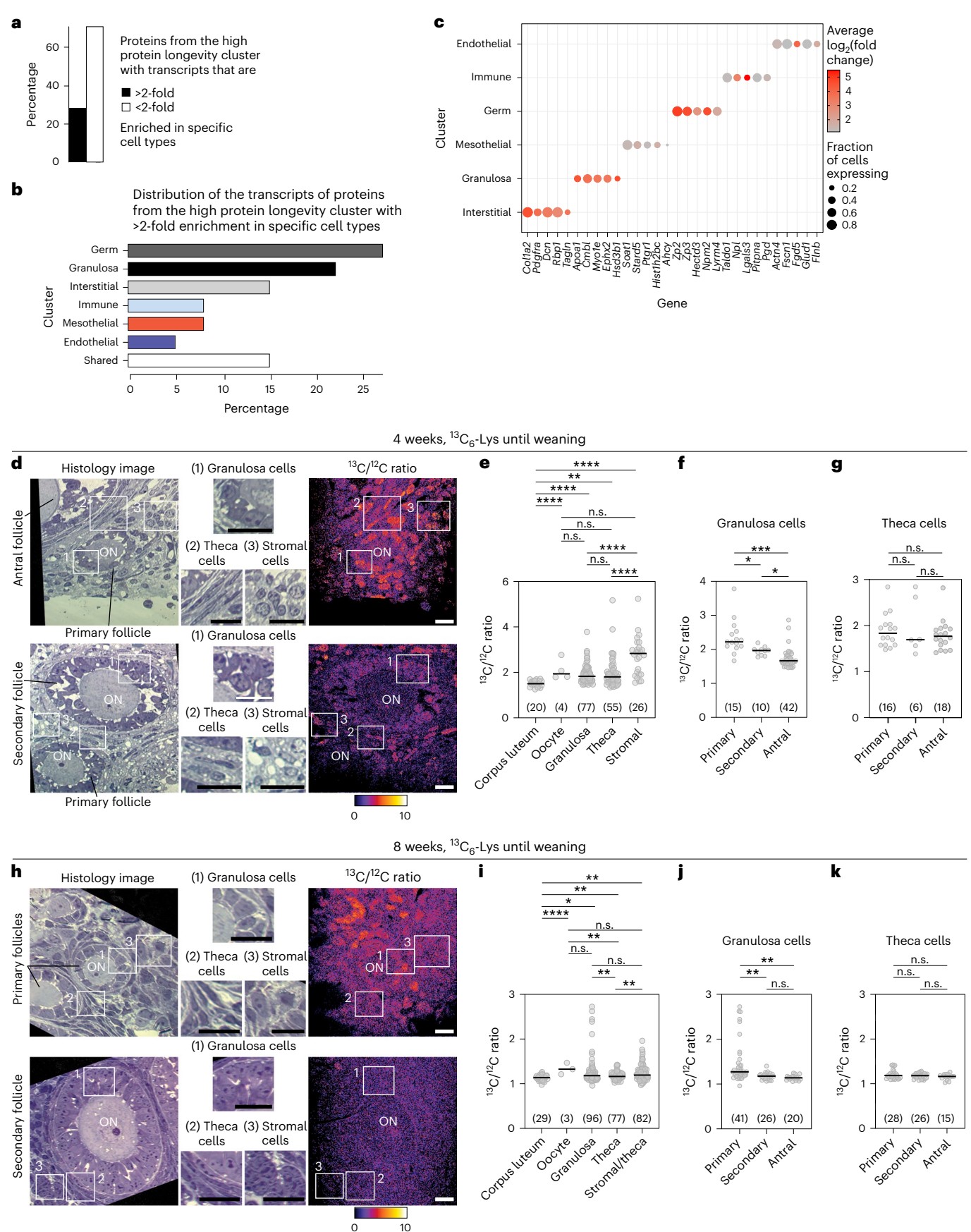

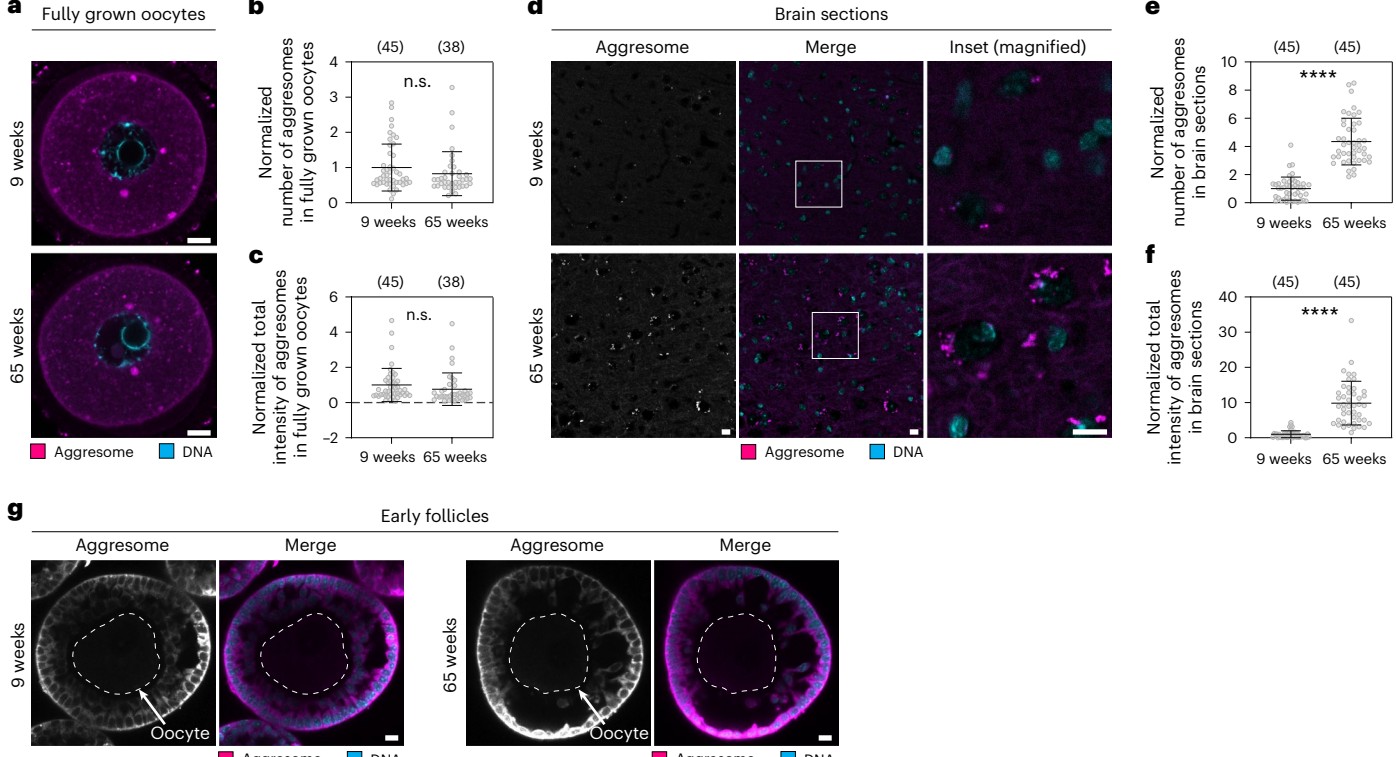

**Fig. 6 | Protein aggregation does not increase in aged oocytes. a,** Representative immunofluorescence images of fully grown, germinal vesicle stage mouse oocytes from 9-week-old and 65-week-old mice stained with the ProteoStat aggresome dye. Magenta, aggresome (ProteoStat); cyan, DNA (Hoechst). **b,** Dot plot showing number of ProteoStat-positive structures in oocytes as shown in **a. c,** Dot plot showing total intensity of ProteoStat-positive structures in oocytes as shown in **a. d,** Representative immunofluorescence images of brain slices from 9-week-old and 65-week-old mice stained with the ProteoStat aggresome dye. Magenta, aggresome (ProteoStat); cyan, DNA (Hoechst). **e,** Dot plot showing number of ProteoStat-positive structures in brain slices as shown in **d. f,** Dot plot showing total intensity of ProteoStat-positive structures in brain slices as shown in **d. g,** Representative immunofluorescence images of early follicles from 9-week-old and 65-week-old mice stained with the ProteoStat aggresome dye. Magenta, aggresome (ProteoStat); cyan, DNA (Hoechst). No obvious aggresome accumulation was detected in either age group. All data from two independent experiments. Number of analysed oocytes and brain areas are in parentheses. Data are shown as mean ± s.d. *P* values were calculated using unpaired two-tailed Student's *t*-test. n.s., not significant; ****$P \leq 0.0001$. Scale bars, 10 μm.

we observed increased aggresome staining in brain sections of aged mice compared with young mice, consistent with previous studies[30] (Fig. 6d–f). This indicates that protein aggregation does not increase with age in oocytes. In line with this result, aggresomes were also not increased in early follicles from aged mice (Fig. 6g).

Another hallmark of impaired proteostasis is a decay in proteasomal activity[31]. We therefore compared proteasomal activity in aged and young oocytes with a reporter assay[32]. In this assay, a reporter construct consisting of the proteasome substrate Ub(G76V)-mClover3 and mScarlet, separated by a self-cleaving T2A peptide, is expressed. The proteasomal activity can be inferred from the ratio of mClover and mScarlet signal intensity. As expected, proteasome-inhibited oocytes (MG-132) expressing this construct accumulated both mScarlet and Ub(G76V)-mClover3, whereas control oocytes (DMSO) only accumulated mScarlet but not Ub(G76V)-mClover3 due to constant proteasomal degradation (Fig. 7a–c). Ub(G76V)-mClover3 was efficiently degraded in both young (9-week) and aged (65-week) oocytes (Fig. 7d,e), with no significant difference in the mClover3/ mScarlet ratio.

To compare the degradation kinetics in both age groups, both young and aged oocytes were allowed to first accumulate Ub(G76V)-mClover3 and mScarlet by MG-132 inhibition, followed by MG-132 washout and live imaging (Fig. 7f). Ub(G76V)-mClover3 was degraded at a similar rate in both young and aged oocytes, with a slightly higher degradation speed in aged oocytes (Fig. 7g,h). Together, these data establish that proteasomal activity does not decay in aged oocytes.

## Important long-lived proteins are lost with ovarian aging

Together, our data establish that proteostasis in oocytes and the ovary involves extreme longevity of a wide range of proteins. Although slow protein turnover can have a positive effect on cellular and organismal longevity[33–39], the eventual decline in long-lived proteins can contribute to aging[15,40–42]. We hence investigated changes in the protein composition of the ovary during ovarian aging (Extended Data Fig. 6a). To this end, we quantified the abundance of 8,890 proteins in ovaries from females at 1 day, and 1, 2, 3, 5, 9, 12 and 50 weeks of age using DIA-MS analysis (Extended Data Fig. 6a, Supplementary Table 6 and Supplementary Data 1).

First, we performed a cluster and gene set enrichment analysis (GSEA) on the data (Supplementary Table 7). GSEA revealed a significant upregulation of pathways associated with active cell division and protein biogenesis (including translation, mitotic spindle checkpoint, RNA processing, ribosome biogenesis and meiotic cell cycle) at 3–5 weeks of age, followed by a steep downregulation of these pathways at 9 weeks of age, when FVB/N female mice have reached full fertility (Extended Data Fig. 6a,b). Interestingly, proteins related to aerobic respiration and oxidative phosphorylation (Extended Data Figs. 6b and 7a,b), microbodies, steroid metabolic process, lipid biosynthesis and mitophagy increased around this time. Proteins related to the regulation of acute inflammatory response (Extended Data Figs. 6b and 7c) and the humoral immune response (Extended Data Figs. 6b and 7d) gradually increased in abundance from 3 weeks onwards, reaching a maximum at 50 weeks, which is consistent with the reported increase in inflammation in the ovary with advancing female age[43].

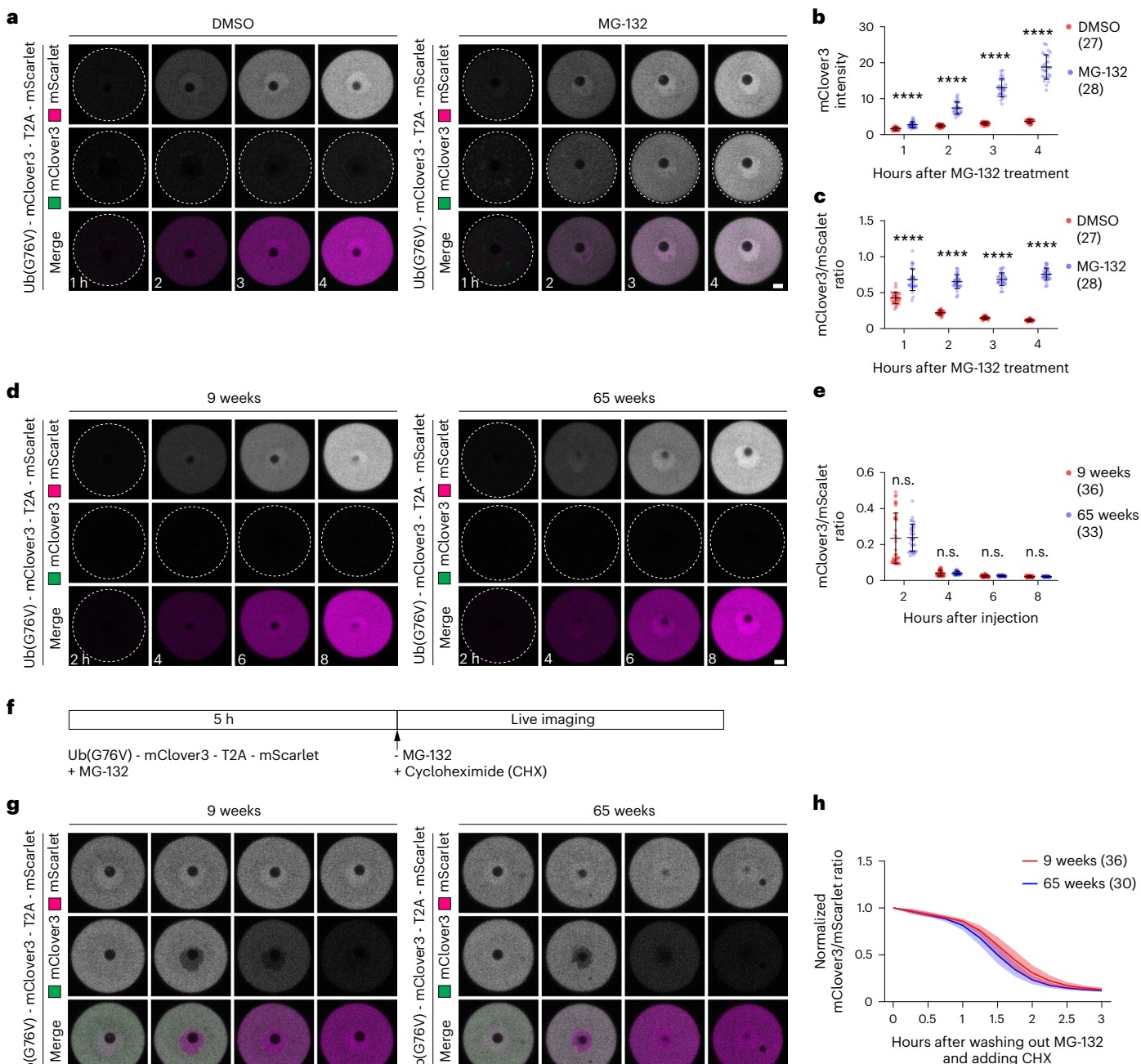

**Fig. 7 | Proteasomal activity does not decay in aged oocytes. a**, Time-lapse images of mouse oocytes from 9-week-old mice expressing Ub(G76V)-mClover3-T2A-mScarlet in the presence of DMSO or 10 μM MG-132. Time is given as hours after DMSO or MG-132 treatment. **b**, Quantification of the mean fluorescence intensity of mClover3 in oocytes as shown in **a**. **c**, Quantification of the fluorescence intensity ratio of mClover3 to mScarlet in oocytes in **a**. **d**, Time-lapse images of mouse oocytes from 9-week-old and 65-week-old mice expressing Ub(G76V)-mClover3-T2A-mScarlet. Time is given as hours after injection of the reporter mRNA. **e**, Quantification of the fluorescence intensity ratio of mClover3 to mScarlet in oocytes in **d**. **f**, Schematic diagram of the experiment shown in **g**. Ub(G76V)-mClover3 and mScarlet were expressed for 5 h in the presence of

MG-132, which blocks the degradation of Ub(G76V)-mClover3. MG-132 was then washed out and oocytes were imaged in the presence of the translation inhibitor cycloheximide (CHX), which blocks the synthesis of new proteins. **g**, Time-lapse images of mouse oocytes from 9-week-old and 65-week-old mouse oocytes expressing Ub(G76V)-mClover3-T2A-mScarlet. Experiment was performed as shown in **f**. Time is given as hours after MG-132 washout and CHX wash-in. **h**, Line graph showing normalized fluorescence intensity ratio of mClover3 to mScarlet in oocytes in **g**. All data from two independent experiments. Number of analysed oocytes in parentheses. Data are shown as mean ± s.d. *P* values were calculated using unpaired two-tailed Student's *t*-test. n.s., not significant; ****$P \leq 0.0001$. Scale bars, 10 μm.

We then focused on the time period from when the females were fully fertile (9 weeks and 12 weeks), to when fertility has markedly declined (50 weeks; Fig. 8a). First, we performed a protein abundance cluster analysis, which revealed six groups of proteins with distinct trends in relative abundance during this period (Fig. 8b and Supplementary Table 8).

Protein abundance clusters 2 and 4 were of particular interest as they contained proteins that were abundant in 9- and 12-week-old females, but dropped steeply in 50-week-old females, when female fertility declines (Fig. 8c,d). Cluster 2 was significantly enriched in processes with important functions related to ovarian aging or aging

in general, including stem cell division proteins, telomere proteins, heterochromatin assembly proteins and DNA double-strand break repair proteins (Fig. 8c and Supplementary Table 9). Cluster 4 was enriched in proteins involved in autophagy (Fig. 8d and Supplementary Table 9), which can promote cellular longevity[44], as well as many heat shock proteins and chaperones, which promote cellular longevity by protecting cells against protein misfolding and aggregation[38,39]. Cluster 4 also contained zona pellucida proteins, which are required for sperm binding, as well as cohesion proteins. Ovarian aging has been attributed to multiple processes that overlap with the enriched gene ontology terms, including a decline in the ability to repair DNA double-strand breaks[45–47], a decline of telomere function[48,49], alterations in the oocyte's zona pellucida[50,51] and a decline in cohesion function[52–55]. Changes in the levels of related proteins could contribute to these defects, and also reflect alterations in the cellular composition of the aging ovary, including a decline in follicle numbers.

Protein abundance cluster 5 contained proteins that were steeply upregulated in the ovaries of 50-week-old females (Fig. 8b) and was enriched in inflammatory response proteins, immune proteins, proteins involved in retinoic acid synthesis, detoxification-related proteins, response to stress proteins, oxygen-related proteins, proteins related to mitogen-activated protein kinase (MAPK) signalling, extracellular matrix proteins and hormone biosynthesis proteins (Fig. 8e and Supplementary Table 9). These ontology terms overlap with known alterations in the aging ovary (that is, an increase in inflammation and fibrosis)[43,56], and reveal additional candidate pathways that have not yet been linked to ovarian aging, including for instance a potential function for retinoids in the aging ovary.

Importantly, many extremely long-lived proteins also declined in the aging ovary (Fig. 8f; Extended Data Fig. 8). Proteins with protective functions and proteins of the proteostasis network were prominently enriched in this group, including chaperones, heat shock proteins, chaperonins, disulfide isomerases, proteasomal and ribosomal proteins, and other proteins that promote protein folding or protein quality control (Fig. 8g and Supplementary Table 10). Thus, a large number of long-lived proteins with key functions in maintaining proteostasis in cells were significantly decreased in the aged ovary. In addition, long-lived proteins involved in other essential processes, such as heterochromatin organization, were decreased (Fig. 8g). We conclude that ovarian aging is associated with extensive changes in the ovarian proteome and a decrease of important long-lived proteins.

## Discussion

Our data establish that many proteins in oocytes and the ovary are unusually stable, with half-lives well above those reported in other cell types and organs, including the liver, heart, cartilage, muscle and the brain[19–21,41,57,58]. Our data confirm the longevity of various proteins, including histones, components of nuclear pore complexes and mitochondria[40,59–62]. Unexpectedly, however, the half-lives of many proteins are much higher in the ovary than in other organs, and many additional proteins are uniquely long-lived in the ovary. These include many proteins involved in proteostasis, and in critical cellular machines, such as mitochondria, ribosomes, proteasomes and cytoskeletal assemblies. In addition, proteins in key metabolic pathways, such as glycolysis,

the pentose phosphate pathway, the citrate cycle and fatty acid beta oxidation are extremely long-lived.

Oocytes must be stored in the ovary for more than a year in mice, and for decades in humans. Our study shows that germline maintenance involves extraordinary stability of hundreds of proteins, many of which persist throughout the life of the mouse. This extreme protein stability could provide major advantages for safeguarding the germline across generations.

Oocytes and their supporting somatic cells are formed before birth, but only become activated again much later, when puberty resumes[63]. This likely allows for adaptations in proteostasis that would not be possible in other cell types. A reduction in protein turnover has been shown to promote proteostasis, cellular longevity and high organismal lifespan[33–37]. Oocytes might similarly use this strategy to preserve a wide range of their components. For example, oocytes provide all of the mitochondria to the developing embryo[64,65] and must therefore preserve mitochondria with minimal damage. A wide range of mitochondrial proteins were extremely stable in both oocytes and ovaries, indicating that mitochondrial components persist for exceptionally long periods of time.

High protein turnover is associated with ATP consumption[66,67], and therefore requires high mitochondrial activity, which generate reactive oxygen species[68], a major driver of cellular damage. Notably, oocyte mitochondria are largely inactive[69], and many longevity promoting interventions reduce mitochondrial activity and protein synthesis[70,71]. Thus, low mitochondrial activity and low protein turnover may be regulatory and functionally coupled in mammalian oocytes. In support of this point, the levels of mTOR signalling-related proteins suggest that mTOR signalling is relatively low in young ovaries and increases with age (Extended Data Fig. 8c). Low mTOR signalling leads to low mitochondrial activity and low protein turnover[72–74], and is therefore likely to be a regulatory mechanism that drives high protein longevity and low mitochondrial activity in the young ovary.

The extreme longevity of mitochondrial proteins and core cellular machinery is accompanied by an extreme longevity of several chaperones and heat shock proteins, including the major mitochondrial heat shock protein HSPD1. Proteins that protect against oxidative damage, such as peroxiredoxins, SOD2 and disulfide isomerases, were also extremely long-lived. These proteins likely promote the high longevity, proper folding and function of other proteins in the oocyte and ovary. In addition, the compartmentalization of subsets of proteins into specialized structures, such as the Balbiani bodies[75,76] or cytoplasmic lattices[77] may help to preserve their stability and function.

Our data suggest that these adaptations in proteostasis help to prevent the accumulation of damage in the germline. Notably, protein aggregation did not increase in aged oocytes and early follicles (Fig. 6a–c,g), and proteasomal activity was not diminished (Fig. 7d–h). This is in stark contrast to the brain, which showed a significant increase in protein aggregation over the same time period (Fig. 6d–f) and exhibits a decay in proteasomal function[29–31].

Although reduced protein synthesis has beneficial effects on cellular longevity and proteostasis, the eventual loss of highly stable proteins may contribute to the demise of oocytes and the ovary with age. Previous work has shown that the cohesin subunit REC8 is long-lived

---

**Fig. 8 | Proteostasis networks are lost with ovarian aging and decreased fertility. a**, Schematic overview of experimental design. Ovaries were collected from female mice at three time points (9-, 12- and 50-week-old mice; three animals per time point analysed). Samples were processed for DIA-MS. **b**, Cluster analysis of protein abundances during reproductive decline. Normalized DIA-MS data for 9-, 12- and 50-week-old ovaries were subject to clustering to minimize variance within clusters. Dendrogram of resulting clusters is shown on the left. Colours indicate DIA intensities centred to a mean of 0 and s.d. of 1 on per protein basis. Resulting protein abundance clusters are highlighted with different colour keys. (**c-e**) Pathway over-representation analysis of protein abundance cluster 2

(**c**), cluster 4 (**d**) and cluster 5 (**e**). **f**, Heatmaps showing protein abundance change for the high protein longevity cluster with assignment to protein abundance clusters. **g**, Pathway over-representation analysis of high protein longevity cluster proteins in protein abundance clusters 2 and 4 in the ovary. **c–e** and **g** show the proportions of genes detected in the gene set with their respective adjusted P values for the most prominent pathways. The P values are based on the hypergeometric test and have been adjusted for multiple hypothesis testing with Benjamini–Hochberg procedure. For the complete list of enriched pathways and their corresponding exact P values, see Supplementary Table 9.

in oocytes[15–17], and that its age-related decline is a driver of the increase in oocyte aneuploidy with advancing female age[52,53]. Our data support the hypothesis that other cohesin subunits and regulatory proteins are also very long-lived and show an age-related decline, including SMC3,

SMC1A and PDS5B. Many other proteins associated with chromatin organization are also long-lived, including HDAC1, SAP30, HIRA, RIF and DNMT3A, and many of them decrease in abundance in the aging ovary. We also observed a decrease in long-lived proteins required for

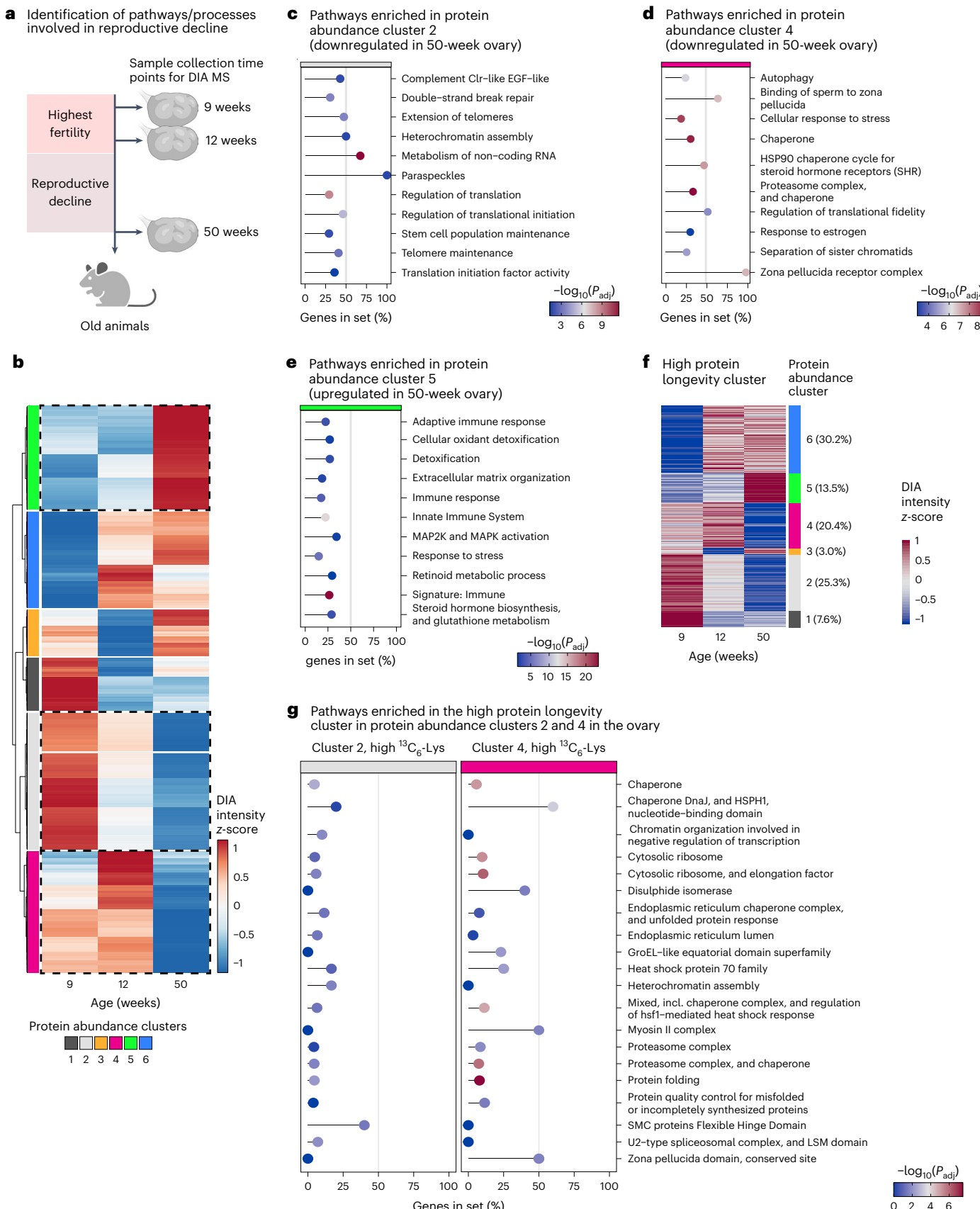

**a** Identification of pathways/processes involved in reproductive decline

**b** Protein abundance clusters
1 2 3 4 5 6

**c** Pathways enriched in protein abundance cluster 2 (downregulated in 50-week ovary)

**d** Pathways enriched in protein abundance cluster 4 (downregulated in 50-week ovary)

**e** Pathways enriched in protein abundance cluster 5 (upregulated in 50-week ovary)

**f** High protein longevity cluster / Protein abundance cluster

**g** Pathways enriched in the high protein longevity cluster in protein abundance clusters 2 and 4 in the ovary

maintaining protein homeostasis in aged ovaries. Thus, although the mechanism of increasing protein longevity and enhancing proteostasis improves some of the senescent characteristics in oocytes, it cannot fully prevent female germline senescence.

In addition, our analysis suggests candidate proteins and pathways for age-related changes in the ovary that are not yet understood at the molecular level. For example, we identified many apoptosis-promoting proteins that are upregulated in the aging ovary and may be involved in the age-related decline in follicle number. We also identified prominent age-related changes in the abundance of proteins involved in DNA damage repair, protection against oxidative damage, chromatin organization, epigenetic regulation, mitochondria and telomeres, which may contribute to the previously reported age-related changes in oocytes and ovaries[45–49,78–87].

Finally, our data show that not only oocytes, but also subsets of ovarian somatic cells are highly enriched in long-lived proteins. These include granulosa cells of early-stage follicles, as well as subsets of thecal and stromal cells. The functions of stromal cells are only poorly understood[88], and will be an exciting topic for future studies. Taken together, these findings are consistent with a model in which ovarian aging is driven not only by oocyte aging, but also by the aging of somatic cells in this organ[89–92].

## Online content

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

¹Department of Meiosis, Max Planck Institute for Multidisciplinary Sciences, Göttingen, Germany. ²Bioanalytical Mass Spectrometry Group, Max Planck Institute for Multidisciplinary Sciences, Göttingen, Germany. ³Bioanalytics Group, Department of Clinical Chemistry, University Medical Center Göttingen, Göttingen, Germany. ⁴Quantitative and Systems Biology Group, Max Planck Institute for Multidisciplinary Sciences, Göttingen, Germany. ⁵Cell Biology Division, MRC Laboratory of Molecular Biology, Cambridge, UK. ⁶Department for Neuro and Sensory Physiology, University Medical Center Göttingen, Göttingen, Germany. ⁷Center for Biostructural Imaging of Neurodegeneration, Göttingen, Germany. ⁸Cluster of Excellence Multiscale Bioimaging: from Molecular Machines to Networks of Excitable Cells, University of Göttingen, Göttingen, Germany. ⁹Present address: Department of Physiology, Development and Neuroscience, University of Cambridge, Cambridge, UK. ¹⁰Present address: Laboratory of Embryology, Institute of Advanced Medical Sciences, Tokushima University, Tokushima, Japan. ¹¹Present address: East Anglian Medical Genetics Service, Cambridge University Hospitals, NHS Foundation Trust, Cambridge, UK. ¹²Present address: Translation Alliance Lower Saxony, Hannover, Braunschweig, Göttingen, Germany. ¹³These authors contributed equally: Katarina Harasimov, Rebecca L. Gorry, Luisa M. Welp, Sarah Mae Penir, Yehor Horokhovskyi. ✉e-mail: henning.urlaub@mpinat.mpg.de; juliane.liepe@mpinat.mpg.de; melina.schuh@mpinat.mpg.de

## Methods

### Ethics

The maintenance and handling of all FVB/N and CD1 mice was performed in the MPI-NAT animal facility according to international animal welfare rules (Federation for Laboratory Animal Science Associations guidelines and recommendations). Requirements of formal control of the German national authorities and funding organizations were satisfied, and the study received approval by the Niedersächsisches Landesamt für Verbraucherschutz und Lebensmittelsicherheit (LAVES).

### Mouse oocyte and ovary collection and preparation

FVB/N mice fed with $^{13}C_6$-Lys feed were kept in static filter cages from Ehret. All other mice were kept in a Blue Line IVC system from Techniplast. All mice were kept in rooms with constant temperature of 21 °C and the humidity of 55%. The light/dark rhythm was 12:12 hours, from 05:00 to 17:00. Health monitoring was carried out in accordance with Federation of European Laboratory Animal Science Associations recommendations with large annual examinations in January and smaller scale in May and September. This study investigates protein turnover and abundance in ovaries and oocytes, and hence, only animals of female sex were analysed.

All $^{13}C_6$-Lys feed was purchased from Silantes. Fully $^{13}C_6$-labelled FVB/N female mice were created by successive matings and a diet consisting exclusively of $^{13}C_6$-Lys feed (Extended Data Fig. 1a). F0 females were fed $^{13}C_6$-Lys feed for 8 weeks, starting from the week 3 after birth, after which they were mated with wild-type FVB/N males who were kept on a standard diet. The F1 pups born from this mating and the F0 mother were further fed together with $^{13}C_6$-Lys-feed until weaning. Thereafter, only F1 offspring was further fed $^{13}C_6$-Lys feed. Upon reaching sexual maturity, F1 $^{13}C_6$-Lys-fed females were mated with wild-type FVB/N males who were kept on a standard diet. The pups of the F2 generation were the first generation of mice used for experiments. Females of F1 generation were mated continuously to produce experimental animals. Fully labelled $^{13}C_6$-Lys-breeders were periodically substituted by the pups from the fully labelled offspring. The percentage of labelling, as highlighted in Extended Data Fig. 1, was determined by MS analysis of blood samples from F0, F1 and F2 animals as described below (Extended Data Fig. 1b–e). We also analysed percentage of labelling in oocytes of F1 generation (Extended Data Fig. 1d), which was similar to the percentage of labelling observed from blood samples (Extended Data Fig. 1c). The F2 offspring had a 98.7% $^{13}C_6$-Lys labelled proteome (Extended Data Fig. 1e).

The offspring of fully labelled females were moved to a wet nurse $^{12}C_6$-Lys-fed CD1 female within 24 h after birth (Figs. 1a and 2a) or were kept with the $^{13}C_6$-Lys female breeders until weaning (3 weeks after birth) to provide a longer 'pulse' period before the 'chase' diet of $^{12}C_6$-Lys feed (Fig. 2b). All animals were fed ad libitum and had unrestricted access to water.

For MS analyses, oocytes were collected from 8-week-old mice (±24 h). Oocytes were prepared in the homemade M2 medium without bovine serum albumin (BSA, Fisher BioReagents; BP9700100), supplemented with 250 µM dbcAMP (Sigma; D0627) to maintain oocytes in the prophase arrest. Upon collection, oocyte samples were snap-frozen in liquid nitrogen in <5 µl of media and stored at −80 °C. All ovaries collected for MS analysis were prepared in M2 medium without BSA or in phosphate-buffered saline (PBS) prior to snap freezing and storing at −80 °C. Ovaries prepared from mice between the age of 6 and 30 weeks were dissected to remove medulla.

### Estimation of the mouse oocyte volume

The diameters of oocytes in primordial follicles were measured from archived paraffin-fixed ovarian slices imaged with the Zeiss LSM800. The diameters of germinal vesicle (GV) oocytes were measured from archived fixed and live confocal images of GV oocytes imaged with the Zeiss LSM800. The volume of the oocytes was then estimated using the formula $v = 4/3\pi \times r^3$, where $r$ is radius.

### Estimation of mouse ovarian volume

Ovaries from FVB/N mice were collected at 24 h, 48 h, 1, 2, 3, 6, 9, 12, 30, 50 and 65 weeks after birth. Excess fat and tissue were removed from the organs, before they were placed into a 35 mm imaging dish and measured on a Zeiss LSM800 confocal microscope. A shelf consisting of three glass cover slip pieces was created in order to hold the ovaries in place without impacting the dimensions of the ovaries themselves; the glass cover slip pieces were affixed to the imaging dish using double-sided tape, and placed in a U-shape. Each ovary was placed within the U-shaped shelf and submerged in a drop of homemade M2 medium, before its dimensions were measured using the $x$, $y$ and $z$ coordinates of the ovary edge as observed using the transmission light on a Zeiss LSM800 microscope. The volume of the ovaries was then estimated by using the formula $v = lwh$ (length × width × height).

### Preparation of mouse blood samples for MS measurements

Blood (20 µl), taken from orbital plexus from F0, F1 or F2 $^{13}C_6$-Lys-fed FVB/N males, was supplied with 80 µl urea lysis buffer (6 M urea, 10 mM HEPES-NaOH pH 8.0) following incubation on ice for 20 min and sonication for 10 min with 30 sec on/off cycles at 4 °C using the highest output level (Bioruptor, Diagenode). Lysate was cleared at 13,000g, 10 min, 4 °C. Protein concentrations were determined using Bradford assay (Bio-Rad) and further sample processing was performed on 20 µg protein amount.

### Preparation of mouse oocyte samples for MS measurements

For DDA MS measurements, two sets of oocytes ($n = 2,475$ and $n = 2,473$) were obtained from 8-week-old mice ($^{12}C_6$-Lys chow after birth until they were 8-weeks old) born from $^{13}C_6$-Lys-labelled females. The oocytes were suspended in 80 µl SDS lysis buffer (4% [w/v] SDS, 150 mM NaCl, 50 mM HEPES-NaOH pH 7.5, 2 mM DTT, 0.5% [v/v] NP40, 1X Roche complete protease inhibitors-EDTA). Oocytes were lysed for 10 min at 99 °C followed by sonication for 10 min with 30 sec on/off cycles at 20 °C using the highest output level (Bioruptor, Diagenode). Samples were further processed as specified below.

### Preparation of mouse ovary samples for MS measurements

For DDA MS measurements, mouse ovaries were collected from female pups pulsed with $^{13}C_6$-Lys until birth at 24 h, 48 h, 1, 2, 3, 6, 9, 12, 30, 50 and 65 weeks after birth, and for the mice that were pulsed with $^{13}C_6$-Lys until weaning at 6, 9, 12, 30, 50 and 65 weeks after birth. For protein abundance DIA-MS measurements, mouse ovaries were collected from wild-type FVB/N females at 24 h, 1, 2, 3, 5, 9, 12 and 50 weeks after birth. For all experiments and all time points, three biological replicates were collected, each containing two ovaries from a single animal. Small ovaries (24 h and 48 h time points) were suspended in 40 µl SDS lysis buffer and lysed for 10 min at 99 °C and by sonication for 10 min with 30 sec on/off cycles at 20 °C using the highest output level (Bioruptor, Diagenode). Larger ovaries (1 week onwards) were suspended in 100 µl SDS lysis buffer and ten Zirconia/Silica beads (2.3 mm; BioSpec Products; 11079125Z) were added to each sample. Lysis was performed using a FastPrep-24 benchtop homogenizer (MP Biomedicals) for three 20 sec on/off cycles at 5.5 m s$^{-1}$ following incubation for 10 min at 99 °C and lysate clearance at 17,000 ×g, 5 min. Protein concentrations were measured for samples derived from 1-week-old and older mice using Bradford (Bio-Rad) or Pierce BCA Protein Assay Kit (ThermoFisher Scientific) according to manufacturer's instructions and further sample processing was performed starting with 50 or 300 µg protein amount for DIA or DDA ovary analyses, respectively.

### Protein digestion and sample cleanup

F0 blood samples were processed by in-gel digestion according to ref. 93. Samples were loaded onto a 4–12% NuPAGE Novex Bis-Tris Mini-gel (Invitrogen), following Coomassie staining. The protein containing lane was cut into 23 pieces. Proteins were reduced with dithiothreitol

(DTT), alkylated with iodoacetamide (IAA) and digested overnight with trypsin (Serva; T6567). Tryptic peptides were extracted from gel pieces, dried in a vacuum concentrator and resuspended in MS buffer for LC-MS/MS measurements. F1 and F2 blood samples were processed in solution. Samples were adjusted to 20 µl using urea lysis buffer followed by reduction of proteins using 5 mM DTT and incubation for 30 min, 25 °C, 650 r.p.m., and alkylation with 20 mM IAA for 30 min, 25 °C, 650 r.p.m., in the dark. Samples were diluted to a final urea concentration of 0.5 M using 50 mM NH₄HCO₃. Protein digest was performed using trypsin (Promega; V5111) at a 1:20 enzyme-to-protein mass ratio and incubation overnight at 37 °C, 650 r.p.m. Desalting of blood samples was performed as described below.

Oocyte and ovary lysates were diluted to 1% (w/v) final SDS concentration using 100 mM NH₄HCO₃ (for oocytes) or 50 mM HEPES-NaOH pH 7.5 (for ovaries) and incubated with 250 U (DIA oocyte samples and DDA/DIA ovary samples) or 500 U (DDA oocyte samples) Pierce Universal Nuclease (ThermoFisher Scientific; 88700) and 1 mM MgCl₂ for 30 min, 37 °C, 300 r.p.m. Proteins were reduced with 5 mM DTT for 30 min, 37 °C, 300 r.p.m.; alkylated with 10 mM iodoacetamide for 30 min, 25 °C, 300 r.p.m., in the dark; and quenched with 5 mM DTT for 5 min, 25 °C, 300 r.p.m. Proteins were further purified according to the bead-based SP3 preparation method[94,95] to remove detergents. Briefly, carboxylate modified magnetic beads (Cytiva; 65152105050350, 45152105050250) were added at a 1:10 protein-to-bead mass ratio and acetonitrile (ACN) was added to 50% (v/v) to induce protein binding to the beads. Washing was performed three to five times with 80% EtOH and once with 100% ACN. For protein digestion, the beads were suspended in 100 mM NH₄HCO₃ with trypsin and rLys-C (Promega; V5111, V1671) at a 1:20 ratio assuming a protein amount of 25 ng per oocyte and 50 µg total protein amount for 24 h and 48 h old ovaries. The digestion-bead mix was incubated for 16 h at 37 °C, 1,000 r.p.m. and the digested peptides were collected according to the SP3 protocol. The beads were rinsed once with 50 µl 100 mM NH₄HCO₃ and sonicated for 30 seconds. The supernatant was pooled with the collected peptide mix. From DDA oocyte and ovary samples, unfractionated input samples (3%, 'input') were removed before desalting and offline fractionation. Desalting of DIA ovary samples was performed as described below and DIA oocyte samples were dried in a vacuum concentrator and resuspended in MS buffer (2% (v/v) ACN, 0.05% (v/v) trifluoroacetic acid (TFA)) and subjected to LC-MS/MS measurements.

Blood samples, DDA oocyte and ovary samples and DIA ovary samples were desalted using C₁₈ Micro Spin Columns (Harvard Apparatus; 74-4601) according to manufacturer's instructions. Briefly, samples were adjusted to 0.1% (v/v) formic acid (FA) and loaded onto equilibrated spin columns. Samples were reloaded once and washed three times with 0.1% (v/v) FA. Peptides were eluted using 50% (v/v) ACN, 0.1% (v/v) FA and 80% (v/v) ACN, 0.1% (v/v) FA. Eluants were dried in a vacuum concentrator. Blood samples and DIA ovary samples were resuspended in MS buffer for LC-MS/MS measurements. DDA oocyte and ovary samples were offline fractionated as described below.

### Basic reversed phase sample fractionation
Cleaned-up peptides from DDA samples were dissolved in 35 µl 10 mM NH₄OH pH 10, 5% (v/v) ACN. Peptides were loaded onto an Xbridge C18 column (Waters; 186003128) using an Agilent 1100 series chromatography system. The column was operated at a flow rate of 60 µl min⁻¹ with a buffer system consisting of 10 mM NH₄OH pH 10 (buffer A) and 10 mM NH₄OH pH 10, 80% (v/v) ACN (buffer B). The column was equilibrated with 5% B and developed over 64 min using the following gradient: 5% B (0–7 min), 8–30% B (8–42 min), 30–50% B (43–50 min), 90–95% B (51–56 min), 5% B (57–64 min). The first 6 min were collected as one flow-through fraction, followed by 48 × 1 min fractions, which were reduced to 12 fractions by concatenated pooling. For DDA oocyte samples, the last 10 min of each run were collected as one 'rest' fraction. All fractions were dissolved in MS buffer for LC-MS/MS measurements.

### LC-MS/MS analysis
Blood samples were measured in duplicate on an Orbitrap Fusion Tribrid Mass Spectrometer (ThermoFisher Scientific), coupled to a Dionex Ultimate 3000 RSLCnano system. Analytes were loaded on a Pepmap 300 C₁₈ column (ThermoFisher Scientific) at a flow rate of 10 µl min⁻¹ in 0.1% (v/v) FA (buffer A) and washed for 3 min with buffer A. Samples were separated on an in-house packed C₁₈ column (30 cm; ReproSil-Pur 120 Å, 1.9 µm, C18-AQ; inner diameter, 75 µm) at a flow rate of 300 nl min⁻¹. Sample separation was performed using a buffer system consisting of buffer A and 80% (v/v) ACN, 0.08% (v/v) FA (buffer B). The main column was equilibrated with 5% B, sample was injected and column was washed for 3 min with 5% B. A linear gradient from 10–42% B over 163 min was applied, to separate peptides, followed by 5 min at 90% B and 8 min at 5% B. Peptides were analysed in positive mode using a data-dependent top speed acquisition method with a cycle time of 3 sec. MS1 scans were acquired in an Orbitrap mass analyser with a resolution set to 120,000 FWHM; AGC target was set to standard. MS2 scans were acquired in an Ion Trap with scan rate set to rapid; AGC target was set to $3 \times 10^3$ (30%). Precursors selected during MS1 scans (scan range $m/z$ 350–1,500) were fragmented using 34% normalized, higher-energy collision-induced dissociation (HCD) fragmentation. Further MS/MS parameters were set as follows: isolation width, 1.6 $m/z$; dynamic exclusion, 30 sec; maximum injection times (MS1/MS2), 50 ms/dynamic.

Each DDA oocyte fraction was measured in triplicates on a Q Exactive HF-X Hybrid Quadrupole-Orbitrap Mass Spectrometer (ThermoFisher Scientific), DDA ovary and DIA oocyte and ovary samples were measured on an Exploris 480 Mass Spectrometer (ThermoFisher Scientific). Mass spectrometers were coupled to Dionex Ultimate 3000 RSLCnano systems. Pre- and main column setup, flow rates and precolumn equilibration and loading were the same as for blood samples. For DDA oocyte and ovary fractions, a linear gradient from 10–42% B over 103 min was applied for separation of peptides on main column. For DDA oocyte input and rest fractions, peptide separation was performed over 163 and 43 min (10–42% B), respectively. DIA oocyte and ovary samples were separated using a linear gradient from 5–14% in 97 min, followed by 14–32% in 100 min and 32–48% in 27 min and a total of 238 min runtime. For all runs, peptide separation was followed by 5 min at 90% B and 8 min at 5% B. For DDA oocyte and ovary analyses, eluting peptides were analysed in positive mode using a data-dependent top 30 acquisition method. MS1 and MS2 resolution were set to 60,000 and 15,000 FWHM, respectively, and AGC targets were $10^6$ and $10^5$. Further MS/MS parameters were set as follows: scan range, $m/z$ 350–1,600; HCD collision energy, 30%; isolation width, 1.4 $m/z$; dynamic exclusion, 20 sec; maximum injection times (MS1/MS2), 50 ms/54 ms. For DIA analyses, data were acquired using positive mode. MS1 scans were performed using the following settings: resolution, 12,000 FWHM; AGC target, $3 \times 10^6$; scan range, 350–1,600 m/z; maximum injection time, 20 ms. Following each MS1 scan, MS2 scans were acquired using tMS² option in Thermo Xcalibur Instrument Setup software, in 70 defined, variable $m/z$ windows (Supplementary Table 11). Further MS2 parameters were set as follows: resolution, 30,000 FWHM; AGC target, $10^6$; maximum injection time, 55 ms; HCD collision energy, 30%. For all DDA and DIA measurements, the lock mass option ($m/z$ 445.120025) was used for internal calibration.

### Peptide database search
Database searches for DDA data were performed using MaxQuant software (version 1.5.2.8 for blood samples and 1.6.0.1 for oocyte and ovary samples)[96,97]. For blood samples, a reviewed (Swiss-Prot) *Mus musculus* (strain C57BL/6J) reference proteome database, including canonical protein sequences, was downloaded from UniProt Knowledgebase (date of download: 6 January 2017; 10,090 proteins); for oocyte and ovary samples, a similar database was used that included canonical and isoform protein sequences (date of download: 10 September 2021;

17,077 proteins). For long-lived protein analysis, data were searched with the following settings: enzyme, trypsin/P; multiplicity, 2; heavy labels, Lys6; fixed modifications, carbamidomethyl (C); variable modifications (included in protein quantification), oxidation (M), acetyl (protein N⁻ term) for blood sample data, and oxidation (M), acetyl (protein N⁻ term), deamidation (N), methyl (KR), for other data; match between runs, enabled. For mixed peptide analysis, settings were the same except for multiplicity was set to 1 and Lys6 ($^{13}$C6-K) was set as variable modification. MaxQuant results for F0, F1 and F2 $^{13}$C$_6$-Lys-fed FVB/N male blood samples were further processed using R studio[98] to determine median peptide $^{13}$C$_6$-Lys incorporation rates for lysine-containing peptides showing $H/L$ ratios. DIA data were analysed with Spectronaut version 15.7.220308.50606 (ref. [99]). A spectral library was generated from all DDA oocyte and ovary data, which were analysed using Pulsar search engine platform and the following settings: enzyme, trypsin/P; minimum peptide length, 7; maximum peptide length, 52; fixed modifications, carbamidomethyl (C); variable modifications oxidation (M), acetyl (protein N-term). BGS Factory Settings were used for identification and quantification of proteins from DIA data except for maximum. Top N precursors used for quantity calculation was set to 5. A minimum of two unique peptides per isotope was required for protein quantification.

### Single-cell RNA sequencing

**Collection and dissociation of postnatal ovaries.** The postnatal ovary samples were collected in 1X PBS. For the generation of each single-cell sequencing library, 6 ovaries were collected from 3 different pups from a single pregnant female mouse. The ovaries were dissociated into single cells using 1 mg ml$^{-1}$ Collagenase Type IV (Gibco; 17104019) at 37 °C for 30 min with pipetting at regular intervals, followed by incubation with Accumax (PAN-Biotech; P10-21200) at 25 °C for 5 min with continuous shaking at 500 r.p.m. The sample was triturated for 1 min using wide orifice low retention tips (Mettler Toledo). The reaction was stopped with 0.04% fetal bovine serum, and the cell suspension was passed through 35 µm (Corning; 352235) and 40 µm (Merck; 136800040) cell strainers respectively. Finally, the sample was centrifuged at 400$g$ for 5 min at 4 °C, the supernatant was aspirated and the pellet was resuspended in 1X PBS with 0.04% BSA.

**Generation of single-cell RNA libraries and sequencing.** The ovarian single-cell suspensions were loaded onto the 10X Genomics Chromium Single Cell system using Chromium Single Cell 3′ Reagent Kits v3 as per manufacturer's instructions. Each reaction well was loaded with approximately 12,000 cells to achieve a recovery estimate of about 7,000 cells per library. Single cells were then partitioned into Gel Bead-In Emulsions (GEMs) in the Chromium controller followed by generation and amplification of cDNA molecules with unique 10X barcodes. The single-cell RNA-seq libraries were subjected to pair-end sequencing on Illumina HiSeq 4000 (Sequencing Core Facility, MPI-MG, Berlin, Germany).

### Protein turnover analysis in oocytes and ovaries, and scRNAseq analysis of mouse ovaries

All related information about the oocyte and ovarian proteome analysis, and the single-cell RNA sequencing of mouse ovaries are described in detail in the Supplementary Note 1.

### Aggresome staining

Aggresomes were stained using the ProteoStat Aggresome Detection Kit (Enzo Life Sciences; ENZ-51035-K100) according to the manufacturer's instructions with some modifications. Early-stage follicles or full-grown oocytes were isolated from 9-week-old or 65-week-old mice and fixed with 2% methanol-free formaldehyde in 1×Assay Buffer for 1 h at room temperature. Fixed oocytes/follicles were washed and permeabilized with 0.5% Triton X-100 in 1×Assay Buffer containing

3 mM EDTA (pH 8.0) with shaking on ice for 1 h. After a brief wash with 1×Assay Buffer containing 0.1% BSA, oocytes/follicles were incubated in 1×Assay Buffer containing 0.1% BSA, 1:2,000 diluted ProteoStat aggresome dye, and 20 µg ml$^{-1}$ Hoechst 33342 for 1 h with shaking at room temperature. The samples were then extensively washed with 1×Assay Buffer containing 0.1% BSA and scanned using a Zeiss LSM880. The ProteoStat aggresome dye was excited with a 561 nm laser line and detected at 579 to 623 nm. Brains isolated from three 9-week-old and three 65-week-old mice were fixed with 4% paraformaldehyde in PBS for at least 5 h on ice and then washed thoroughly with PBS. The fixed samples were placed in 25% sucrose in PBS and gently shaken overnight in a cold room. They were then embedded in OCT Compound (Tissue-Tek; 4583) and sectioned at 14 µm. The brain slices were washed briefly with PBS and treated with the same permeabilization solution as described above for 30 min at room temperature. They were then washed again with PBS and stained with 1:2,000 diluted ProteoStat aggresome dye in 1×Assay Buffer for 30 min at room temperature. After extensive washing with PBS, Vectashield Antifade Mounting Medium with DAPI (VECTOR; H-1200) was added, and the slides were sealed for imaging. Fifteen areas of the cerebral cortex were randomly selected and scanned in each brain.

### Proteasome activity assay

To construct the proteasome activity reporter pGEMHE-Ub(G76V)-mClover3-T2A-mScarlet, Ub(G76V) was amplified from Ub(G76V)-EGFP (a gift from Nico Dantuma, Addgene plasmid 11941)[32] and assembled into pGEMHE together with three mClover3 sequences and T2A-mScarlet. pGEMHE-Ub(G76V)-mClover3-T2A-mScarlet was linearized with PacI and then transcribed in vitro using the HiScribe T7 ARCA mRNA Kit (NEB; E2065S). The transcribed mRNA was further purified using the RNeasy Mini Kit (Qiagen; 74104). To test whether Ub(G76V)-mClover3 is efficiently degraded in oocytes, 4 pl of 0.3 µM *Ub(G76V)-mClover3-T2A-mScarlet* mRNA was injected into oocytes collected from either 9-week-old or 65-week-old mice, and then imaged in the presence or absence of 10 µM MG-132. After injection, two proteins, Ub(G76V)-mClover3 and mScarlet, were generated because of the presence of the self-cleaving T2A peptide. Ub(G76V)-mClover3 is a proteasome substrate, while mScarlet serves as a control. To follow the degradation kinetics of Ub(G76V)-mClover3, 4 pl of 0.3 µM Ub(G76V)-mClover3-T2A-mScarlet mRNA was injected into oocytes collected from either 9-week-old or 65-week-old mice and expressed for 5 h in the presence of 10 µM MG-132. MG-132 can prevent the degradation of Ub(G76V)-mClover3, allowing it to accumulate in oocytes. MG-132 was then washed out and the oocytes were imaged in medium containing 100 µg ml$^{-1}$ of the translation inhibitor cycloheximide and using a Zeiss LSM880.

### Secondary ion mass spectrometry

**Sample preparation and embedding for SIMS imaging.** Ovaries were collected at 4 and 8 weeks from mice kept with the fully $^{13}$C$_6$-Lys-labelled FVB/N mothers until weaning (3 weeks after birth) and cut carefully into three or four slices (approximately 100–200 µm in thickness, depending on size of ovary), on PBS-soaked filter paper under a stereoscope. Slices were transferred to 2% (w/v) EM-grade glutaraldehyde for fixation (RT, 2 h). Slices were washed in PBS three times, before quenching in 100 mM ammonium chloride for 30 min at RT. Slices were washed in PBS three more times before embedding in LR White resin (medium grade, London Resin Company), according to published protocols[100]. In brief, the samples were dehydrated using a series of ethanol dilutions (30%, 50%, 70% in ddH$_2$O), and were then incubated in a 1:1 mixture of LR White and 70% ethanol, before being placed in pure LR White resin, and being finally placed in resin containing LR White accelerator (London Resin Company). A final polymerization was performed at 60 °C. The samples were then processed to 200 nm thin sections, employing an ultramicrotome (type EM UC6, Leica Microsystems), and the sections

were mounted on silicon wafers (Siegert Wafer GmbH). Consecutive sections were mounted on glass slides and were stained with a dilution of toluidine blue (1% in $H_2O$ containing 2% sodium borate), before histological imaging.

**SIMS imaging and image processing.** Imaging was performed using a NanoSIMS 50L instrument (Cameca, France). The negative ion mode was employed, using a 8 kV $Cs^+$ primary ion source. The selected imaging area was pre-implanted before imaging for several minutes, with an ion current of approximately 600 pA, to achieve a proper steady-state of ionization. The entrance and aperture slits were optimized for separating isobaric mass peaks optimally, concentrating on imaging $^{12}C^{14}N^-$ and $^{13}C^{14}N^-$. Imaging was performed using a primary ion current of 2.5 pA, with an empirically adjusted dwell time. The images of individual samples were taken simultaneously, with identical dwell time, to enable precise isotope ratio calculations. Histology images of the samples were acquired using an Axio Imager M2 upright microscope (Zeiss). These images were overlapped onto the SIMS images, using Adobe Photoshop (2020). $^{13}C/^{12}C$ ratios were then determined in regions of interest (ROIs) identified by an experienced user in the histology images, using a self-written routine in MATLAB (the Mathworks) as described previously[100].

### Statistics and reproducibility

**Sample size.** In the single-cell RNA sequencing and proteomics datasets, no groups of conditions were compared to each other, and therefore, no sample-size calculation was performed.

For the oocyte protein aggregation and proteosomal activity experiments, no statistical methods were used to predetermine sample size. Retrospectively, achieved sample sizes were determined to be adequate based on the magnitude and consistency of measurable differences between groups. Most importantly, sample size per experiment was dictated by the number of oocytes that could be processed for microinjection and live imaging within a reasonable time by the researcher without affecting oocyte quality.

**Data exclusion.** In the oocyte protein turnover experiments (DDA MS), median over technical replicates were only computed if the DDA intensity of $^{13}C_6$-Lys labelled protein was detected in at least two out of four technical replicates, otherwise the data point is omitted (set as n.a.). The mean and s.d. of $F$ (fractions of $^{13}C_6$-Lys labelled proteins) were calculated if $F$ was detected in at least one of the two biological replicates.

A protein was defined as 'heavy only' if it was detected as $^{13}C_6$-Lys labelled protein in at least two out of four technical replicates in at least one biological replicate and there is no signal for $^{12}C_6$-Lys labelled protein in any of the four technical replicates of both biological replicates detected. Accordingly, a protein was defined as 'light only' if it was detected as $^{12}C_6$-Lys labelled protein in at least two out of four technical replicates in at least one biological replicate and if there is no $^{13}C_6$-Lys labelled protein in any of the four technical replicates of both biological replicates detected.

No data points were excluded in the oocyte protein aggregation and proteosomal activity experiments.

**Replication.** In the oocyte protein turnover experiments (DDA MS), two sets of oocytes ($n = 2,475$, $n = 2,473$) were obtained from a total of 8-week old mice ($^{12}C_6$-Lys chow after birth until they were 8-weeks old) born from $^{13}C_6$-Lys labelled females. Oocytes from multiple animals were pooled for a single replicate to reach the minimum input required for the MS processing. Each DDA oocyte fraction was measured in quadruplicate.

In the ovary protein turnover experiments (DDA MS) with $^{13}C_6$-Lys pulse until birth and $^{12}C_6$-Lys chase from birth, ovaries were collected from the female progeny at 11 time points (24 hours, 48 hours, 1, 2, 3, 6, 9, 12, 30, 50 and 65 weeks). For each time point, three biological replicates were collected, each containing two ovaries from a single animal.

In the ovary protein turnover experiments (DDA MS) with $^{13}C_6$-Lys pulse until birth and $^{12}C_6$-Lys chase from weaning, ovaries were collected from the female progeny at six time points (6, 9, 12, 30, 50 and 65 weeks). For each time point, three biological replicates were collected, each containing two ovaries from a single animal.

In the protein abundance experiments (DIA-MS), ovaries were collected from unlabelled FVB/N female mice at eight time points (24 hours, 1, 2, 3, 5, 9, 12, and 50 weeks). For each time point, three biological replicates were collected, each containing two ovaries from a single animal.

In the single-cell RNA sequencing experiments, each library was generated from six ovaries collected from three different pups in a single pregnant CD1 female mouse.

All data in the oocyte protein aggregation and proteosomal activity experiments are from at least two independent experiments or multiple biological replicates. All attempts at replication were successful.

**Randomization.** The single-cell RNA sequencing and proteomics datasets did not have different treatments or conditions that could be randomized. For the oocyte protein aggregation and proteosomal activity experiments, mouse oocytes were collected from multiple ovaries or animals and then pooled. They were then randomly assigned to different experimental groups.

**Blinding.** The single-cell RNA sequencing and proteomics datasets did not have different groups of conditions, and therefore, these experiments did not require blinding.

For the oocyte protein aggregation and proteosomal activity experiments, the investigators were not blinded to allocation during experiments and outcome assessment, as each experiment was performed by one researcher alone. Thus, blinding during group allocation was not possible to ensure samples received the right treatment or manipulation during experiment. Blinding was also not possible during data analysis as it was performed by the same researcher that conducted the experiment.

### Reporting summary

Further information on research design is available in the Nature Portfolio Reporting Summary linked to this article.

## Data availability

The scRNAseq datasets are deposited in the NCBI Gene Expression Omnibus database[101] under accession number GSE237012. The proteomics data are deposited to the ProteomeXchange Consortium via the UCSD MassIVE repository with the dataset identifier MSV000092528 and PXD044113, for MassIVE and ProteomeXchange, respectively[102]. Source data are provided with this paper. All other data supporting the findings of this study are available from the corresponding author on reasonable request.

## Code availability

R scripts for data processing and visualization of proteomics and single-cell RNA sequencing data are available on figshare (https://doi.org/10.6084/m9.figshare.25604232.v1).

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

## Acknowledgements

We thank the staff from the Animal Facility and Proteomics Facility at the Max Planck Institute for Multidisciplinary Sciences for technical assistance and support; M. Daniel, S. Schlott and L. Wartosch for their assistance with maintenance and handling of fully labelled $^{13}C_6$-Lys mice; S. Schlott for preparing mouse brain and ovary slices, and oocyte isolation; M. Eggert Martínez for help in establishing the proteasome activity assay; Life Science Editors for critical comments on the manuscript; and all the members of Urlaub, Liepe and Schuh labs for helpful discussions. The research leading to these results received financial support from the Max Planck Society, a Deutsche Forschungsgemeinschaft (DFG) Leibniz Prize to M.S. (SCHU 3047/1-1), a European Research Council (ERC) Starting Grant (ERC-StG 945528 IMAP) to J.L. and the DFG SFB1565 (project number 469281184) to H.U.; S.M.P. and Y.H. were supported by the International Max Planck Research School for Genome Science, part of the Göttingen Graduate Center for Neurosciences, Biophysics and Molecular Biosciences. Some graphics in Figs. 1a,c, 2a,b, 4b and 8a, and Extended Data Figs. 2a and 6a were created with Biorender.com.

## Author contributions

M.S. conceived the study. K.H., L.M.W., A.L.T.-T., H.U., J.L. and M.S. designed the screen to identify long-lived proteins in mouse oocytes and ovaries. R.L.G., K.H., L.M.W., Y.H., H.U., J.L. and M.S. designed experiments and analyses to determine relative protein abundances throughout the lifetime of a mouse. K.H. and V.S. created fully labelled $^{13}C_6$-Lys mice. K.H. performed and collected all samples for oocyte and ovary MS analyses with help from A.-S.F.; K.H., L.M.W., A.S., A.-S.F. and H.U. optimized protocols for processing mouse oocytes and ovaries for MS. A.S. processed oocyte samples for MS analysis. L.M.W. and M.R. processed ovary samples for MS analysis. L.M.W. performed the initial processing of all MS data. J.L. developed methods to identify long-lived proteins in mouse oocytes with input from L.M.W., H.U., R.L.G. and M.S.; J.L. conducted all modelling related to the determination of ovarian protein turnover and changes in ovarian protein abundance, with input from Y.H., R.L.G., K.H., L.M.W., S.M.P., H.U. and M.S.; Y.H. and J.L. performed over-representation analysis, gene set enrichment analysis and clustering analysis with input from S.M.P., R.L.G., K.H., L.M.W., H.U. and M.S.; R.L.G., K.H., K.G., S.O.R. and M.S. designed NanoSIMS experiments. R.L.G. and K.G. processed samples for NanoSIMS imaging. K.G. performed NanoSIMS experiments. R.L.G. analysed NanoSIMS data with input from S.O.R. and M.S.; S.H. and M.S. designed the single-cell RNA sequencing experiment. S.H. established protocols for sample preparation of mouse postnatal ovaries for single-cell RNA sequencing. S.H. and D.S. processed mouse postnatal ovaries for single-cell RNA sequencing. S.M.P. analysed the single-cell RNA sequencing data. S.C. and K.T. designed the aggresome staining and proteasomal activity measurement experiments. K.T. designed and constructed the proteasomal activity reporter and performed assay characterization. S.C. performed all aggresome staining and proteasomal activity experiments and analysed the data. M.S. prepared the first draft of the main text. Subsequent editing and modifications to the text were done by all authors. The Supplementary Information was written by K.H., R.L.G., S.M.P., D.S., L.M.W., Y.H., K.G., S.O.R. and J.L. with input from all authors. Figure legends were written by K.H., S.M.P., Y.H. and J.L. with input from all authors. K.H., S.M.P., R.L.G., J.L. and M.S. prepared the figures with input from all the authors. Funding for this study was secured by J.L., H.U. and M.S.; H.U., J.L. and M.S. supervised the study.

## Funding

## Competing interests

The authors declare no competing interests.

## Additional information

**Extended data** is available for this paper at https://doi.org/10.1038/s41556-024-01442-7.

**Correspondence and requests for materials** should be addressed to Henning Urlaub, Juliane Liepe or Melina Schuh.

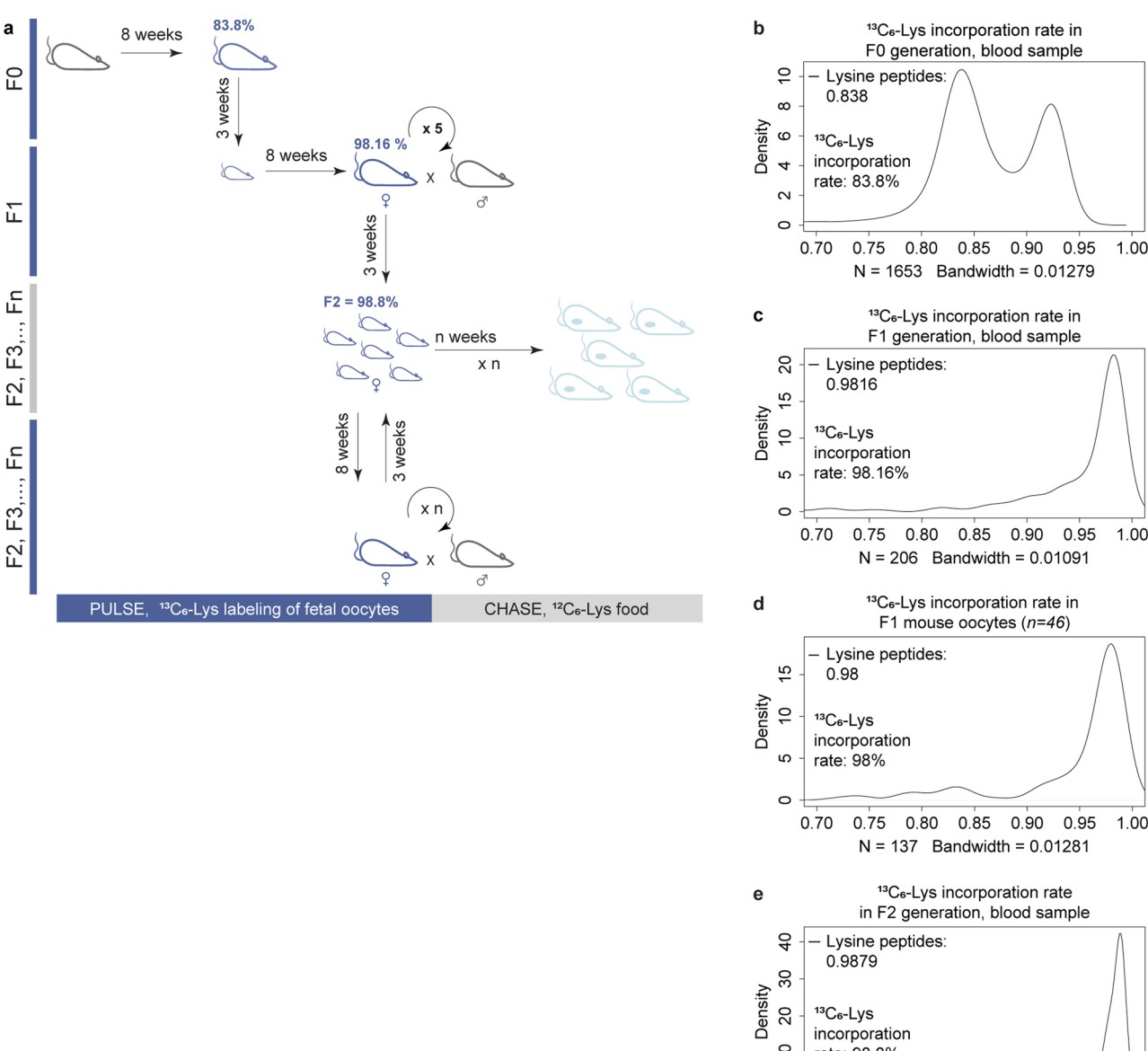

**Extended Data Fig. 1 | Generation of fully-$^{13}C_6$-Lys-labelled FVB/N female mice. (a)** Schematic overview of complete $^{13}C_6$-Lys labelling of mice. Fully-$^{13}C_6$-Lys-labelled FVB/N female mice were created by successive matings and a diet consisting exclusively of $^{13}C_6$-Lys feed. F0 females were fed with the $^{13}C_6$-Lys feed for 8 weeks, after which they were mated with wild-type FVB/N males who were kept on a standard diet. The F0 mother was further fed with the $^{13}C_6$-Lys feed and the F1 pups born from this mating were fed with the $^{13}C_6$-Lys feed until weaning. Thereafter, only the F1 offspring was further fed with the $^{13}C_6$-Lys feed. Upon reaching sexual maturity, the F1 $^{13}C_6$-Lys-fed females were mated with wild-type FVB/N males who were kept on a standard diet. The pups of the F2 generation were the first generation of mice used for experiments. Females of the F1 generation were continuously mated to produce experimental animals. Fully labelled $^{13}C_6$-Lys breeders were periodically substituted by the pups from the fully labelled offspring, and therefore, the F2 labelling efficiency represents the minimal labelling efficiency, as all subsequent generations had a larger fraction of $^{13}C_6$-Lys due to longer $^{13}C_6$-Lys feeding time. The percentage of $^{13}C_6$-Lys incorporation in blood samples in each generation is indicated in the top left corner of female mouse pictograms. **(b-e)** Histograms show $^{13}C_6$-Lys incorporation rates in (b) F0 blood sample for 1,653 lysine-containing peptides, (c) in F1 blood sample for 206 lysine-containing peptides, (d) in F1 oocyte samples for 137 lysine-containing peptides, and (e) in F2 blood sample for 162 lysine-containing peptides. Median $^{13}C_6$-Lys incorporation rates are indicated. Heavy isotope ($^{13}C_6$) incorporation rates were calculated from available lysine-containing peptides showing heavy ($^{13}C_6$) to light ($^{12}C_6$) lysine ratios. Source numerical data are available in source data.

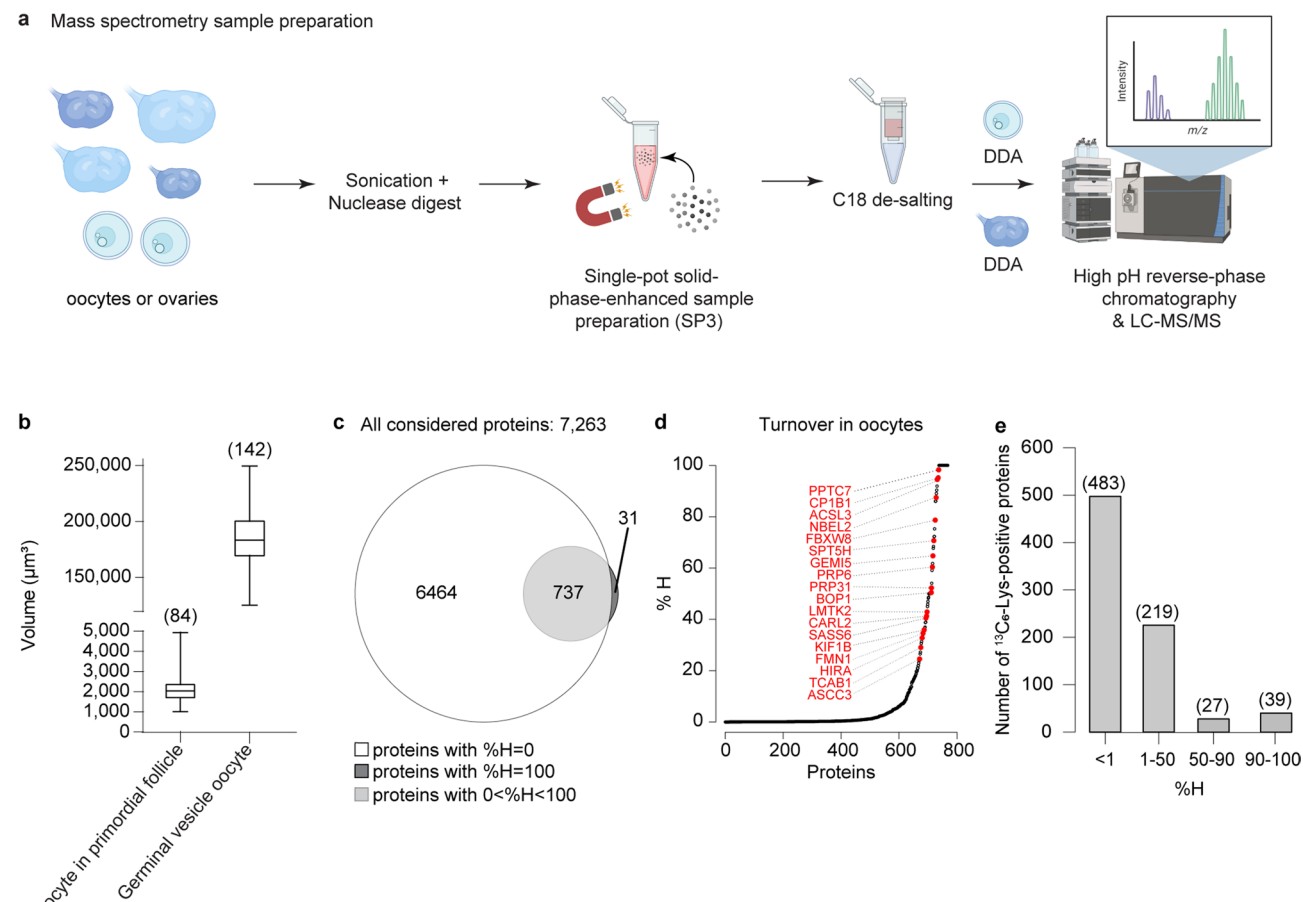

**Extended Data Fig. 2 | Sample preparation for MS DIA or DDA analysis and protein turnover in oocytes.** (**a**) Schematic representation of sample preparation for MS DIA or DDA analysis. (**b**) Box plot showing the volume (µm³) of oocytes in fixed primordial follicles and at germinal vesicle stage from live and fixed samples. Boxplots indicate median, 1st quartile, 3rd quartile, as well as minimum and maximum after outlier removal. Number of oocytes indicated in parenthesis. (**c**) Venn diagram of $^{12}C_6$-Lys containing proteins and $^{13}C_6$-Lys containing proteins detected in mouse oocytes. (**d**) Rank plot showing inferred remaining fraction of $^{13}C_6$-Lys labelled proteins (%H) in oocytes for each quantified protein (individual proteins denoted as black circles). Selected examples are highlighted in red. (**e**) Bar plot showing number of proteins with %H of 0–1, 1–50, 50–90, or 90–100%. Source numerical data are available in source data.

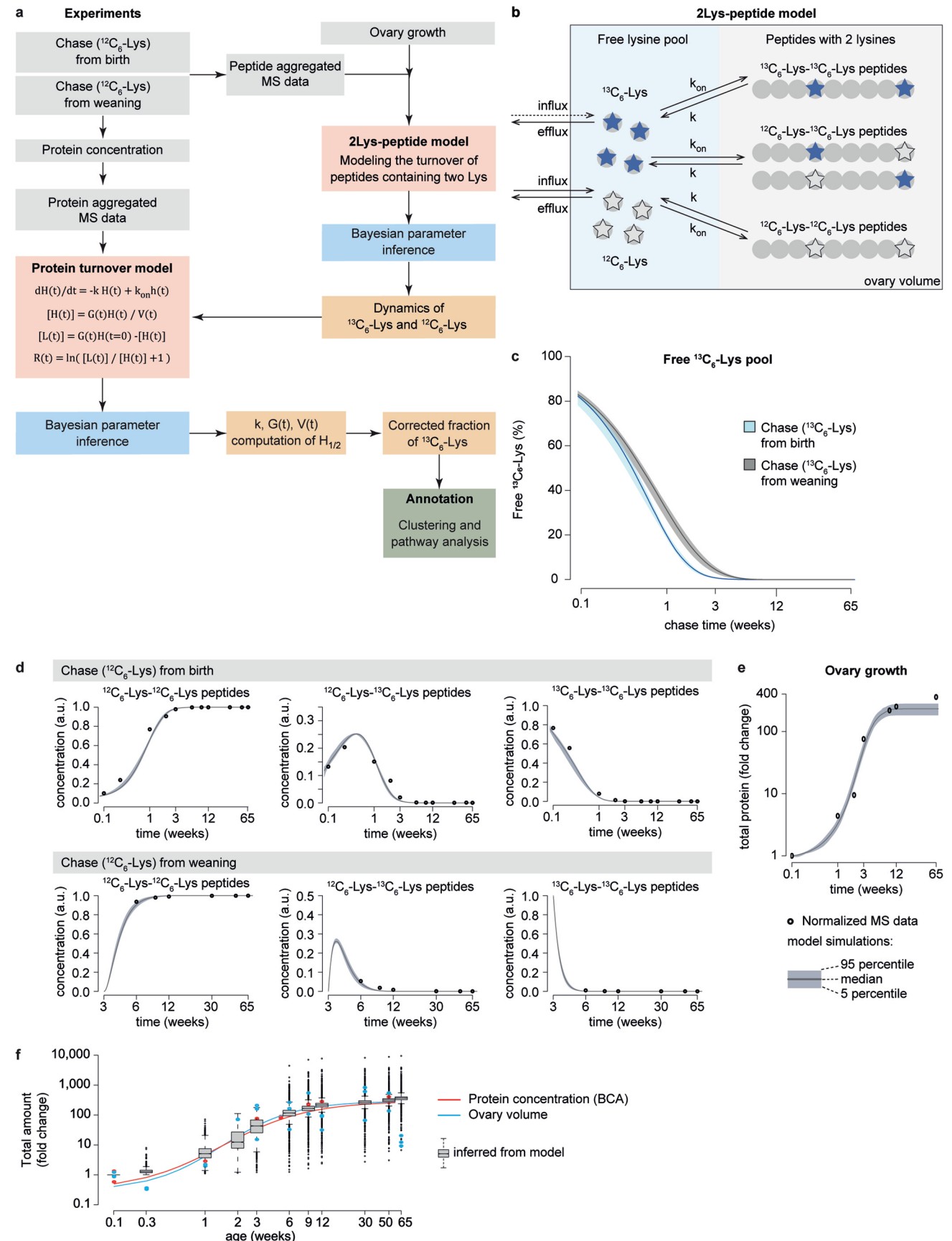

**Extended Data Fig. 3 | See next page for caption.**

**Extended Data Fig. 3 | Modelling protein turnover in the ovary.** (**a**) Overview of the protein turnover modelling approach described in supplemental materials. (**b**) Graphical illustration of the 2Lys-peptide model to determine the fraction of free $^{13}C_6$-Lys during ovarian ageing. (**c**) Estimated fraction of free $^{13}C_6$-Lys for the 2Lys-peptide model for the chase ($^{12}C_6$-Lys) from birth and from weaning experiments. Solid lines indicate median; shaded areas indicate 5% and 95% confidence ranges. (**d-e**) Experimental data and model fits of 2Lys-peptide model (d) and total ovary protein fold change (e) over chase time for the chase ($^{12}C_6$-Lys) from birth and from weaning experiments. Dots indicate experimental data; lines and shaded areas indicate median and confidence ranges of model fits. (**f**) Change of total protein amount in the ovary over mouse age. Total protein amount was determined from BCA measurements and compared to changes in the volume of the ovary. Boxplots indicate the estimated fold change in total protein amount. Boxplots indicate median, 1st quartile, 3rd quartile, as well as minimum and maximum after outlier removal over 3,078 modelled proteins. Even though the volume of the ovary was not considered in the protein turnover model fitting, the estimated fold changes are in good agreement with experimentally measured total protein amount fold changes. Source numerical data are available in source data.

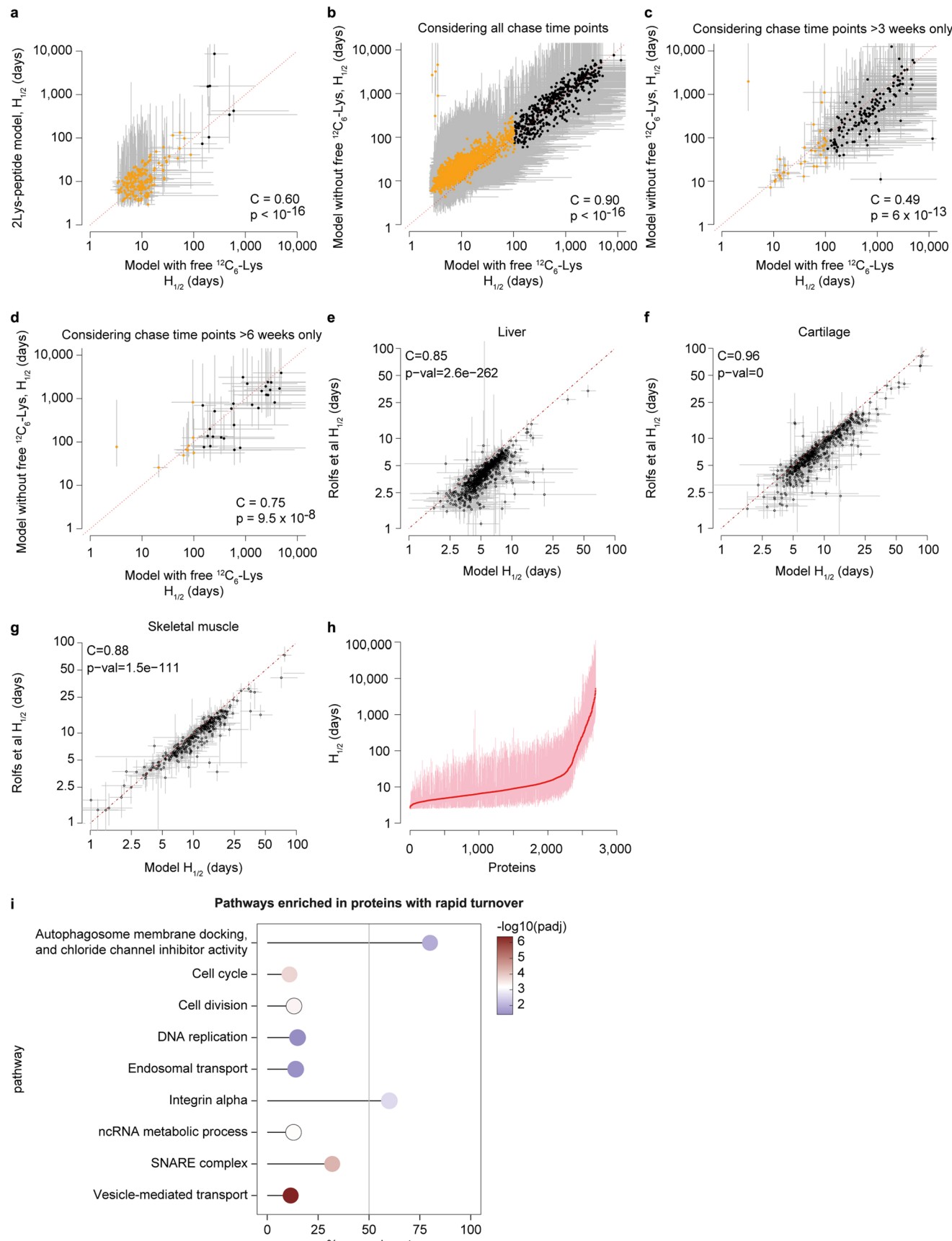

**Extended Data Fig. 4 | See next page for caption.**

**Extended Data Fig. 4 | Comparison of the $H_{1/2}$ values estimated from different models.** (**a-d**) Comparison of estimated $H_{1/2}$ values resulting from the protein turnover model considering free $^{13}C_6$-Lys pool with 2Lys-peptide based model (a) and the 'classical' protein turnover model not allowing reincorporation of $^{13}C_6$-Lys into newly synthesized proteins considering all chase time points (b), or only chase time points larger than 3 weeks (c) or 6 weeks (d), respectively. Dots indicate medians, grey lines indicate confidence ranges. All proteins with estimated $H_{1/2}$ values < 100 days are indicated as orange dots. (**e-g**) Comparison of previously published $H_{1/2}$ values (Rolfs et al.[20]) and $H_{1/2}$ values calculated for the same published datasets using the protein turnover model developed in this manuscript for ovaries. Shown are comparisons for liver (e), cartilage (f) and skeletal muscles (g). In (a-g) test for association between paired samples using Spearman correlation coefficient (C) was performed with p-values (p) estimated using algorithm AS 89. $p < 10^{-16}$ indicates approximated p-values. (**h**) Rank plot showing median (red dots) and confidence ranges (pink lines) of estimated $H_{1/2}$ values for all modelled proteins in ovaries. (**i**) Over-representation analysis of proteins with rapid turnover in ovaries. Shown are the percentage of genes detected in the gene set with their respective adjusted p-values for the most prominent pathways. The p-values are based on the hypergeometric test and have been adjusted for multiple hypothesis testing with Benjamini–Hochberg (BH) procedure. For the complete list of enriched pathways and their corresponding exact p-values, see Supplementary Table 4. Source numerical data are available in source data.

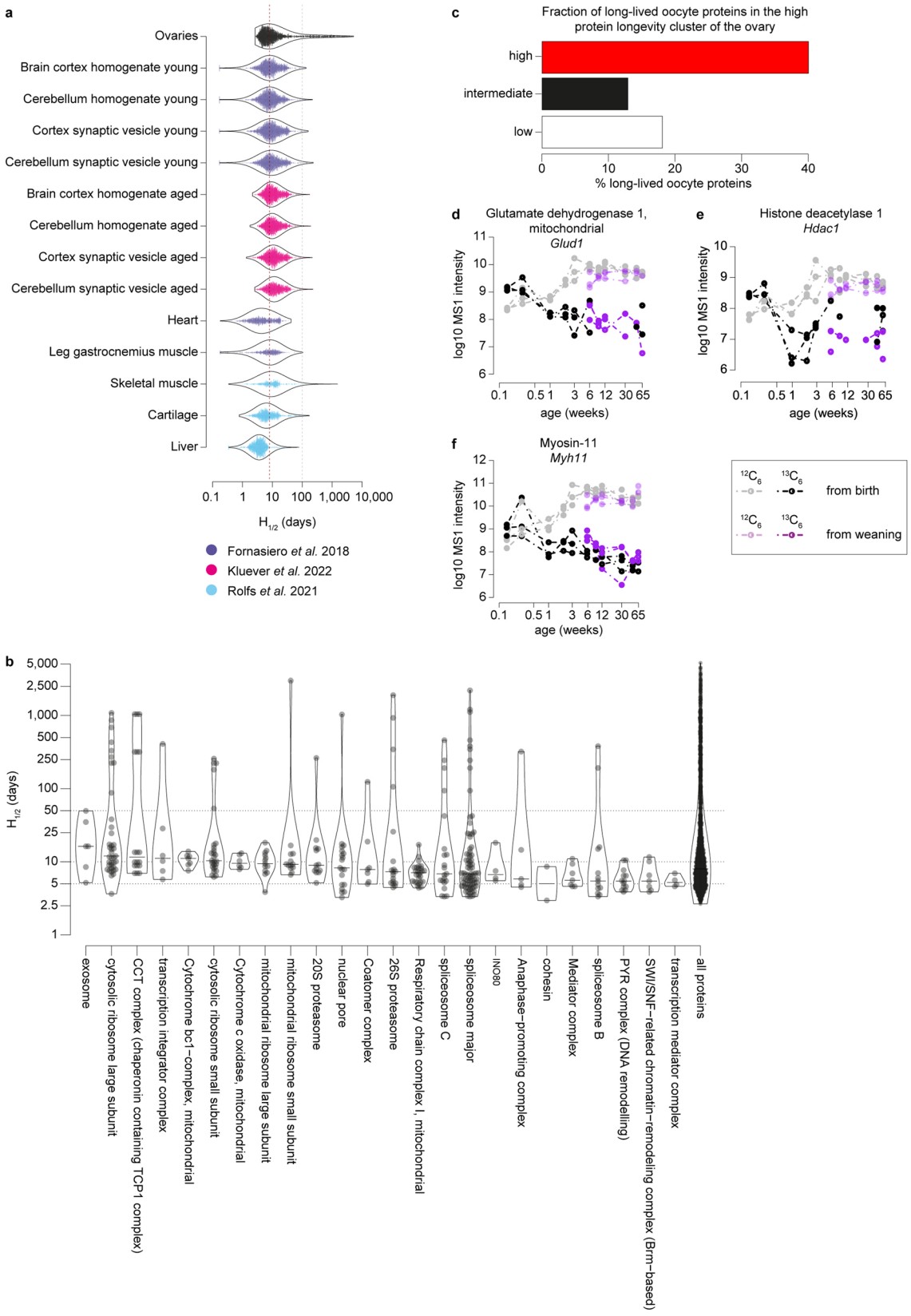

**Extended Data Fig. 5 | See next page for caption.**

**Extended Data Fig. 5 | Comparison of the $H_{1/2}$ values of the different proteins in the ovary with their corresponding $H_{1/2}$ values in other organs, and examples of long-lived proteins in the ovary.** (**a**) Distribution of $H_{1/2}$ determined for various mouse and rat tissues across three studies (Fornasiero et al.[19]; Kluever et al.[21]; Rolfs et al.[20]) compared to the estimated $H_{1/2}$ distribution in mouse ovary samples in this study. Red line indicates the median $H_{1/2}$ values in the ovary, while the grey line indicates a $H_{1/2}$ value of 100. (**b**) Long-lived members of protein complexes. Shown are protein complexes with long-lived proteins as violin plots. Dots indicate individual proteins. Complexes are sorted by median complex $H_{1/2}$

values. (**c**) Fraction of long-lived oocyte proteins in the high protein longevity cluster of the ovary. (**d-f**) Example MS1 intensities over time in log10 scale for $^{13}C_6$-Lys- (black and purple) and $^{12}C_6$-Lys- (grey and pink) labelled peptides for two pulse lengths (pulse with $^{13}C_6$-Lys until birth and pulse with $^{13}C_6$-Lys until weaning) for glutamate dehydrogenase 1, mitochondrial (d); histone deacetylase 1 (e); and Myosin-11 (f). Circles indicate experimental data points for three biological replicates, dashed lines are only for visual aid. Source numerical data are available in source data.

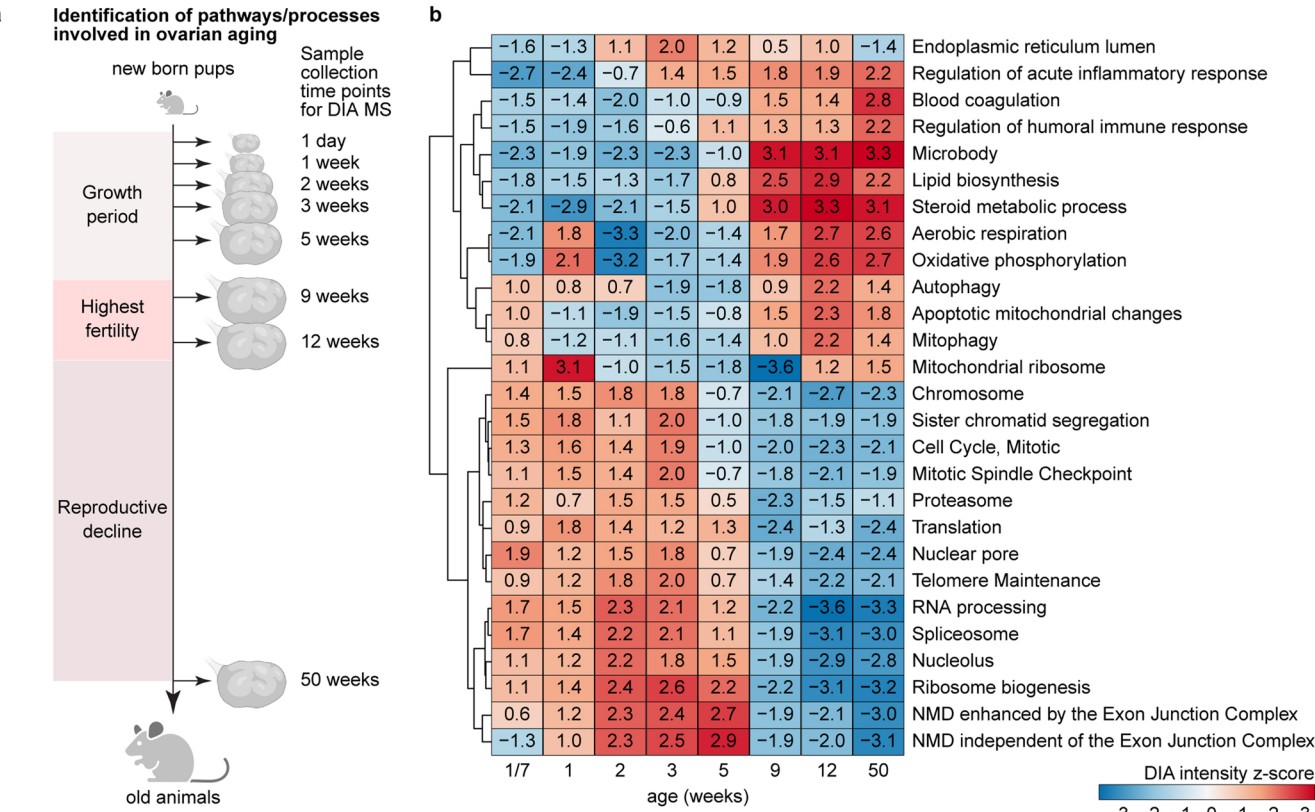

**Extended Data Fig. 6 | Changes in enrichment scores of different pathways throughout ovarian development and aging.** (**a**) Schematic overview of experimental design. Ovaries were collected from female mice at 8 time points (1 day, 1, 2, 3, 5, 9, 12 and 50-week-old mice; three animals per time point analysed). Samples were processed for DIA-MS. (**b**) Gene set enrichment analysis (GSEA) revealed up- and down-regulated pathways over mouse age. For each time point (1 day, 1, 2, 5, 9, 12 and 50 weeks) normalized DIA-MS data for 8,890 detected proteins were subject to GSEA. Normalized enrichment scores for each pathway and time point are shown as heatmap upon hierarchical clustering (dendrogram shown on the left). Each term is significantly enriched at least in one time point. Source numerical data are available in source data.

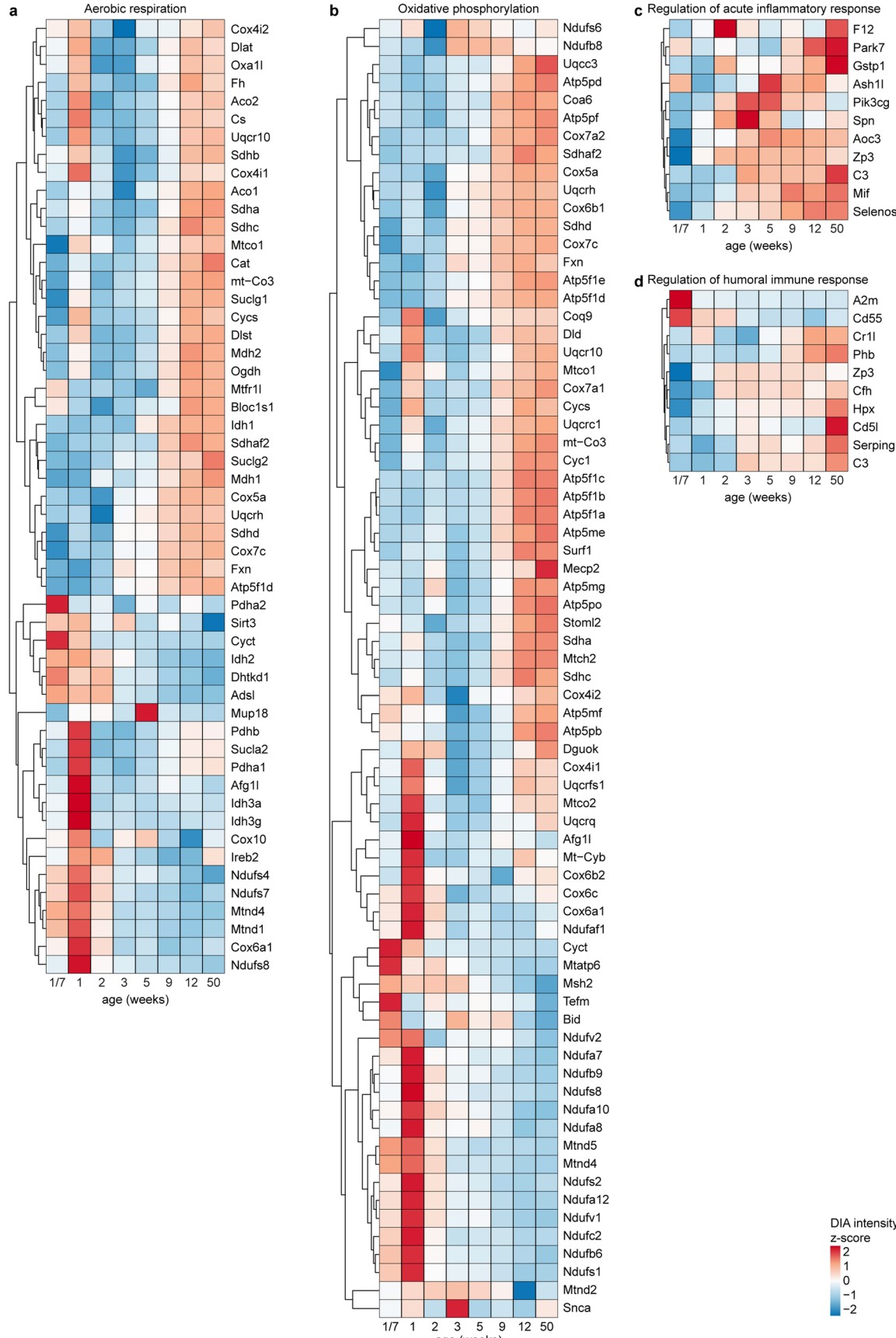

**Extended Data Fig. 7 | Changes in abundance of different proteins belonging to different gene sets throughout ovarian development and aging.** (**a-d**) Normalized DIA-MS intensity data showing the abundance changes over time of the proteins in the selected pathways significantly enriched in the gene-set enrichment analysis (GSEA) of the DIA-MS data: aerobic respiration (a), oxidative phosphorylation (b), regulation of acute inflammatory response (c), and regulation of humoral immune response (d). Source numerical data are available in source data.

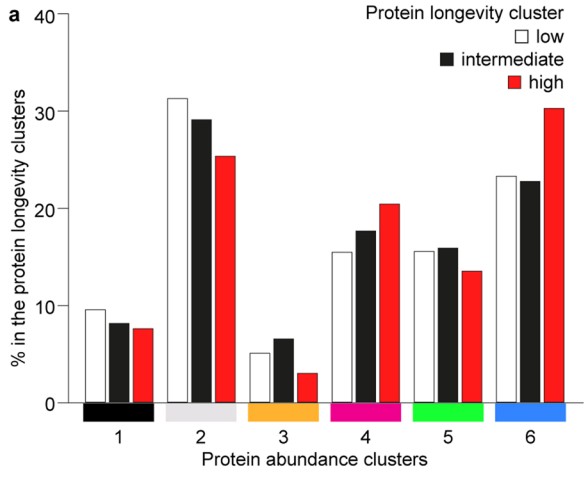

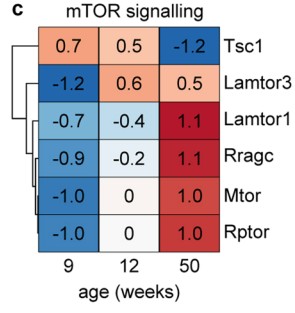

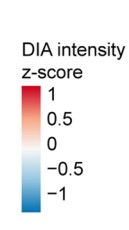

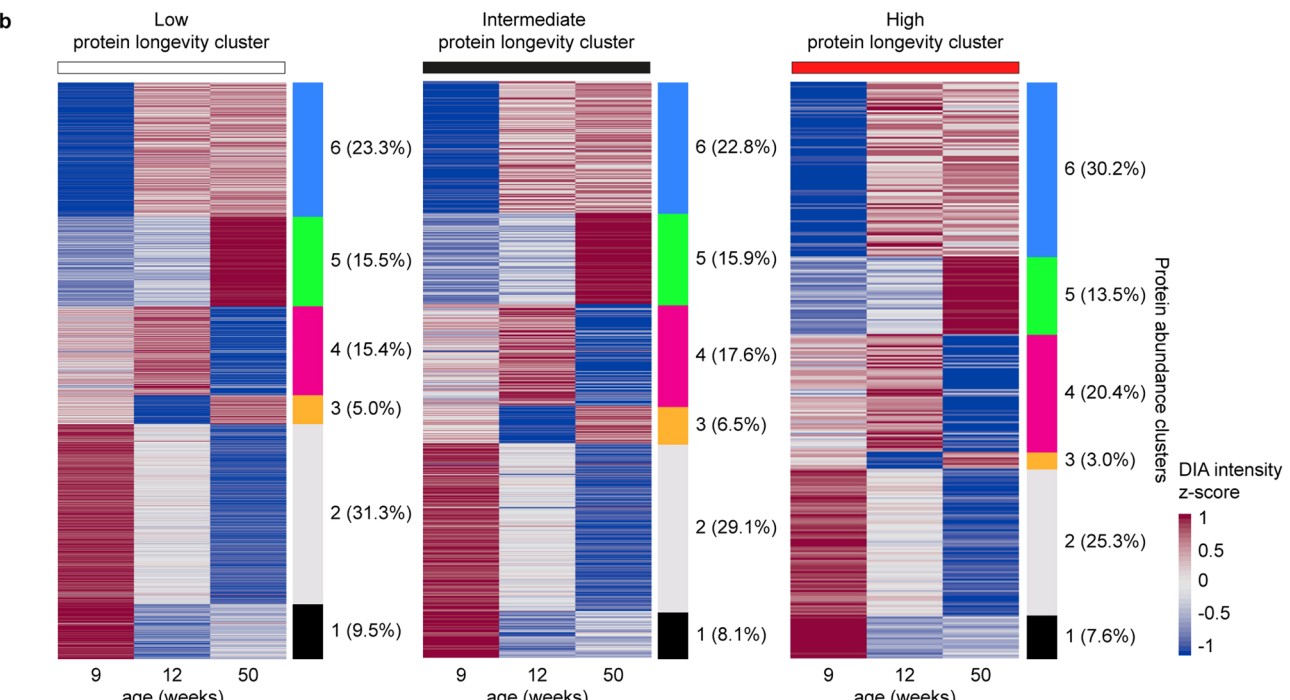

**Extended Data Fig. 8 | Distribution of ovary proteins in the different protein longevity clusters and protein abundance clusters, and abundance changes over time of proteins in the mTOR signalling pathway. (a)** Percentage of ovary proteins that are detected across the three protein longevity clusters, as well as in one of the six protein abundance clusters derived from the DIA-MS data.

**(b)** Heatmaps showing protein abundance change for low, intermediate, and high protein longevity clusters with assignment to protein abundance clusters.
**(c)** Normalized DIA-MS intensity data showing the abundance changes over time of proteins in the mTOR signalling pathway. Source numerical data are available in source data.

|  |  |
|---|---|
|  | Prof Melina Schuh |
|  | Prof Henning Urlaub |

# Reporting Summary

## Statistics

For all statistical analyses, confirm that the following items are present in the figure legend, table legend, main text, or Methods section.

| n/a | Confirmed |  |
|---|---|---|
| ☐ | ☒ | The exact sample size (*n*) for each experimental group/condition, given as a discrete number and unit of measurement |
| ☐ | ☒ | A statement on whether measurements were taken from distinct samples or whether the same sample was measured repeatedly |
| ☐ | ☒ | The statistical test(s) used AND whether they are one- or two-sided<br>*Only common tests should be described solely by name; describe more complex techniques in the Methods section.* |
| ☒ | ☐ | A description of all covariates tested |
| ☐ | ☒ | A description of any assumptions or corrections, such as tests of normality and adjustment for multiple comparisons |
| ☐ | ☒ | A full description of the statistical parameters including central tendency (e.g. means) or other basic estimates (e.g. regression coefficient) AND variation (e.g. standard deviation) or associated estimates of uncertainty (e.g. confidence intervals) |
| ☐ | ☒ | For null hypothesis testing, the test statistic (e.g. *F*, *t*, *r*) with confidence intervals, effect sizes, degrees of freedom and *P* value noted<br>*Give P values as exact values whenever suitable.* |
| ☐ | ☒ | For Bayesian analysis, information on the choice of priors and Markov chain Monte Carlo settings |
| ☒ | ☐ | For hierarchical and complex designs, identification of the appropriate level for tests and full reporting of outcomes |
| ☒ | ☐ | Estimates of effect sizes (e.g. Cohen's *d*, Pearson's *r*), indicating how they were calculated |

*Our web collection on statistics for biologists contains articles on many of the points above.*

## Software and code

Policy information about availability of computer code

| Data collection | For mass spectrometry data acquisition: Thermo Fisher Scientific software: Thermo Xcalibur Instrument Setup, Thermo Scientific Xcalibur, Version 4.4.16.14, Tune Application, Version 4.0.309.28 (for monitoring detection). For confocal microscopy image acquisition: Zen Blue 3.2 and Zen Black 2.1. |
|---|---|
| Data analysis | R (version 4.1.2); MaxQuant software versions 1.6.0.1 (for analysis of DDA mass spectrometry data) and 1.5.2.8 (for analysis of blood samples); Spectronaut version 15.7.220308.50606 (for analysis of DIA-MS data).<br>R scripts for data processing and visualization of proteomics and single-cell RNA sequencing data are available on figshare (https://doi.org/10.6084/m9.figshare.25604232.v1).<br>No custom algorithm or software were generated in this study. |

For manuscripts utilizing custom algorithms or software that are central to the research but not yet described in published literature, software must be made available to editors and reviewers. We strongly encourage code deposition in a community repository (e.g. GitHub). See the Nature Portfolio guidelines for submitting code & software for further information.

## Data

Policy information about availability of data

All manuscripts must include a data availability statement. This statement should provide the following information, where applicable:
- Accession codes, unique identifiers, or web links for publicly available datasets
- A description of any restrictions on data availability
- For clinical datasets or third party data, please ensure that the statement adheres to our policy

> The scRNAseq datasets are deposited in the NCBI Gene Expression Omnibus database under accession number GSE237012. The proteomics data are deposited to the ProteomeXchange Consortium via the UCSD MassIVE repository with the dataset identifier MSV000092528.

## Research involving human participants, their data, or biological material

Policy information about studies with human participants or human data. See also policy information about sex, gender (identity/presentation), and sexual orientation and race, ethnicity and racism.

| | |
|---|---|
| Reporting on sex and gender | N/A |
| Reporting on race, ethnicity, or other socially relevant groupings | N/A |
| Population characteristics | N/A |
| Recruitment | N/A |
| Ethics oversight | N/A |

Note that full information on the approval of the study protocol must also be provided in the manuscript.

# Field-specific reporting

Please select the one below that is the best fit for your research. If you are not sure, read the appropriate sections before making your selection.

☒ Life sciences ☐ Behavioural & social sciences ☐ Ecological, evolutionary & environmental sciences

For a reference copy of the document with all sections, see nature.com/documents/nr-reporting-summary-flat.pdf

# Life sciences study design

All studies must disclose on these points even when the disclosure is negative.

| | |
|---|---|
| Sample size | scRNA sequencing and proteomics data: no groups of conditions were compared to each other, and therefore, no sample-size calculation was performed. <br> Figures 6 and 7: No statistical methods were used to predetermine sample size. In retrospective, achieved sample sizes were determined to be adequate based on the magnitude and consistency of measurable differences between groups. Most importantly, sample size per experiment was dictated by the number of oocytes that could be processed (microinjection and live imaging) within a reasonable time by the researcher without affecting oocyte quality. |
| Data exclusions | Proteomics data: <br> In the oocyte protein turnover experiments (data-dependent acquisition [DDA] mass spectrometry [MS]), median over technical replicates were only computed if the DDA intensity of 13C6-Lys labeled protein was detected in at least 2 out of 4 technical replicates, otherwise the data point is omitted (set as n.a.). The mean and standard deviation of F, fractions of heavy (13C6-Lys) proteins left, were calculated if F was detected in at least one of the two biological replicates. <br><br> Figures 6 and 7: <br> No data points were excluded. |
| Replication | 1. In the oocyte protein turnover experiments (data-dependent acquisition [DDA] mass spectrometry [MS]), two sets of oocytes (n=2,475, n= 2,473) were obtained from a total of 8-week old mice (12C6-Lys chow after birth until they were 8-weeks old) born from 13C6-Lys-labeled females. Oocytes from multiple animals were pooled for a single replicate to reach the minimum input required for the MS processing. Each DDA oocyte fraction was measured in quadruplicates. <br><br> 2. In the ovary protein turnover experiments (DDA MS) with 13C6-Lys pulse until birth and 12C6-Lys chase from birth, ovaries were collected from the female progeny at 11 time points (24 hours, 48 hours, 1, 2, 3, 6, 9, 12, 30, 50, and 65 weeks). For each time point, three biological replicates were collected, each containing two ovaries from a single animal. |

3. In the ovary protein turnover experiments (DDA MS) with 13C6-Lys pulse until birth and 12C6-Lys chase from weaning, ovaries were collected from the female progeny at 6 time points (6, 9, 12, 30, 50, and 65 weeks). For each time point, three biological replicates were collected, each containing two ovaries from a single animal.

4. In the protein abundance experiments (data-independent acquisition [DIA] MS), ovaries were collected from unlabeled FVB/N female mice at 8 time points (24 hours, 1, 2, 3, 5, 9, 12, and 50 weeks). For each time point, three biological replicates were collected, each containing two ovaries from a single animal.

5. In the single-cell RNA sequencing experiments, each library was generated from 6 ovaries collected from 3 different pups in a single pregnant CD1 female mouse.

6. Figures 6 and 7: All data are from at least two independent experiments / multiple biological replicates. All attempts at replication were successful.

| Randomization | Protein turnover and scRNA sequencing experiments did not have different treatments or conditions that could be randomized. |
| --- | --- |
| | Figures 6 and 7: Mouse oocytes were collected from multiple ovaries/animals and then pooled. They were then randomly assigned to different experimental groups. |

| Blinding | scRNA sequencing and proteomics data: No groups of conditions were compared to each other, and therefore, these experiments did not require blinding. |
| --- | --- |
| | Figures 6 and 7: Investigators were not blinded to allocation during experiments and outcome assessment, as each experiment was performed by one researcher alone. Thus, blinding during group allocation was not possible to ensure samples received the right treatment/manipulation during experiment. Blinding was also not possible during data analysis, as data analysis was performed by the same researcher that conducted the experiment. |

# Reporting for specific materials, systems and methods

We require information from authors about some types of materials, experimental systems and methods used in many studies. Here, indicate whether each material, system or method listed is relevant to your study. If you are not sure if a list item applies to your research, read the appropriate section before selecting a response.

## Materials & experimental systems

| n/a | Involved in the study |
| --- | --- |
| ☒ | Antibodies |
| ☒ | Eukaryotic cell lines |
| ☒ | Palaeontology and archaeology |
| ☐ | ☒ Animals and other organisms |
| ☒ | Clinical data |
| ☒ | Dual use research of concern |
| ☒ | Plants |

## Methods

| n/a | Involved in the study |
| --- | --- |
| ☒ | ChIP-seq |
| ☒ | Flow cytometry |
| ☒ | MRI-based neuroimaging |

## Animals and other research organisms

Policy information about studies involving animals; ARRIVE guidelines recommended for reporting animal research, and Sex and Gender in Research

| Laboratory animals | Mus musculus, FVB/N and CD1 strains. All mice were kept in rooms with constant temperature of 21°C and the humidity of 55%. The light/dark rhythm was 12:12 hours, from 5 am to 5 pm. Health monitoring was carried out in accordance with FELASA recommendations with large annual examinations in January and smaller scale in May and September. We used animals of different ages, between 1 day and 65 weeks old. Age of animals is clearly indicated in each figure. |
| --- | --- |
| Wild animals | The study did not involve wild animals. |
| Reporting on sex | In this study we investigated protein turnover and abundance in ovaries and oocytes. Hence, only animals of female sex were analysed. |
| Field-collected samples | This study did not involve field-collected samples. |
| Ethics oversight | The maintenance and handling of all FVB/N mice was performed according to international animal welfare rules (Federation for Laboratory Animal Science Associations guidelines and recommendations). Requirements of formal control of the German national authorities and funding organizations were satisfied, and the study received approval by the Niedersächsisches Landesamt für Verbraucherschutz und Lebensmittelsicherheit (LAVES). |

Note that full information on the approval of the study protocol must also be provided in the manuscript.

## Plants

| | |
|---|---|
| Seed stocks | N/A |
| Novel plant genotypes | N/A |
| Authentication | N/A |

