## [Peer Review File · Nature Cell Biology]

Peer Review Information

Journal: Nature Cell Biology

Manuscript Title: The maintenance of oocytes in the mammalian ovary involves extreme protein longevity

Corresponding author name(s): Professor Melina Schuh

Editorial Notes:

Reviewer Comments & Decisions:

Decision Letter, initial version:
--

*Please delete the link to your author homepage if you wish to forward this email to co-authors.

Dear Melina,

Thank you for submitting your manuscript, "The maintenance of oocytes in the mammalian ovary involves extreme protein longevity", to Nature Cell Biology, and I am very sorry we could not send you our decision more quickly. Your manuscript has now been seen by 3 referees, who are experts in oocyte biology (Referee #1); proteostasis, aging (Referee #2); and proteomics (Referee #3). As you will see from their comments (attached below), they found the work of potential interest but have raised substantial concerns that in our view would need to be addressed with considerable revisions before we can consider publication in Nature Cell Biology.

As per our standard editorial process, we have now internally discussed the reviews with our Chief Editor to determine what key reviewer points should be addressed with priority. To guide the scope of the revisions, I have listed these points below. As you know, our standard revision period is six months, and we are committed to providing a fair and constructive peer-review process, so please feel free to contact me if you would like to discuss any of the referee comments further or if you anticipate challenges or delays addressing the reviews.

In particular, in our view, it would be essential to:

1- The reviewers requested more information and checks on the methods and analytical aspects that should be addressed thoroughly:

Rev#1 points #2, 3, 4

2- The proteostasis reviewer suggested functional analyses to probe the hypothesis that long-lived proteins may play a role in proteostasis - including in aging. We agree some further test of this line of thinking is needed, even if some analyses (e.g., studies of autophagy-deficient mice as they age) may be beyond the scope of a reasonable revision period unless they have been ongoing in the lab or the animals are otherwise readily available. However, using appropriate cell models at least, we feel that further work could be done to answer the reviewer's questions:

"One question the authors could address is the role of the most stable proteins in proteostasis. Can overexpression in tissue culture or in an organismic model (e.g., *C. elegans*) promote resistance to proteotoxic stress and/or longevity? How resistant are ovarian cells to stress conditions? The authors could test proteasomal activity/protein aggregation at young and old stages in oocytes. What happens to stable proteins during ageing if autophagy or proteasomal degradation is blocked, if this is

experimentally possible.”

Please see also Rev#3's point #3

3- Lastly, the reviewers had questions about some of the data that should be clarified and resolved:

Rev#1 point #5 and other points

Rev#3 points #1, 2, 4, 5, 7, 8, 9, 10

4- Finally, please pay close attention to our guidelines on statistical and methodological reporting (listed below) as failure to do so may delay the reconsideration of the revised manuscript. In particular, please provide:

We would be happy to consider a revised manuscript that would satisfactorily address these points, unless a similar paper is published elsewhere, or is accepted for publication in Nature Cell Biology in the meantime.

- ensure that it conforms to our format instructions and publication policies (see below and <https://www.nature.com/nature/for-authors>).

- provide a point-by-point rebuttal to the full referee reports verbatim, as provided at the end of this letter.

- provide the completed Reporting Summary (found here <https://www.nature.com/documents/nr-reporting-summary.pdf>). This is essential for reconsideration of the manuscript will be available to editors and referees in the event of peer review. For more information see <http://www.nature.com/authors/policies/availability.html> or contact me.

When submitting the revised version of your manuscript, please pay close attention to our [href="https://www.nature.com/nature-portfolio/editorial-policies/image-integrity">Digital Image Integrity Guidelines](https://www.nature.com/nature-portfolio/editorial-policies/image-integrity). and to the following points below:

- that unprocessed scans are clearly labelled and match the gels and western blots presented in figures.

- that control panels for gels and western blots are appropriately described as loading on sample processing controls

-- all images in the paper are checked for duplication of panels and for splicing of gel lanes.

Nature Cell Biology is committed to improving transparency in authorship. As part of our efforts in this direction, we are now requesting that all authors identified as 'corresponding author' on published papers create and link their Open Researcher and Contributor Identifier (ORCID) with their account on the Manuscript Tracking System (MTS), prior to acceptance. ORCID helps the scientific community achieve unambiguous attribution of all scholarly contributions. You can create and link your ORCID from the home page of the MTS by clicking on 'Modify my Springer Nature account'. For more information please visit www.springernature.com/orcid.

This journal strongly supports public availability of data. Please place the data used in your paper into a public data repository, or alternatively, present the data as Supplementary Information. If data can only be shared on request, please explain why in your Data Availability Statement, and also in the correspondence with your editor. Please note that for some data types, deposition in a public repository is mandatory - more information on our data deposition policies and available repositories appears below.

[Redacted]

We hope that you will find our referees' comments and editorial guidance helpful. Please do not hesitate to contact me if there is anything you would like to discuss. Thank you again for considering NCB for your work.

Best wishes,

Melina

Melina Casadio, PhD
Senior Editor, Nature Cell Biology
ORCID ID: <https://orcid.org/0000-0003-2389-2243>

Reviewers' Comments:

Reviewer #1:

Remarks to the Author:

This is yet another impressive body of work from the Schuh lab! The manuscript discovers in eggs and ovaries there is an abundance of long-lived proteins that appears to exceed that of other cell types. The measuring of protein lifetime and turnover in oocytes and ovaries, includes an analysis of the impact of ageing.

In terms of the relevance of my review, I have considerable experience in oocyte cell biology, but proteomics remains far from my area of expertise, and I'm afraid I don't have time to master the details for the purpose of this review. Furthermore, a lot of the data is dependent on modelling that I am not able to critique in detail. It does however raise a few questions for me, which are elaborated below. The technical approaches, the analytical approaches, and modelling requires a truly multi-disciplinary review team to cover this one. I will focus on oocyte biology and conceptual matters relating to data interpretation. I am mightily impressed with the whole thing, so please do take these comments as a constructive effort to make sure the data is as right as it can be and, to improve the life of the interested reader.

1. First, a comment on the presentation of the manuscript. It is somewhat challenging to review because the information in the text in the main body of the manuscript is limited, and the figures, extended data and supplementary data and methods are all extensive, and an important part of the story. It is a shame to squeeze such a manuscript into a tight word limit and I would prefer to see more of the data in the figures (moved from extended data) and more description of the modelling as part of the central story. Currently the Figures are largely representative summaries of bioinformatic analysis, while much of the data is 'extended'. Further, the complexity of the quantitative approaches, and any possible limitations, can only be appreciated by those brave enough to delve deeply into the methods section.

2. One question I have concerning the assay is that it assumes the presence of $^{13}\text{C}_6$ -Lys in a given protein originates from the time it was synthesised during the original pulse. Is it possible that on protein catabolism the $^{13}\text{C}_6$ -Lys re-enters the amino acid pool for utilisation in subsequent protein synthesis? I note this has been incorporated into the model to determine the potential impact on the %H (methods and Extended data 4C). The modelling as outlined in the methods is impressive, but there is little description of assumptions and limitations that could influence the data. A circumspect description of the model including potential limitations is worth considering.

3. In the same light, the identification of peptides with both $^{13}\text{C}_6$ -Lys and $^{12}\text{C}_6$ -Lys, demonstrates that there is overlap between the pulse and the chase, which the authors acknowledge and make an effort to deal with (described in detail in the methods, but only a cursory mention in the text). Oocytes may also be very effective at pooling and recycling amino acids and this could potentially explain some of the longevity. Somewhere in the data I expect there is an answer to this question, but I'm not sure if it is based on the actual measurements of $^{13}\text{C}_6$ -Lys, the modelling, or both.

4. The modelling used to measure protein longevity in oocytes raises a couple of questions about some aspects of the manuscript.

(1) One area of immediate concern is that a critical part of the model is accounting for the dilution of proteins as the oocyte/ ovary undergoes a dramatic increase in volume during the pulse phase. Yet the measurements of primordial oocyte diameter have been made using histological sections while

that of fully grown oocytes seems to be from archived fixed samples/sections. The variability in the data seems huge (Extended Fig 2b). Fixation and histological processing will not provide an accurate measure of oocyte diameter. Given the formula for determining volume of a sphere, inaccuracies at this point are a potential source of compromise in the modelling. I would recommend the authors determine if the measurements of oocyte diameter reflect the physiological reality. If differences exist, please determine the impact this has on the analysis of half-life.

(2) How valid it is to make comparisons of protein half-life in oocytes and ovaries with that obtained for other tissues in prior studies (Extended data 7). Presumably the complexity of the modelling in oocytes and ovaries means that the approaches used are very different. It would be ideal if the authors could compare other tissues analysed by the same approach as used here.

(3) The vast majority of proteins in different tissues have the same profile as shown in Extended Data 7). The key finding in this data is, if I understand it correctly, all about the right-hand end of the graph where there is a higher density of label present than in other tissues right out to 1000 days – well over a mouse lifetime! Now this may well be right, but I would want to try to prove myself wrong to be sure of this. Testing different parameters of the model to see if any are unexpectedly able to modulate longevity might be worth testing. Isn't it a little concerning that the density does not continue to gradually decline between 100 and 1000 days – it actually seems to increase a little, before taking a little dip after 1000 days?

(4) Conceptually it makes sense that oocytes may have reduced protein turnover with more effort going into creating a stable environment to minimise the effects of ageing in the germ plasm. However, it is less obvious why somatic cells in the ovary, including stromal cells, have a similarly high proportion of long-lived proteins. Again it raises the question for me if this represents a fundamental difference in the modelling approach in this study compared to the previously published examples used as comparators. Would other tissues show similar long lived proteins if the model in this manuscript was applied to them?

5. In Extended data 7d, is it possible to provide an indication as to how the log of MS1 relates to the level of protein that would be present in the ovary. For most proteins it appears there is about 1000-fold reduction (from 10 to 7 on the log scale) between birth and around 30 weeks, with some variability between proteins. Is an MS1 of 7 effectively when there is no 13C6-Lys remaining?

While on this figure, what is happening to HDAC1. It seems to have a major kick in 13C6-Lys in ovaries between 1 and 6 weeks, despite the pulse stopping at birth?

Other points:

Ln 118-122: The pulse labelling till weaning will label oocytes undergoing the first wave of growth from the start to completion. This wave typically starts just after birth and can be super ovulated by administering PMSG/hCG just after weaning (21-23 days). Thus, the following statement doesn't seem strictly correct, and the significance of it is not explained. Clarification would help. "This approach labels all proteins synthesized before birth, as well as those 121 synthesized up to when the first wave of resting primordial follicles are recruited into the growing follicle pool 11,13 122 ." It is actually labelling protein synthesis to the end of the first wave of oocyte/follicle growth.

Ln 126-131 – It is almost impossible to read this with any continuity as there is a need to continually flick between multiple figures. As mentioned above - Is there a model whereby the essential data

needed to evaluate a particular point is presented in single figure? It's unnecessarily obfuscating for the reader.

Extended data Fig 2 (Legend) and methods: This seems an arbitrary definition of a fully labelled protein? What is happening in the other two replicates where the C6-Lys is not detected and what is happening with the biological replicates. There is a little more elaboration in Methods, but please provide a reason for why you chose 2/4 replicates.

"A protein was defined as fully labelled (%H value of 100%) if it was detected as a 13 C6-Lys labelled protein in at least two out of four technical replicates in at least one biological replicate and there is no signal for 12 1496 C6-Lys labeled protein in any of the 1497 four technical replicates of both biological replicates detected."

Ln 387-390: Use of M2 with no BSA – was any other supplement used to prevent the oocytes from sticking and ease handling?

Reviewer #2:

Remarks to the Author:

Harasimov et al. used large-scale mass spectrometry to study the stability of the proteome in the mammalian ovary. Oocytes are produced at birth and must be maintained undamaged to ensure 100% female reproductive capacity. In addition to genome stability, the proteome must be well protected to ensure the propagation of germ cells over several generations. The cellular proteostasis network supports efficient protein synthesis, protein folding and protein degradation. However, it remained unclear how the oocyte-specific proteome is protected. The authors performed large-scale proteomics combined with pulse-chase/time course experiments in female mice and identified a surprisingly large number of proteins that are extremely long-lived, with a 10-fold increase compared to other post-mitotic tissues! These highly stable oocyte/ovarian proteins are functionally diverse and are associated with mitochondrial function, metabolism, cytoskeleton, chromatin organisation and also proteostasis. However, the same proteins show a rapid decline in the ageing ovary, suggesting that their stability maintains oocytes throughout the reproductive life of the female.

The manuscript is very well written and reflects a tour de force in addressing proteome remodelling in the mammalian ovary associated with oocyte maintenance and age-related decline in reproduction. Although the work is mainly based on comprehensive proteomic approaches, it provides an amazingly detailed catalogue of highly stable proteins that change with age in the ovary. As such, the work is fundamental to future studies investigating the mechanisms that support proteome stability, which I believe is beyond the scope of the current manuscript.

Questions:

One question the authors could address is the role of the most stable proteins in proteostasis. Can overexpression in tissue culture or in an organismic model (e.g. *C. elegans*) promote resistance to proteotoxic stress and/or longevity? How resistant are ovarian cells to stress conditions? The authors could test proteasomal activity/protein aggregation at young and old stages in oocytes. What happens to stable proteins during ageing if autophagy or proteasomal degradation is blocked, if this is experimentally possible.

Reviewer #3:

Remarks to the Author:

The authors present a deep proteomic analysis of proteome dynamics in murine oocytes and ovaries. Using ^{13}C pulse chase experiments, the authors were able to monitor the loss of ^{13}C incorporation in oocytes proteomes across multiple time ranges. The result is a series of datasets consisting of proteins from diverse biological processes (OXPHOS to cohesin). The study is clearly a tour de force of proteomics analyses, with 4948 oocytes analyzed and presented in just the first figure. Moreover, the integration of multiple techniques aids in the potential for use within this study and beyond. All that being said, I had a few comments and questions that arose in reading this work that I think should be addressed prior to publication. My concern is that with all of these data the main output seems to be a list of proteins and GO/GSEA terms (autophagy, ER lumen, inflammation from Fornasiero 2018, Kleuver 2022, and Keele 2023), that could be relationally inferred from other mouse aging studies. Thus, there is sometimes a tenuous connection between the wealth of data and a future looking utility of this work.

Major Comments

1. Much of the most interesting data is relegated to the extended data figures. Two panels (Fig 1c and Fig 3d) are disguised tables. While I appreciated the graphics, these held limited value for me as they are effectively lists of >40 proteins. The values without context (i.e., baseline) comparisons are not especially useful, too. What %H is interesting and why? This context seems in part to be buried in Fig S7a. Simple things to add this context that I can imagine: is the %H for the proteasome higher than say chaperones like HSP90? What about the CCT complex that is strongly enriched in Fig 1b?
2. As in work that the authors site (namely Kluever et al 2022, Fig 1D), it would be useful to show the raw data to enable interpretation (and consistent with the authors previous work in Fornasiero 2018 and Kleuver 2022). There are also interesting suggestions from this data, such as why are proteasome subunits in oocytes/ovaries extremely long lived but short lived in Kleuver et al data? Are other proteasome subunits in the intermediate half-life cluster?
 - a. Similarly, different subunits of Complex I in Oxphos have distinct temporal profiles (e.g., NDUFB8 vs. Ndufs8), why is this? Are these from structurally distinct subcomplexes?
3. From the intro statements like this I expected the data to be related back to the "whys" of how these proteome changes explained the principles of female germline maintenance: "Our data suggest that mammals use these principles for maintaining the female germline over long periods of time." I kept waiting to read how the ovary and oocyte protein and protein complex changes were different than other tissues.
4. Nuclear proteome H1/2 is lower than the proteome overall: "nuclear proteins had shorter H1/2 values (Fig. 2g)". Can the authors speculate on why this might be? What proteins in the nucleus are driving this. The effect sizes in Fig 2g are all pretty small, this would suggest that perhaps 1-2 complexes are driving this, how to these relate to the proteins?
5. "hence seems possible that low mitochondrial activity and low protein turnover are regulatory and functionally coupled in mammalian oocytes." Was this actually compared within the data? Is there a significant correlation between mitochondrial energetic proteins and protein chaperones/proteasome/etc?
6. From my interpretation, Fig. S10d seems to represent a logical fallacy: the clustering of proteins based on ^{13}C results in clusters with different levels of ^{13}C .
7. In Fig 4a., are the white (~35%) and black (~65%) bars indicating that all proteins in the high ^{13}C cluster are all enriched or unenriched in at least one specific cell type? Fig 4d and 4h don't appear to have a legend for the coloring. The stromal cells in Fig 4e also appear to have a bimodal distribution?

8. How do the authors reconcile complex phenotypes, such as the drop and then increase in 13C HDAC1 intensity in Figure S7d? Further, do other epigenetic regulators such as HDAC1 and DNMT3A have longer lifetimes than the average proteins? Even if simply in the intermediate categories, this would be striking.
9. Why do the authors believe that there is no significant correlation with other tissues that they've studied (Figure S8)? Especially since the distributions of the ovary proteome in Figure S7a appear to overlap well with other tissues.
10. "Proteins with protective functions and proteins of the proteostasis network..." were enriched in the long lived proteome. Are known substrates of the chaperones, chaperonins, and other proteostasis machinery long lived in these datasets?

Minor Feedback

1. The 13C enrichment appears to have a median value 97.99-98.7% from Fig S1, so half of the peptides have an incorporation less than this. Yet, the manuscript repeatedly mentioned that the incorporation was greater than 98.7% (such as in Fig 2a and 2b).
2. Figure S7a and S7b are genuinely some of the most striking in the paper, and I think better highlight the high/intermediate/low phenotypes than much of the main text allusions to these proteins. I would suggest pushing these into the main text.
3. Why did the authors choose to use trypsin instead of LysC to increase peptide identifications?

AUTHOR AFFILIATIONS – should be denoted with numerical superscripts (not symbols) preceding the

names. Full addresses should be included, with US states in full and providing zip/post codes. The corresponding author is denoted by: "Correspondence should be addressed to [initials]."

Methods should be written concisely, but should contain all elements necessary to allow interpretation and replication of the results. As a guideline, Methods sections typically do not exceed 3,000 words. The Methods should be divided into subsections listing reagents and techniques. When citing previous methods, accurate references should be provided and any alterations should be noted. Information must be provided about: antibody dilutions, company names, catalogue numbers and clone numbers for monoclonal antibodies; sequences of RNAi and cDNA probes/primers or company names and catalogue numbers if reagents are commercial; cell line names, sources and information on cell line identity and authentication. Animal studies and experiments involving human subjects must be reported in detail, identifying the committees approving the protocols. For studies involving human

subjects/samples, a statement must be included confirming that informed consent was obtained. Statistical analyses and information on the reproducibility of experimental results should be provided in a section titled "Statistics and Reproducibility".

All Nature Cell Biology manuscripts submitted on or after March 21 2016 must include a Data availability statement as a separate section after Methods but before references, under the heading "Data Availability". For Springer Nature policies on data availability see <http://www.nature.com/authors/policies/availability.html>; for more information on this particular policy see <http://www.nature.com/authors/policies/data/data-availability-statements-data-citations.pdf>. The Data availability statement should include:

- Accession codes for primary datasets (generated during the study under consideration and designated as "primary accessions") and secondary datasets (published datasets reanalysed during the study under consideration, designated as "referenced accessions"). For primary accessions data should be made public to coincide with publication of the manuscript. A list of data types for which submission to community-endorsed public repositories is mandated (including sequence, structure, microarray, deep sequencing data) can be found here <http://www.nature.com/authors/policies/availability.html#data>.
- Unique identifiers (accession codes, DOIs or other unique persistent identifier) and hyperlinks for datasets deposited in an approved repository, but for which data deposition is not mandated (see here for details <http://www.nature.com/sdata/data-policies/repositories>).
- At a minimum, please include a statement confirming that all relevant data are available from the authors, and/or are included with the manuscript (e.g. as source data or supplementary information), listing which data are included (e.g. by figure panels and data types) and mentioning any restrictions on availability.
- If a dataset has a Digital Object Identifier (DOI) as its unique identifier, we strongly encourage including this in the Reference list and citing the dataset in the Methods.

We recommend that you upload the step-by-step protocols used in this manuscript to the Protocol Exchange. More details can found at www.nature.com/protocolexchange/about.

All imaging data should be accompanied by scale bars, which should be defined in the legend. Cropped images of gels/blots are acceptable, but need to be accompanied by size markers, and to retain visible background signal within the linear range (i.e. should not be saturated). The boundaries of panels with low background have to be demarked with black lines. Splicing of panels should only be

considered if unavoidable, and must be clearly marked on the figure, and noted in the legend with a statement on whether the samples were obtained and processed simultaneously. Quantitative comparisons between samples on different gels/blots are discouraged; if this is unavoidable, it should only be performed for samples derived from the same experiment with gels/blots were processed in parallel, which needs to be stated in the legend.

The total number of Supplementary Figures (not including the “unprocessed scans” Supplementary Figure) should not exceed the number of main display items (figures and/or tables (see our Guide to Authors and March 2012 editorial <http://www.nature.com/ncb/authors/submit/index.html#suppinfo>; <http://www.nature.com/ncb/journal/v14/n3/index.html#ed>). No restrictions apply to Supplementary Tables or Videos, but we advise authors to be selective in including supplemental data.

GUIDELINES FOR EXPERIMENTAL AND STATISTICAL REPORTING

REPORTING REQUIREMENTS – We are trying to improve the quality of methods and statistics reporting in our papers. To that end, we are now asking authors to complete a reporting summary

that collects information on experimental design and reagents. The Reporting Summary can be found here <https://www.nature.com/documents/nr-reporting-summary.pdf>) If you would like to reference the guidance text as you complete the template, please access these flattened versions at <http://www.nature.com/authors/policies/availability.html>.

Author Rebuttal to Initial comments

Response to referees – Nature Cell Biology submission NCB-A52386-T

Reviewers' Comments:

Reviewer #1:

5 We thank the reviewer for their enthusiastic assessment and expert suggestions. We have addressed the reviewer's points as detailed below.

Remarks to the Author: This is yet another impressive body of work from the Schuh lab! The manuscript discovers in eggs and ovaries there is an abundance of long-lived proteins that appears to exceed that of other cell types. The measuring of protein lifetime and turnover in oocytes and ovaries, includes an analysis of the impact of ageing.

10 *In terms of the relevance of my review, I have considerable experience in oocyte cell biology, but proteomics remains far from my area of expertise, and I'm afraid I don't have time to master the details for the purpose of this review. Furthermore, a lot of the data is dependent on modelling that I am not able to critique in detail. It does however raise a few questions for me, which are elaborated below. The technical approaches, the analytical approaches, and modelling requires a truly multi-disciplinary review team to cover this one. I will focus on oocyte biology and conceptual matters relating to data interpretation. I am mightily impressed with the whole thing, so please do take*
15 *these comments as a constructive effort to make sure the data is as right as it can be and, to improve the life of the interested reader.*

20 *1. First, a comment on the presentation of the manuscript. It is somewhat challenging to review because the information in the text in the main body of the manuscript is limited, and the figures, extended data and supplementary data and methods are all extensive, and an important part of the story. It is a shame to squeeze such a manuscript into a tight word limit and I would prefer to see more of the data in the figures (moved from extended data) and more description of the modelling as part of the central story. Currently the Figures are largely representative summaries of bioinformatic analysis, while much of the data is 'extended'. Further, the complexity of the quantitative approaches, and any possible limitations, can only be appreciated by those brave enough to delve deeply into the methods section.*

25 We thank the reviewer for this very helpful comment. Following the reviewer's suggestion, we have now moved many extended data figure panels into the main figures, and increased the number of main figures from 5 to 8. In addition, we substantially expanded the modelling description in the main text (p. 4, lines 126-145).

30 *2. One question I have concerning the assay is that it assumes the presence of $^{13}\text{C}_6$ -Lys in a given protein originates from the time it was synthesised during the original pulse. Is it possible that on protein catabolism the $^{13}\text{C}_6$ -Lys re-enters the amino acid pool for utilisation in subsequent protein synthesis? I note this has been incorporated into the model to determine the potential impact on the %H (methods and Extended data 4C). The modelling as outlined in the methods is impressive, but there is little description of assumptions and*
35 *limitations that could influence the data. A circumspect description of the model including potential limitations is worth considering.*

40 We thank the reviewer for this helpful comment and the suggestion to include a more detailed description of the modeling and its potential limitations in the manuscript. As the reviewer suggests, the free pool of $^{13}\text{C}_6$ -Lys can get recycled, resulting in re-incorporation of $^{13}\text{C}_6$ -Lys into newly synthesized proteins during the chase period. This recycling of $^{13}\text{C}_6$ -Lys was incorporated into the modelling as the free $^{13}\text{C}_6$ -Lys pool. The

protein turnover model (former Extended Data Fig. 3c, now Fig. 2f) therefore accounts for the recycling of the $^{13}\text{C}_6$ -Lys during the chase period. In the revised manuscript, Fig. 2f illustrates the fraction of $^{13}\text{C}_6$ -Lys over all free Lys over the time course of the chase period. This fraction was derived from the analysis of 2Lys-peptides, which can either be composed of two $^{13}\text{C}_6$ -Lys, two $^{12}\text{C}_6$ -Lys or a mix of both, allowing for the inference of the free $^{13}\text{C}_6$ -Lys pool¹⁻⁴ (new Extended Data Fig. 3c-e, former Extended Data Fig. 3d-f). Please find a more detailed description of this point in our response to point 3 below. The protein turnover model thus allows for degradation of $^{13}\text{C}_6$ -Lys labelled proteins with rate k_{deg} , but also for new synthesis of $^{13}\text{C}_6$ -Lys labelled proteins with rate k_{on} . In the model, k_{on} is defined as a function of the fraction of $^{13}\text{C}_6$ -Lys over all free Lys in order to consider the possibility of $^{13}\text{C}_6$ -Lys re-incorporation into newly synthesized proteins.

45

50

Fig. 2f: (former **Extended Data Fig. 3c**) Graphical illustration of the employed protein turnover model to estimate $H_{1/2}$ values in ovaries.

We agree with the reviewer that the description of our modelling approach in the main text of the manuscript was too brief. We have therefore expanded the model description and figures to clarify these points (p. 4, lines 126-135). We now also discuss model limitations and assumptions (p. 4, lines 136-145):

55

Revised text:

“The turnover rates were determined by mathematical modeling (Fig. 2e-f), as described in detail in the Material and Methods section (Extended Data Fig. 3a). In brief, for each protein, we calculated the $^{12}\text{C}_6$ -Lys (L) to $^{13}\text{C}_6$ -Lys (H) ratio as $R = \ln(L/H + 1)$ at the different time points (Fig. 2g-h). These experimentally determined ratios were then compared to ratios obtained from a protein turnover model (Fig. 2f). The protein turnover model describes the abundance of $^{13}\text{C}_6$ -Lys-labeled proteins over the chase period, taking into account protein degradation (with rate k) and protein synthesis (with rate k_{on}) due to recycling of free (unincorporated) $^{13}\text{C}_6$ -Lys. The rate k_{on} is defined as a function of the free $^{13}\text{C}_6$ -Lys pool, such that the higher the remaining free $^{13}\text{C}_6$ -Lys, the higher the rate of $^{13}\text{C}_6$ -Lys protein synthesis rate due to $^{13}\text{C}_6$ -Lys recycling, and, hence, the slower the decay of $^{13}\text{C}_6$ -Lys labeled protein.

60

65

We determined the free $^{13}\text{C}_6$ -Lys and $^{12}\text{C}_6$ -Lys pools throughout ovarian development by analyzing the concentration of different species of uncleaved peptides containing two lysines^{3,4} (Extended Data Fig. 3c-e). 2Lys peptides can either be composed of two $^{13}\text{C}_6$ -Lys, two $^{12}\text{C}_6$ -Lys or a mix of both, allowing for the inference of the free $^{13}\text{C}_6$ -Lys pool^{3,4} (Extended Data Fig. 3c-e). As additional parameters, ovarian growth and changes

70

in individual protein concentrations (DIA MS data) were included in the modeling to account for changes in ovarian protein composition (Extended Data Fig. 3f; details in Materials & Methods). The resulting modelled turnover rates reflect the average behavior of all proteins in the ovary, and cannot differentiate if protein turnover differs for free protein and protein in complexes or in different cell types.”

75 “Lastly, we employed our protein-centric model to pulse-chase SILAC data derived from mouse liver, cartilage and skeletal muscle published by Rolfs et al. (2021)⁴ to estimate protein half-lives, and to compare them to the original published values (Extended Data Fig. 4e-g). For all three datasets we obtained good agreement of inferred half-lives with correlation coefficients of 0.85, 0.96 and 0.88 for liver, cartilage and skeletal muscle, respectively, increasing the confidence in our modelling approach.”

80

3. *In the same light, the identification of peptides with both ¹³C₆-Lys and ¹²C₆-Lys, demonstrates that there is overlap between the pulse and the chase, which the authors acknowledge and make an effort to deal with (described in detail in the methods, but only a cursory mention in the text). Oocytes may also be very effective at pooling and recycling amino acids and this could potentially explain some of the longevity. Somewhere in the data I expect there is an answer to this question, but I’m not sure if it is based on the actual measurements of ¹³C₆-Lys, the modelling, or both.*

85

We fully agree with the reviewer that amino acid recycling is crucial to be considered when determining protein longevity from in vivo pulse-chase experiments. Following the reviewer’s suggestions, we have now expanded the text on the importance of ¹³C₆-Lys re-incorporation into newly synthesized proteins, and explain in the revised manuscript, how this information is used to determine protein turnover rates in ovaries.

90

As mentioned above, we took advantage of 2Lys-peptides to determine the pools of free ¹³C₆-Lys and free ¹³²C₆-Lys inside ovaries, in line with previous publications^{1, 2}. In particular, we used modelling and Bayesian parameter inference to indirectly learn the free ¹³C₆-Lys and free ¹²C₆-Lys concentration over the chase period from the MS data. To this end, we searched the MS data for peptides that carry 2 Lysines (2-Lys peptides) and tracked their MS1 intensities over the chase period (Extended Data Fig. 3d). Here, we distinguish three possible states of 2Lys-peptides: (i) ¹²C₆-Lys-¹²C₆-Lys peptides, (ii) ¹²C₆-Lys-¹³C₆-Lys peptides and (iii) ¹³C₆-Lys-¹³C₆-Lys peptides. Furthermore, we set up a 2Lys-peptide model (Extended Data Fig. 3b) that describes the concentration of the three possible states of all 2-Lys peptides over time. With the 2-Lys peptide data and the 2Lys-peptide model at hand, we were able to estimate the fraction of free ¹³C₆-Lys over all free Lysines over the chase period (Extended Data Fig. 3c).

95

100

105 **Extended Data Fig. 3b-c:** (former **Extended Data Fig. 3d-e**) **(b)** Graphical illustration of the 2Lys-peptide model to determine the fraction of free $^{13}\text{C}_6\text{-Lys}$ during ovarian ageing. **(c)** Experimental data and model fits for the 2Lys-peptide model for the chase ($^{12}\text{C}_6\text{-Lys}$) from birth and from weaning experiments.

110 A similar approach to modelling the reincorporation of $^{13}\text{C}_6\text{-Lys}$ has been used in previous studies, including, Fornasiero *et al.* (2018)³, Kluever *et al.* (2022)⁵, and Rolfs *et al.* (2021)⁴. To verify our modelling approach, we applied our modelling framework to the published data of Rolfs *et al.* (2021)⁴ and compared the derived half-lives with the originally published half-lives (see also answer to point 4.2 below), which showed good agreement (correlation coefficients ranging between 0.85 and 0.96).

115 We are confident that the highly long-lived proteins in the oocyte or ovary are not due to a particularly high efficiency of amino acid recycling or $^{13}\text{C}_6\text{-Lys}$ retention, as this should lead to a shift in the longevity of all proteins. Instead, we observe that specific subsets of proteins are particularly long-lived (Supplementary Data 3). In addition, our analysis of oocytes (Figure 1) does not include any modeling, and yet, we observed 66 proteins with more than 50% $^{13}\text{C}_6\text{-Lys}$ (Extended Data Fig. 2e), and many other proteins with high $^{13}\text{C}_6\text{-Lys}$ fractions. Such a high number of proteins with high $^{13}\text{C}_6\text{-Lys}$ is unlikely to be due to recycling alone, as
120 the volume of the oocytes increases around 100-fold during the chase period (Extended Data Fig. 2b).

4. *The modelling used to measure protein longevity in oocytes raises a couple of questions about some aspects of the manuscript.*

125 (4.1) *One area of immediate concern is that a critical part of the model is accounting for the dilution of proteins as the oocyte/ ovary undergoes a dramatic increase in volume during the pulse phase. Yet the measurements of primordial oocyte diameter have been made using histological sections while that of fully grown oocytes seems to be from archived fixed samples/sections. The variability in the data seems huge (Extended Data Fig 2b). Fixation and histological processing will not provide an accurate measure of oocyte diameter. Given the formula for determining volume of a sphere, inaccuracies at this point are a potential source of compromise in the modelling. I would recommend the authors determine if the measurements of oocyte diameter reflect the physiological reality. If differences exist, please determine the impact this has on the analysis of half-life.*

130 We thank the reviewer for requesting clarification of this point. The volume measurements of oocytes shown in Extended Data Fig. 2b and referred to by the reviewer above were not used for modelling, but were only shown to illustrate to the readers that the oocytes grow a lot during the chase period. We agree
135 that the measurements cannot be fully accurate and have therefore now changed the wording in the main

text to “around 100 fold”. Importantly though, as mentioned above, these data were not used for modelling and therefore did not influence the %H values. The oocyte %H values in Fig. 1c represent the raw %H values (ratio of $^{13}\text{C}_6\text{-Lys} / ^{12}\text{C}_6\text{-Lys}$ intensity *100%) in fully grown oocytes following the chase from birth ($^{12}\text{C}_6\text{-Lys}$), without any modelling. We have now clarified this point in the revised manuscript (p. 3, lines 86-87).

140

In the ovary dataset, we computed half-lives based on the protein turnover model. This model considers the growth of the ovary to account for the dilution of $^{13}\text{C}_6\text{-Lys}$ -labelled proteins (Fig. 2e-f). It is important to note that we did not use experimentally measured ovary growth in the model training. Instead, we estimated the dilution factors, *i.e.*, the factors by which the ovary grows between time points, together with the remaining model parameters from the SILAC turnover data. While we agree with the reviewer that precise volumes of ovaries in 3D are difficult to determine, we additionally obtained BCA protein amount measurements that we expect to be well reliable (Extended Data Fig. 3f). We then compared the estimated dilution factors from the model to these experimentally determined ovary volume measurements (using microscopy) and total protein amount measurements (using a BCA assay) and found good agreement (Extended Data Fig. 3f). All experimental measurements were in good agreement with our estimated dilution factors.

145

150

155

Extended Data Fig. 3f: Modeling protein turnover in the ovary. (f) Change of total protein amount in the ovary over mouse age. Total protein amount was determined from BCA measurements and compared to changes in the volume of the ovary. Boxplots indicate the estimated fold change in total protein amount. Even though the volume measurements of the ovary were not considered during protein turnover model fitting, the estimated fold changes are in good agreement with experimentally measured total protein amount fold changes.

160

(4.2) How valid it is to make comparisons of protein half-life in oocytes and ovaries with that obtained for other tissues in prior studies (Extended data 7). Presumably the complexity of the modelling in oocytes and ovaries means that the approaches used are very different. It would be ideal if the authors could compare other tissues analysed by the same approach as used here.

165

We thank the reviewer for this excellent suggestion. In the revised manuscript we have now applied our modelling framework to three datasets published by Rolfs *et al.* (2021)⁴. Rolfs *et al.* (2021)⁴ carried out pulse-chase experiments with different tissues. The pulse and chase periods were different, but the overall experimental design was related. Using our modelling approach, we determined protein half-lives for liver, cartilage and skeletal muscle and compared them to the original published half-lives (Extended Data Fig. 4e-g). For all three datasets we obtained good agreement of inferred half-lives with correlation coefficients of 0.85, 0.96 and 0.88 for liver, cartilage and skeletal muscle, respectively. We did not observe extremely

170

long-lived proteins in any of these datasets, while we do with the same modelling approach in the ovary dataset. These data are now shown in new Extended Data Fig. 4e-g in the revised manuscript.

175

180

Extended Data Fig. 4e-g: Comparison of the $H_{1/2}$ values estimated from different models. Comparison of previously published $H_{1/2}$ values (Rolfs et al. (2001)⁴) and $H_{1/2}$ values calculated for the same published data sets using the protein turn-over model developed in this manuscript for ovaries. Shown are comparisons for liver (e), cartilage (f) and skeletal muscles (g). Correlation coefficients (C) and p-values (p) are displayed.

185

Furthermore, we would like to highlight Fig. 3d (former Extended Data Fig. 7b). This heatmap, consisting of three parts, shows the experimentally determined percentage of $^{13}\text{C}_6$ -Lys labelled proteins over time (middle and right panels). These percentages were computed solely based on the detected fraction of $^{13}\text{C}_6$ -Lys-labelled proteins' intensity over total protein intensity and are not affected by any modelling approach. In this heatmap, it is clearly visible that a substantial subset of proteins is detected as $^{13}\text{C}_6$ -Lys-labelled even after 65 weeks of chase, indicating the presence of very long-lived proteins in the ovary. In addition, we observed many extremely long-lived proteins in oocytes where we only computed raw values of % $^{13}\text{C}_6$ -Lys without using any mathematical modelling (details in reply to point 4 above).

190

195 **Figure 3d:** (former Extended Data Figure 7b) (d) Cluster analysis of inferred $^{13}\text{C}_6\text{-Lys}$ levels in the proteins of the aging ovaries. The dendrogram on the left corresponds to the clustering of inferred percentage $^{13}\text{C}_6\text{-Lys}$. The leftmost bar shows the medians of inferred proteins half-lives, coloring on \log_{10} -scale. The second and third bars indicate the latest time point at which the $^{13}\text{C}_6\text{-Lys}$ pulse was detected in the data corresponding to chase from birth and weaning. The rightmost bar labels the three identified protein longevity clusters corresponding to proteins with high, intermediate and low amounts of inferred $^{13}\text{C}_6\text{-Lys}$ content. The leftmost heatmap shows the inferred percentage $^{13}\text{C}_6\text{-Lys}$; time points from 6 weeks onwards were used for clustering. Central and rightmost heatmaps show experimental data of the chase $^{13}\text{C}_6\text{-Lys}$ from birth and weaning, respectively.

205 (4.3) *The vast majority of proteins in different tissues have the same profile as shown in Extended Data 7). The key finding in this data is, if I understand it correctly, all about the right-hand end of the graph where there is a higher density of label present than in other tissues right out to 1000 days – well over a mouse lifetime! Now this may well be right, but I would want to try to prove myself wrong to be sure of this. Testing different parameters of the model to see if any are unexpectedly able to modulate longevity might be worth testing. Isn't it a little concerning that the density does not continue to gradually decline between 100 and 1000 days – it actually seems to increase a little, before taking a little dip after 1000 days?*

215 To challenge our modelling approach, we re-computed protein half-lives in mouse liver, cartilage and skeletal muscle based on the published SILAC pulse-chase data by Rolfs *et al.* (2021)⁴, which further increased the confidence in our modelling approach and the inferred protein half-lives (please see also answer to point 4.2). It is important to note that the distribution of inferred protein half-lives is shown on \log_{10} -scale. We provide the same distribution in linear scale in this report (Reviewer Fig. 1). The half-lives indeed gradually decline between 100 and 1000 days (right graph). Furthermore, it is important to keep in mind that very long half-lives are estimated with large confidence intervals (Extended Data Fig. 4d).

220 **Reviewer Fig. 1:** Distribution of inferred $H_{1/2}$ (in days) in mouse ovaries plotted on linear x-axis.

As also suggested by the reviewer, we modified our model approach in various ways and compared the resulting half-lives to the full modelling approach (Extended Data Fig. 4a-d). Specifically, we compared our protein-centric turnover model including re-incorporation of free $^{13}\text{C}_6$ -Lys into newly synthesised proteins (Fig. 2f) to (i) a peptide-centric 2Lys-peptide model (Extended Data Fig. 4a) and (ii) a protein-centric model not including re-incorporation of free $^{13}\text{C}_6$ -Lys into newly synthesized proteins (Fig. 2f with $k_{on}=0$) using either all chase time points (Extended Data Fig. 4b), or only chase time points larger than 3 weeks or 6 weeks, respectively (Extended Data Fig 4c-d). The 2Lys-peptide approach takes advantage of the fact that very abundant proteins sometimes have enough 2-Lys peptides resulting from missed tryptic cleavages during sample preparation, so that only peptides with two $^{13}\text{C}_6$ -Lys can be taken into consideration for modelling, which are highly unlikely to be generated by reincorporation of $^{13}\text{C}_6$ -Lys, and the free $^{13}\text{C}_6$ -Lys pool can then be neglected. The calculated $H_{1/2}$ values calculated in this way were well consistent with our protein-centric turnover model. In addition, we also calculated the $H_{1/2}$ values from late time points only (larger than 3 weeks or 6 weeks). Growth has substantially slowed down at this stage and the free $^{13}\text{C}_6$ -Lys pool is strongly depleted. In further support of our model, modelling of these late time points only (allowing for exclusion of the free $^{13}\text{C}_6$ -Lys pool) gave very similar results to our full protein-centric model using all collected data and time points (Extended Data Fig. 4c,d).

240 These control analyses provide further validation for our modelling approach and the identification of extremely long-lived proteins in the ovary.

245 **Extended Data Fig. 4: Comparison of the H1/2 values estimated from different models.** (a-d) Comparison of
 250 estimated H_{1/2} values resulting from the protein turnover model considering free ¹³C₆ pool with 2Lys-peptide based
 model (a) and the ‘classical’ protein turnover model not allowing reincorporation of ¹³C₆-Lys into newly synthesized
 proteins considering all chase time points (b), or only chase time points larger than 3 weeks (c) or 6 weeks (d),
 respectively. Dots indicate medians, grey lines indicate confidence ranges. All proteins with estimated H_{1/2} values
 <100 days are indicated as orange dots. Correlation coefficients (C) and p-values (p) are displayed.

255 *(4.4) Conceptually it makes sense that oocytes may have reduced protein turnover with more effort going
 into creating a stable environment to minimise the effects of ageing in the germ plasm. However, it is less
 obvious why somatic cells in the ovary, including stromal cells, have a similarly high proportion of long-lived
 proteins. Again it raises the question for me if this represents a fundamental difference in the modelling
 approach in this study compared to the previously published examples used as comparators. Would other
 tissues show similar long lived proteins if the model in this manuscript was applied to them?*

260 We thank the reviewer for this comment and excellent suggestion. The analysis of 13C/12C protein ratios
 in specific cell types is independent of modelling (Fig. 5d-k). Instead, we determined the 13C/12C ratio of
 individual cell types in the ovary using nanoSIMS imaging of ovaries from 4-week and 8-week-old mice. In a
 nanoSIMS analysis, the local ratio of 13C/12C can be directly measured.

265 Furthermore, following the reviewer’s suggestion, in the revised manuscript we applied our modelling
 framework to three datasets published by Rolfs *et al.* (2021)⁴. In this way, we determined protein half-lives
 for the liver, cartilage and skeletal muscle, and compared them to the originally published half-lives
 (Extended Data Fig. 4e-g). For all three datasets, we obtained good agreement of the inferred half-lives with
 correlation coefficients of 0.85, 0.96 and 0.88 for liver, cartilage and skeletal muscle, respectively. In none
 of these datasets did we observe extremely long-lived proteins, as we do with the same modelling approach
 in the ovary dataset (please also see answer to point 4.2).

270 5. *In Extended data 7d, is it possible to provide an indication as to how the log of MS1 relates to the level of protein that would be present in the ovary. For most proteins it appears there is about 1000-fold reduction (from 10 to 7 on the log scale) between birth and around 30 weeks, with some variability between proteins. Is an MS1 of 7 effectively when there is no 13C6-Lys remaining?*

While on this figure, what is happening to HDAC1. It seems to have a major kick in 13C6-Lys in ovaries between 1 and 6 weeks, despite the pulse stopping at birth?

275 We thank the reviewer for requesting clarification of these points. MS1 protein intensity is proportional to protein concentration, and hence, can be compared over time. However, protein concentration is strongly affected by the dilution of ¹³C₆-Lys-labelled proteins due to ovary growth. In the first 30 weeks, the ovary grows approximately 200- to 300-fold. This means, the MS1 protein intensity of ¹³C₆-Lys-labelled proteins is diluted 200-300 times, in addition to the decay of ¹³C₆-Lys-labelled proteins. Therefore, a decay of the MS1 signal from 10¹⁰ to 10⁷, *i.e.*, a 1000-fold decrease of signal, would correspond to a ~5-fold decrease of the amount of ¹³C₆-Lys-labelled protein, because of the 200-fold dilution due to ovarian growth. However, this rough estimation is not exact and ignores further factors, such as the potential re-incorporation of free ¹³C₆-Lys into newly synthesized proteins. For these reasons, it is not possible to directly “read” the change in protein amount from the MS1 signal intensities, but only after incorporating these aspects into the model.

285 Concerning HDAC1: in both the short and long pulse data sets, the ¹³C₆-Lys signal of HDAC decreases initially but then stays largely constant over the coming weeks. Two individual values in the short pulse data set (at 6 and 12 weeks) appear higher than the other baseline values. However, it is important to note that MS data derived from *in vivo* studies are noisy. Indeed, at the 6- and 12-week time points where a high ¹³C₆-Lys signal is present in one biological replicate, the ¹³C₆-Lys signal was not detected in the other two biological replicates. Therefore, this is unlikely to represent a real increase in ¹³C₆-Lys, but is more likely to represent noise. To avoid over-interpretation and artifacts in modelling, we accounted for noisy data by only considering time points for which signals were detected in at least two out of three biological replicates (please see also page 1368 in Supplementary Data 2 - second row shows only data points considered for modelling).

Other points:

300 *Ln 118-122: The pulse labelling till weaning will label oocytes undergoing the first wave of growth from the start to completion. This wave typically starts just after birth and can be super ovulated by administering PMSG/hCG just after weaning (21-23 days). Thus, the following statement doesn't seem strictly correct, and the significance of it is not explained. Clarification would help. "This approach labels all proteins synthesized before birth, as well as those 121 synthesized up to when the first wave of resting primordial follicles are recruited into the growing follicle pool 11,13 122 ." It is actually labelling protein synthesis to the end of the first wave of oocyte/follicle growth.*

305 We thank the reviewer for highlighting this important point. We have implemented this change in the new version of the manuscript (p. 3, lines 117-118).

310 *Ln 126-131 – It is almost impossible to read this with any continuity as there is a need to continually flick between multiple figures. As mentioned above - Is there a model whereby the essential data needed to evaluate a particular point is presented in single figure? It's unnecessarily obfuscating for the reader.*

We are grateful to the reviewer for this helpful comment. We have reformatted the text and figures of the manuscript and have now moved related extended data figure panels into the main figures. We hope that it is now easier to follow the figures and texts.

315 *Extended data Fig 2 (Legend) and methods: This seems an arbitrary definition of a fully labelled protein? What is happening in the other two replicates where the C6-Lys is not detected and what is happening with the biological replicates. There is a little more elaboration in Methods, but please provide a reason for why you chose 2/4 replicates. "A protein was defined as fully labelled (%H value of 100%) if it was detected as a 13 C6-Lys labelled protein in at least two out of four technical replicates in at least one biological replicate and there is no signal for 12 1496 C6-Lys*
320 *labeled protein in any of the 1497 four technical replicates of both biological replicates detected."*

We thank the reviewer for this very helpful comment. In the revised manuscript, we have included a rationale for this definition (p. 16, lines 670-681; p. 46, lines 1660-1664). MS data often have missing values. Detecting very low abundant ¹³C₆-Lys-labelled proteins is challenging and we would not expect the detection of all proteins across all technical replicates. In MS data analysis, the missing value problem is compensated to some extent by analyzing
325 multiple technical replicates. These technical replicates are then aggregated together, regardless of how many technical replicates a signal was detected in. In order to identify proteins that are still ¹³C₆-Lys labelled after the long chase period, we decided to be stricter in this aspect to avoid identifying false positives due to noise and missing values. Therefore, we considered a signal as robust only if it was detected in at least two out of four technical replicates. On the other hand, to define the absence of a signal, we again decided to be stricter and required the
330 absence of the signal in all technical replicates.

Ln 387-390: Use of M2 with no BSA – was any other supplement used to prevent the oocytes from sticking and ease handling?

335 All oocyte handling for the MS experiments was performed in BSA-free M2 medium. For this sample collection, pipettes that are long and slightly wider than the oocyte were used. The oocytes were kept in the lower extended part of the pipets, which helps to minimize sticking to the glass surface. Oocytes were first washed in a 4-well plate containing BSA-free M2 media and transferred to a 1.5 mL tube. After washing, a total of 10-20 oocytes were transferred at a time to a 1.5 mL tube to facilitate handling. Once the correct number of oocytes were in the tube, excess liquid was removed so that the oocytes were just covered with liquid. The oocyte samples were then snap
340 frozen in liquid nitrogen and stored at -80°C until further processing. It is important to highlight that no oil should be used during the culture, as even trace amounts can interfere with mass spectrometry measurements.

Reviewer #2:

Remarks to the Author:

345 We thank the reviewer for their enthusiastic assessment and expert suggestions. We have addressed the reviewer's points as detailed below.

350 *Harasimov et al. used large-scale mass spectrometry to study the stability of the proteome in the mammalian ovary. Oocytes are produced at birth and must be maintained undamaged to ensure 100% female reproductive capacity. In addition to genome stability, the proteome must be well protected to ensure the propagation of germ cells over several generations. The cellular proteostasis network supports efficient protein synthesis, protein folding and protein degradation. However, it remained unclear how the oocytespecific proteome is protected. The authors performed large-scale proteomics combined with pulsechase/ time course experiments in female mice and identified a surprisingly large number of proteins that are extremely long-lived, with a 10-fold increase compared to other post-mitotic tissues! These highly stable oocyte/ovarian proteins are functionally diverse and are associated with mitochondrial function, metabolism, cytoskeleton, chromatin organisation and also proteostasis. However, the same*
355 *proteins show a rapid decline in the ageing ovary, suggesting that their stability maintains oocytes throughout the reproductive life of the female.*

360 *The manuscript is very well written and reflects a tour de force in addressing proteome remodelling in the mammalian ovary associated with oocyte maintenance and age-related decline in reproduction. Although the work is mainly based on comprehensive proteomic approaches, it provides an amazingly detailed catalogue of highly stable proteins that change with age in the ovary. As such, the work is fundamental to future studies investigating the mechanisms that support proteome stability, which I believe is beyond the scope of the current manuscript.*

365 *Questions: One question the authors could address is the role of the most stable proteins in proteostasis. Can overexpression in tissue culture or in an organismic model (e.g. C. elegans) promote resistance to proteotoxic stress and/or longevity? How resistant are ovarian cells to stress conditions? The authors could test proteasomal activity/protein aggregation at young and old stages in oocytes. What happens to stable proteins during ageing if autophagy or proteasomal degradation is blocked, if this is experimentally possible.*

370 We thank the reviewer for these excellent suggestions. Following the reviewer's suggestion, we quantified proteasomal activity in young and aged oocytes with a live cell imaging-based reporter assay⁶. We found that oocytes maintain full proteasomal activity throughout the reproductive lifespan of mice, providing further evidence that proteostasis is optimized in the germline. These new data are now shown in new Fig. 6 of the revised manuscript. The corresponding results are described on p. 7, lines 277-286.

375 **Fig. 6: Protein aggregation does not increase in aged oocytes.** (a) Representative immunofluorescence images of
 fully grown mouse oocytes from 9-week and 65-week old mice stained with the ProteoStat aggresome dye. Magenta,
 aggresome (ProteoStat); Cyan, DNA (Hoechst). (b) Dot plot showing number of ProteoStat-positive structures in
 oocytes as shown in (a). (c) Dot plot showing total intensity of ProteoStat-positive structures in oocytes as shown in
 (a). (d) Representative immunofluorescence images of brain slices from 9-week-old and 65-week-old mice
 stained with the ProteoStat aggresome dye. Magenta, aggresome (ProteoStat); Cyan, DNA (Hoechst). (e) Dot plot showing
 total intensity of ProteoStat-positive structures in brain slices as shown in (d). (f) Dot plot showing total intensity of
 ProteoStat-positive structures in brain slices as shown in (d). (g) Representative immunofluorescence images of early
 follicles from 9-week-old and 65-week-old mice stained with the ProteoStat aggresome dye. Magenta, aggresome
 (ProteoStat); Cyan, DNA (Hoechst). No obvious aggresome accumulation was detected in either age group. All data
 from two independent experiments. Number of analyzed oocytes and brain areas are in brackets. Data are shown
 as mean \pm standard deviation (SD). P values were calculated using unpaired two-tailed Student's t test. N.S., not
 significant; **** $P \leq 0.0001$. Scale bars, 10 μ m.

390 Following another excellent suggestion from the reviewer, we quantified protein aggregation in young and old
 oocytes. Using the Proteostat dye for aggresomes, we found that protein aggregation was low in young and old
 oocytes, with no detectable increase in old oocytes. In contrast, brain sections from the same young and old mice
 showed a marked increase in protein aggregation with age, consistent with previous studies. These data provide
 further evidence that proteostasis is optimized in the germline, allowing for the maintenance of germline cells over
 long periods of time. These new data are now shown in new Fig. 7 of the revised manuscript. The corresponding
 395 results are described on p. 7, 287-302.

Fig. 7: Proteasomal activity does not decay in aged oocytes. (a) Time-lapse images of mouse oocytes from 9-week-old mice expressing Ub(G76V)-3x mClover3-T2A-mScarlet in the presence of DMSO or 10 μ M MG-132. Time is given as hours after DMSO or MG-132 treatment. (b) Quantification of the mean fluorescence intensity of mClover3 in oocytes as shown in (a). (c) Quantification of the fluorescence intensity ratio of mClover3 to mScarlet in oocytes in (a). (d) Time-lapse images of mouse oocytes from 9-week and 65-week old mice expressing Ub(G76V)-3x mClover3-T2A-mScarlet. Time is given as hours after injection of the reporter mRNA. (e) Quantification of the fluorescence intensity ratio of mClover3 to mScarlet in oocytes in (d). (f) Schematic diagram of the experiment shown in (g). Ub(G76V)-3x mClover3 and mScarlet were expressed for 5 h in the presence of MG-132, which blocks the degradation of Ub(G76V)-3x mClover3. MG-132 was then washed out and oocytes were imaged in the presence of the translation inhibitor cycloheximide (CHX), which blocks the synthesis of new proteins. (g) Time-lapse images of mouse oocytes from 9-week-old and 65-week-old mouse oocytes expressing Ub(G76V)-3x mClover3-T2A-mScarlet.

400

405

Experiment was performed as shown in (f). Time is given as hours after MG-132 wash-out and CHX wash-in. **(h)** Line graph showing normalized fluorescence intensity ratio of mClover3 to mScarlet in oocytes in (g).

410

We agree that our data highlight potential candidate proteins that could promote resistance to proteotoxic stress and/or longevity, and that in-depth studies of the identified proteins and their potential protective function will be an exciting direction for future research. We also agree that a study of stress resistance in ovarian cells compared to other tissue types is an interesting direction of research. We considered proteasomal inhibition assays followed by quantification of protein aggregation/ubiquitination levels after treatment. However, we were concerned that differences in the efficiency of proteasomal inhibition by MG-132 in different cell types/tissues, as described in previous studies ^{7, 8}, would confound the interpretability of these data and therefore instead focused on the reviewer's alternative experimental suggestions. We feel that these new results further increase the impact of our study and are confident that they will be of interest to a wide range of readers.

415

420 **Reviewer #3:**

Remarks to the Author:

We thank the reviewer for their positive assessment and expert suggestions. We addressed the reviewer's points as detailed below.

425 *The authors present a deep proteomic analysis of proteome dynamics in murine oocytes and ovaries. Using 13C pulse chase experiments, the authors were able to monitor the loss of 13C incorporation in oocytes proteomes across multiple time ranges. The result is a series of datasets consisting of proteins from diverse biological processes (OXPHOS to cohesin). The study is clearly a tour de force of proteomics analyses, with 4948 oocytes analyzed and presented in just the first figure. Moreover, the integration of multiple techniques aids in the potential for use within this study and beyond. All that being said, I had a few comments and questions that arose in reading this work that I think should be addressed prior to publication. My concern is that with all of these data the main output seems to be a list of proteins and GO/GSEA terms (autophagy, ER lumen, inflammation from Fornasiero 2018, Kleuver 2022, and Keele 2023), that could be relationally inferred from other mouse aging studies. Thus, there is sometimes a tenuous connection between the wealth of data and a future looking utility of this work.*

430 *Major Comments*

435 1. *Much of the most interesting data is relegated to the extended data figures. Two panels (Fig 1c and Fig 3d) are disguised tables. While I appreciated the graphics, these held limited value for me as they are effectively lists of >40 proteins. The values without context (i.e., baseline) comparisons are not especially useful, too. What %H is interesting and why? This context seems in part to be buried in Fig S7a. Simple things to add this context that I can imagine: is the %H for the proteasome higher than say chaperones like HSP90? What about the CCT complex that is strongly enriched in Fig 1b?*

440 We thank the reviewer for these helpful comments. We have now restructured the text and figures in the revised manuscript to highlight more of the important data in the main figures. Following the reviewer's excellent suggestion, we have now also analysed the distribution of $H_{1/2}$ values of various protein complexes, such as the proteasome, ribosomes and spliceosomes. We now show these data in new Extended Data Fig. 5b. We understand the reviewer's point that the data in Fig. 1c and Fig. 3d are also included in the tables and can be searched there. However, we wanted to make interesting long-lived proteins that are related to important biological processes in oocytes and ovaries easily accessible to scientists working in these areas, and have therefore also highlighted subsets of the most relevant proteins in the main figures. We believe this will help biologists engage with the data and inspire them to explore the full dataset.

445

450

Extended Data Fig. 5b: Comparison of the $H_{1/2}$ values of the different proteins in the ovary with their corresponding $H_{1/2}$ values in other organs, and examples of long-lived proteins in the ovary. (b) Long lived members of protein complexes. Shown are protein complexes with long-lived proteins as violin plots. Dots indicate individual proteins. Complexes are sorted by median complex $H_{1/2}$ values.

455

2. *As in work that the authors cite (namely Kluever et al 2022, Fig 1D), it would be useful to show the raw data to enable interpretation (and consistent with the authors previous work in Fornasiero 2018 and Kleuver 2022). There are also interesting suggestions from this data, such as why are proteasome subunits in oocytes/ovaries extremely long lived but short lived in Kluever et al data? Are other proteasome subunits in the intermediate half-life cluster? a. Similarly, different subunits of Complex I in Oxphos have distinct temporal profiles (e.g., NDUFB8 vs. NDUFS8), why is this? Are these from structurally distinct subcomplexes?*

460

We thank the reviewer for these excellent comments and suggestions. In the revised manuscript, we now show half-lives in ovaries in comparison to other mouse tissues, including young and aged brain, heart, liver, cartilage and two types of muscles derived from Fornasiero *et al.* (2018)³, Kluever *et al.* (2022)⁵, and Rolfs *et al.* (2021)⁴, including proteasomes, chaperones, mitochondrial proteins and many others (Supplementary Data 3 page 1-2, Reviewer Fig. 2). The $H_{1/2}$ values for different subunits of respiratory chain complexes are well consistent (Supplementary Data 3 page 3). Interestingly, the respiratory chain complexes in the ovary

465

470 have lifetimes resembling those in tissues such as liver and heart, but shorter than in brain tissues. In
contrast, a wide range of proteins has a much higher lifetime than in brain and other tissues (Supplementary
Data 3). It will be an exciting direction for future research to identify the underlying regulatory mechanisms
and biological functions of the high longevity of subsets of proteins. In the discussion, we now also highlight
475 the exciting possibility that subsets of proteins might be long-lived in the ovary and oocytes, because of
storage of precursors of cellular machineries in dormant oocytes that are only required after they resume
growth or complete meiosis (page 10, lines 411-414). This might explain the heterogeneity in longevity for
some of the proteins that are part of the same cellular machinery.

enriched terms

Reviewer Fig. 2: Comparison of half-lives across various mouse tissues. Shown are all gene sets, which are over-represented in the high protein longevity cluster shown in Fig. 4a (former Fig. 3c). (part of Supplementary Data 3).

480

Reviewer Fig. 2 (continued): Comparison of half-lives across various mouse tissues. Shown are all gene sets, which are over-represented in high protein longevity cluster shown in Fig. 4a (former Fig. 3c). (part of Supplementary Data 3)

485 **Reviewer Fig. 2 (continued):** Comparison of half-lives across various mouse tissues. Shown are proteins that belong to the different subunits of respiratory chain complexes. (part of Supplementary Data 3)

3. From the intro statements like this I expected the data to be related back to the “whys” of how these proteome changes explained the principles of female germline maintenance: “Our data suggest that mammals use these principles for maintaining the female germline over long periods of time.” I kept waiting to read how the ovary and oocyte protein and protein complex changes were different than other tissues.

490

We thank the reviewer for this helpful comment. In the revised manuscript, we have now included a direct comparison of protein half-lives determined in various studies³⁻⁵ (Supplementary Data 3) and performed additional experiments to shed light on potential mechanisms of female germline maintenance. For a direct comparison to other studies, we have: (i) modified former Extended Data Fig. 7a (now Extended Data Fig. 5a) and (ii) show the half-lives of proteins related to enriched terms in the ovary across various mouse tissues, including young and aged brain, heart, liver, cartilage and muscles (Supplementary Data 3).

495

In addition, we also tested experimentally whether oocytes are better capable of proteostasis over long time periods than neurons. In particular, we quantified proteasomal activity in young and aged oocytes with a live cell imaging-based reporter assay⁶. We found that oocytes maintain full proteasomal activity throughout the reproductive lifespan of mice, providing further evidence that proteostasis is optimized in the germline. These new data are now shown in Fig. 6 of the revised manuscript. The corresponding results are described on p. 7, lines 277-286.

500

505

Fig. 6: Protein aggregation does not increase in aged oocytes. (a) Representative immunofluorescence images of fully grown mouse oocytes from 9-week-old and 65-week old mice stained with the ProteoStat aggresome dye. Magenta, aggresome (ProteoStat); Cyan, DNA (Hoechst). (b) Dot plot showing number of ProteoStat-positive structures in oocytes as shown in (a). (c) Dot plot showing total intensity of ProteoStat-positive structures in oocytes

510 as shown in (a). **(d)** Representative immunofluorescence images of brain slices from 9-week-old and 65-week old
mice stained with the ProteoStat aggresome dye. Magenta, aggresome (ProteoStat); Cyan, DNA (Hoechst). **(e)** Dot
plot showing total intensity of ProteoStat-positive structures in brain slices as shown in (d). **(f)** Dot plot showing total
intensity of ProteoStat-positive structures in brain slices as shown in (d). **(g)** Representative immunofluorescence
515 images of early follicles from 9-week-old and 65-week-old mice stained with the ProteoStat aggresome dye.
Magenta, aggresome (ProteoStat); Cyan, DNA (Hoechst). No obvious aggresome accumulation was detected in
either age group. All data from two independent experiments. Number of analyzed oocytes and brain areas are in
brackets. Data are shown as mean \pm standard deviation (SD). P values were calculated using unpaired two-tailed
Student's t test. N.S., not significant; **** $P \leq 0.0001$. Scale bars, 10 μm .

520 In addition, we quantified protein aggregation in young and old oocytes. Using an aggresome dye, we found
that protein aggregation was low in young and old oocytes, with no detectable increase in old oocytes. In
contrast, brain sections from the same young and old mice showed a marked increase in protein
aggregation with age, consistent with previous studies. These data provide further evidence that
525 proteostasis is optimized in the germline, allowing for the maintenance of germline cells over long periods
of time. These new data are now shown in Fig. 7 of the revised manuscript. The corresponding results are
described on p. 7, lines 287-302.

Fig. 7: Proteasomal activity does not decay in aged oocytes. (a) Time-lapse images of mouse oocytes from 9-week-old mice expressing Ub(G76V)-3x mClover3-T2A-mScarlet in the presence of DMSO or 10 μ M MG-132. Time is given as hours after DMSO or MG-132 treatment. (b) Quantification of the mean fluorescence intensity of mClover3 in oocytes as shown in (a). (c) Quantification of the fluorescence intensity ratio of mClover3 to mScarlet in oocytes in (a). (d) Time-lapse images of mouse oocytes from 9-week and 65-week old mice expressing Ub(G76V)-3x mClover3-T2A-mScarlet. Time is given as hours after injection of the reporter mRNA. (e) Quantification of the fluorescence intensity ratio of mClover3 to mScarlet in oocytes in (d). (f) Schematic diagram of the experiment shown in (g). Ub(G76V)-3x mClover3 and mScarlet were expressed for 5 h in the presence of MG-132, which blocks the degradation of Ub(G76V)-3x mClover3. MG-132 was then washed out and oocytes were imaged in the presence of the translation inhibitor cycloheximide (CHX), which blocks the synthesis of new proteins. (g) Time-lapse images of mouse oocytes from 9-week-old and 65-week old mouse oocytes expressing Ub(G76V)-3x mClover3-T2A-mScarlet.

540 Experiment was performed as shown in (f). Time is given as hours after MG-132 wash-out and CHX wash-in. **(h)** Line graph showing normalized fluorescence intensity ratio of mClover3 to mScarlet in oocytes in (g).

545 4. *Nuclear proteome H_{1/2} is lower than the proteome overall: “nuclear proteins had shorter H_{1/2} values (Fig. 2g)”. Can the authors speculate on why this might be? What proteins in the nucleus are driving this. The effect sizes in Fig 2g are all pretty small, this would suggest that perhaps 1-2 complexes are driving this, how to these relate to the proteins?*

550 We thank the reviewer for highlighting this interesting point. Following the reviewer’s comment, we analyzed the nuclear and mitochondrial proteins in more detail, focusing specifically on particularly short- and long-lived proteins in the nucleus and mitochondria, respectively (Reviewer Fig. 3). We do indeed find a larger pool of shorter-lived proteins in the nucleus compared to mitochondria. These short-lived nuclear proteins are involved in cell cycle progression, DNA replication and derive mainly from the nucleoplasm. Interestingly, some MCM complex proteins (DNA helicase involved in DNA replication) are among the most short-lived proteins in the nucleus, while other MCM proteins are among the most long-lived proteins in the nucleus, respectively. These data are now illustrated in Supplementary Data 3 page 1-3 (Reviewer Fig. 2) in the revised manuscript. Furthermore, we illustrated a selection of protein complexes in Extended Data Fig. 5b (please see also answer to point 1).

555

560 **Reviewer Fig. 3:** Rank plot of determined H_{1/2} for all proteins (grey), mitochondrion (red) and nucleus (blue). The 10 most short-lived proteins in the nucleus and the most long-lived proteins in the mitochondria are labelled.

5. *“hence seems possible that low mitochondrial activity and low protein turnover are regulatory and functionally coupled in mammalian oocytes.” Was this actually compared within the data? Is there a significant correlation between mitochondrial energetic proteins and protein chaperones/proteasome/etc?*

565 We thank the reviewer for highlighting this point. We wanted to express with this sentence that the previously established low mitochondrial activity in oocytes⁹ may be regulated by the same signalling pathways that are responsible for the low protein turnover in oocytes, and may therefore be regulatory and functionally coupled. In support of this point, an analysis of our data revealed that the levels of mTOR signalling-related proteins indicate low activity of mTOR signalling in young ovaries and high activity of mTOR signalling in aged ovaries (Extended Data Fig. 7c). Low mTOR signalling leads to low mitochondrial activity and low protein turnover¹⁰⁻¹², and could therefore be a regulatory mechanism that leads to low mitochondrial activity and low protein turnover in the young ovary. We now discuss this exciting possibility in the revised version of our manuscript (p. 10, lines 402-406)

575 **Extended Data Fig. 7c:** Normalized DIA-MS intensity data showing the abundance changes over time of proteins in the mTOR signaling pathway.

6. *From my interpretation, Fig. S10d seems to represent a logical fallacy: the clustering of proteins based on 13C results in clusters with different levels of 13C.*

580 We thank the reviewer for requesting clarification of this point. Former Fig. S10d (now Extended Data Fig. 7a) shows what percentage of proteins assigned to the low, intermediate, and high ¹³C₆-Lys cluster (describing protein longevity determined via SILAC pulse-chase DDA MS measurements) belongs to each of the protein abundance clusters (describing the change in protein concentration determined via label-free DIA MS measurements). To avoid any potential confusion we have now renamed the “¹³C₆-Lys protein clusters” into “Protein longevity clusters”. We now refer to these different classifications as either “protein longevity clusters” or “protein abundance clusters” throughout.

585

7. *In Fig 4a., are the white (~35%) and black (~65%) bars indicating that all proteins in the high 13C cluster are all enriched or unenriched in at least one specific cell type? Fig 4d and 4h don't appear to have a legend for the coloring. The stromal cells in Fig 4e also appear to have a bimodal distribution?*

590 We thank the reviewer for requesting clarification of these points. These figures have become Fig. 5 in the revised version of the manuscript. In Fig. 5a, we wanted to identify in which specific cell types the long-lived proteins (high protein longevity cluster) are expressed. For this purpose, we performed single-cell RNA sequencing on ovaries from postnatal day 2 mice. We analyzed which of the proteins are >2-fold enriched

595 in a cell type, and which ones are less than 2-fold enriched in cell types. Around 30% of the proteins are more than 2-fold enriched in a single cell type. The remaining proteins are more broadly expressed (less than 2-fold enrichment in a specific cell type).

We thank the reviewer for highlighting the missing legend for the coloring. We have now added the look up table as a bar in the panel and indicated the specific boundary values, and explain these values in the legend (p. 41, lines 1571-1576).

600 The bimodal distribution of the stromal cells is indeed interesting. The stromal cells of the ovary have supportive roles in the tissue, and take over different functions¹³. The term “stromal cells” is used to refer to cells with different functionalities in the stroma of the ovary. It is possible that subsets of stromal cells differ in their longevity, which could explain the bimodal distribution.

605 8. *How do the authors reconcile complex phenotypes, such as the drop and then increase in ¹³C HDAC1 intensity in Figure S7d? Further, do other epigenetic regulators such as HDAC1 and DNMT3A have longer lifetimes than the average proteins? Even if simply in the intermediate categories, this would be striking.*

610 We thank the reviewer for highlighting this point. In both the short and long pulse data sets, the ¹³C₆-Lys signal of HDAC1 decreases initially but then stays largely constant over the coming weeks. Two individual values in the short pulse data set (at 6 and 12 weeks) appear higher than the other baseline values. However, it is important to note that MS data derived from *in vivo* studies are noisy. Indeed, at the 6- and 12-week time points where a high ¹³C₆-Lys signal is present in one biological replicate, the ¹³C₆-Lys signal was not detected in the other two biological replicates. Therefore, this is unlikely to represent a real increase in ¹³C₆-Lys, but more likely to represent noise. To avoid over-interpretation and artifacts in modelling, we accounted for noisy data by only considering time points for which signals were detected in at least two out of three biological replicates (please see also page 1368 in Supplementary Data 2 - second row shows only data points considered for modelling).

615 We agree that the high longevity of epigenetic regulators is very interesting. In addition to HDAC1, we detected several other long-lived epigenetic regulators in the ovary, which are now shown in
620 Supplementary Data 3 page 4 (Reviewer Fig. 4).

Epigenetic regulators

625 **Reviewer Fig. 4:** Comparison of half-lives across various mouse tissues. Shown are different epigenetic regulators. (part of Supplementary Data 3)

9. Why do the authors believe that there is no significant correlation with other tissues that they've studied (Figure S8)? Especially since the distributions of the ovary proteome in Figure S7a appear to overlap well with other tissues.

630 Following the reviewer's suggestion, we now provide more detailed side-by-side comparisons of the half-lives inferred in ovaries compared to half-lives previously published in other mouse tissues (Extended Data Fig. 5a, Supplementary Data 3). In general, we found that the median half-life in ovaries is comparable to young³ and aged brain⁵, but higher compared to liver, cartilage and heart⁴. However, more interestingly, we detect a large group of extremely long-lived proteins in the ovary. A comparison on the protein level revealed that, indeed, many proteins in the ovary have similar half-lives in other tissues, except for those very long-lived proteins (Supplementary Data 3). We now discuss this point in more detail and refer to
635 Supplementary Data 3 at multiple positions of the revised manuscript.

640 **Extended Data Fig. 5a:** Distribution of $H_{1/2}$ determined for various mouse and rat tissues across three studies³⁻⁵ compared to the estimated $H_{1/2}$ distribution in mouse ovary samples in this study. Red line indicates the median $H_{1/2}$ values in the ovary, while the grey line indicates a $H_{1/2}$ value of 100.

10. *“Proteins with protective functions and proteins of the proteostasis network...” were enriched in the long lived proteome. Are known substrates of the chaperones, chaperonins, and other proteostasis machinery long lived in these datasets?*

645 We thank the reviewer for the suggestion to analyze this interesting point. In the revised manuscript, we investigated the turnover of chaperones in comparison to other tissues (Supplementary Data 3 page 5, Reviewer Fig. 5). We find that in both oocytes and ovaries many chaperones are long lived. Indeed, 69% of the detected chaperones in the ‘high’ or ‘intermediate heavy’ cluster have at least one long-lived target also detected in the ‘high/intermediate heavy’ cluster. Our analysis has shown that (i) long lived proteins in the ‘high heavy’ cluster in ovaries contain the highest fraction of chaperones and these chaperones have on average more interactors than chaperones in other clusters; and (ii) interactors of long-lived chaperones can be found across all protein clusters, while interactors of shorter-lived proteins are predominantly also shorter lived. Many chaperone target proteins are not known yet, and we therefore expect that even more chaperones will have long-lived protein targets, likely including oocyte-specific proteins that are not yet well understood. Known examples include proteins of the chaperonin-containing T-complex (detected as long-lived in both oocytes and ovaries), which interacts e.g., with actin (ACTG1 [$H_{1/2}$ = 143 days; confidence range: 81-435 days]) and tubulins (TUBA1A [$H_{1/2}$ = 60 days; confidence range: 43-93 days], TUBB4B [$H_{1/2}$ = 68.5 days; confidence range: 41-124 days]; and TUBA1C [$H_{1/2}$ = 28 days, confidence range: 8-287 days])¹⁴. Another example is HYOU1 ($H_{1/2}$ =197 days; confidence range: 138-412 days), which has a pivotal role in cytoprotective cellular mechanisms triggered by oxygen deprivation and which may also play a role as a molecular chaperone and participate in protein folding¹⁵. HYOU1 interacts with HSPA5 ($H_{1/2}$ =53 days; confidence range: 40-73 days) and VCP ($H_{1/2}$ =172 days; confidence range: 106-344 days), both of which have important role in protein folding and quality control in the endoplasmic reticulum lumen^{16, 17}. VCP has further an important role in the formation of the transitional endoplasmic reticulum and is required for the fragmentation of Golgi stacks during mitosis and for their reassembly after mitosis¹⁸.

650

655

660

665

Chaperones

Reviewer Fig. 5: Comparison of half-lives across various mouse tissues. Shown are different chaperones. (part of Supplementary Data 3)

Minor Feedback

- 670 1. *The ^{13}C enrichment appears to have a median value 97.99-98.7% from Fig S1, so half of the peptides have an incorporation less than this. Yet, the manuscript repeatedly mentioned that the incorporation was greater than 98.7% (such as in Fig 2a and 2b).*

675 We thank the reviewer for requesting clarification of this point. We analyzed the blood samples from females up to the F2 generation. Over the course of the experiment (several years), we continuously replaced the mothers with new mothers (F3, F4 etc) who will have a higher labelling density compared to the F2 generation. We intended to highlight this point with the >98.8% $^{13}\text{C}_6$ -Lys labeled mother. We understand that this label might be confusing and have therefore removed it in the revised manuscript from Fig. 2a and 2b, and now rather explain in the legend of Extended Data Fig. 1 that females were continuously replaced and therefore had an even higher $^{13}\text{C}_6$ -Lys density as the study progressed (p. 45, lines 1640-1644).
680 Thus, the value calculated in F2 is the minimal labeling efficiency, as all subsequent generations had a larger fraction of $^{13}\text{C}_6$ -Lys due to longer $^{13}\text{C}_6$ -Lys feeding time.

- 685 2. *Figure S7a and S7b are genuinely some of the most striking in the paper, and I think better highlight the high/intermediate/low phenotypes than much of the main text allusions to these proteins. I would suggest pushing these into the main text.*

We thank the reviewer for this helpful suggestion. Following the reviewer's suggestion, we have now restructured the manuscript and moved former Extended Data Fig. 7b to Fig. 3d, and show the full $\text{H}_{1/2}$ distribution of the ovary in Fig. 3a of the main text.

- 690 3. *Why did the authors choose to use trypsin instead of LysC to increase peptide identifications?*

We thank the reviewer for raising this point and agree, that LysC treatment instead of trypsin is in general favorable in terms of increasing the number of quantifiable C-terminal-lysine peptides. However, as we used exhaustive sample fractionation to increase analysis depth, we decided to use trypsin, to avoid generation of very long peptides and increase protein sequence coverage.

695

References:

1. Doherty, M.K., Whitehead, C., McCormack, H., Gaskell, S.J. & Beynon, R.J. Proteome dynamics in complex organisms: using stable isotopes to monitor individual protein turnover rates. *Proteomics* **5**, 522-533 (2005).
- 700 2. Guan, S., Price, J.C., Ghaemmaghami, S., Prusiner, S.B. & Burlingame, A.L. Compartment modeling for mammalian protein turnover studies by stable isotope metabolic labeling. *Analytical Chemistry* **84**, 4014-4021 (2012).
3. Fornasiero, E.F. *et al.* Precisely measured protein lifetimes in the mouse brain reveal differences across tissues and subcellular fractions. *Nature Communications* **9** (2018).
- 705 4. Rolfs, Z. *et al.* An atlas of protein turnover rates in mouse tissues. *Nature Communications* **12** (2021).
5. Kluever, V. *et al.* Protein lifetimes in aged brains reveal a proteostatic adaptation linking physiological aging to neurodegeneration. *Science advances* **8** (2022).
6. Dantuma, N.P., Lindsten, K., Glas, R., Jellne, M. & Masucci, M.G. Short-lived green fluorescent proteins for quantifying ubiquitin/proteasome-dependent proteolysis in living cells. *Nat Biotechnol* **18**, 538-543 (2000).
- 710 7. Han, L. *et al.* Proteasome inhibitor MG132 inhibits the process of renal interstitial fibrosis. *Exp Ther Med* **17**, 2953-2962 (2019).
8. Tarjanyi, O. *et al.* Prolonged treatment with the proteasome inhibitor MG-132 induces apoptosis in PC12 rat pheochromocytoma cells. *Sci Rep* **12**, 5808 (2022).
9. Rodríguez-Nuevo, A. *et al.* Oocytes maintain ROS-free mitochondrial metabolism by suppressing complex I. *Nature* **2022** 607:7920 **607**, 756-761 (2022).
- 715 10. Ma, X.M. & Blenis, J. Molecular mechanisms of mTOR-mediated translational control. *Nature Reviews Molecular Cell Biology* **2009** 10:5 **10**, 307-318 (2009).
11. Zhao, J., Zhai, B., Gygi, S.P. & Goldberg, A.L. MTOR inhibition activates overall protein degradation by the ubiquitin proteasome system as well as by autophagy. *Proceedings of the National Academy of Sciences of the United States of America* **112**, 15790-15797 (2015).
- 720 12. Morita, M. *et al.* mTOR coordinates protein synthesis, mitochondrial activity and proliferation. *Cell Cycle* **14**, 473-480 (2015).
13. Kinnear, H.M. *et al.* The ovarian stroma as a new frontier. *Reproduction* **160**, R25-R39 (2020).
14. Kelly, J.J. *et al.* Snapshots of actin and tubulin folding inside the TRiC chaperonin. *Nat Struct Mol Biol* **29**, 420-429 (2022).
- 725 15. Rao, S. *et al.* Biological Function of HYOU1 in Tumors and Other Diseases. *Onco Targets Ther* **14**, 1727-1735 (2021).
16. van den Boom, J. & Meyer, H. VCP/p97-Mediated Unfolding as a Principle in Protein Homeostasis and Signaling. *Mol Cell* **69**, 182-194 (2018).
- 730 17. Wang, J., Lee, J., Liem, D. & Ping, P. HSPA5 Gene encoding Hsp70 chaperone BiP in the endoplasmic reticulum. *Gene* **618**, 14-23 (2017).
18. Rabouille, C., Levine, T.P., Peters, J.M. & Warren, G. An NSF-like ATPase, p97, and NSF mediate cisternal regrowth from mitotic Golgi fragments. *Cell* **82**, 905-914 (1995).

Decision Letter, first revision:

Our ref: NCB-A52386A

14th March 2024

Dear Dr. Schuh,

Thank you for submitting your revised manuscript "The maintenance of oocytes in the mammalian ovary involves extreme protein longevity" (NCB-A52386A). I am really very sorry for the long delay before communicating our decision with you. The revision has now been seen by the original referees and their comments are below. The reviewers find that the paper has improved in revision, and therefore we'll be happy in principle to publish it in Nature Cell Biology, pending minor revisions to comply with our editorial and formatting guidelines.

Please note that the current version of your manuscript is in a PDF format; could you please email us a copy of the file in an editable format (Microsoft Word or LaTeX), as we can not proceed with PDFs at this stage? Many thanks for your attention to this point.

With the Word file in-hand, we will begin performing detailed checks on your paper and will send you a checklist detailing our editorial and formatting requirements in about 2 weeks. Please do not upload the final materials and make any revisions until you receive this additional information from us.

Thank you again for your interest in Nature Cell Biology. Please do not hesitate to contact me if you have any questions.

Sincerely,

Melina

Melina Casadio, PhD
Senior Editor, Nature Cell Biology
ORCID ID: <https://orcid.org/0000-0003-2389-2243>

Reviewer #1 (Remarks to the Author):

Thanks for the opportunity to carefully read all the comments. The authors have undertaken significant changes and improved the manuscript by including more data in the figure panels to accompany the schematics. and provided additional data throughout. I am happy the authors have addressed all my questions and made changes as appropriate.

Reviewer #2 (Remarks to the Author):

The authors have comprehensively and convincingly addressed all my comments and those of the other reviewers. I fully support the publication of the revised manuscript.

Reviewer #3 (Remarks to the Author):

The authors addressed all of my concerns and comments. I very much enjoyed reading through the revised paper and comments.

Decision Letter, final checks:

Our ref: NCB-A52386A

27th March 2024

Dear Dr. Schuh,

Thank you for your patience as we've prepared the guidelines for final submission of your Nature Cell Biology manuscript, "The maintenance of oocytes in the mammalian ovary involves extreme protein longevity" (NCB-A52386A). Please carefully follow the step-by-step instructions provided in the attached file, and add a response in each row of the table to indicate the changes that you have made. Ensuring that each point is addressed will help to ensure that your revised manuscript can be swiftly handed over to our production team.

In recognition of the time and expertise our reviewers provide to Nature Cell Biology's editorial process, we would like to formally acknowledge their contribution to the external peer review of your manuscript entitled "The maintenance of oocytes in the mammalian ovary involves extreme protein longevity". For those reviewers who give their assent, we will be publishing their names alongside the published article.

Nature Cell Biology offers a Transparent Peer Review option for new original research manuscripts submitted after December 1st, 2019. As part of this initiative, we encourage our authors to support increased transparency into the peer review process by agreeing to have the reviewer comments, author rebuttal letters, and editorial decision letters published as a Supplementary item. When you

submit your final files please clearly state in your cover letter whether or not you would like to participate in this initiative. Please note that failure to state your preference will result in delays in accepting your manuscript for publication.

Cover suggestions

COVER ARTWORK: We welcome submissions of artwork for consideration for our cover. For more information, please see our guide for cover artwork.

Nature Cell Biology has now transitioned to a unified Rights Collection system which will allow our Author Services team to quickly and easily collect the rights and permissions required to publish your work. Approximately 10 days after your paper is formally accepted, you will receive an email in providing you with a link to complete the grant of rights. If your paper is eligible for Open Access, our Author Services team will also be in touch regarding any additional information that may be required to arrange payment for your article.

Please note that *Nature Cell Biology* is a Transformative Journal (TJ). Authors may publish their research with us through the traditional subscription access route or make their paper immediately open access through payment of an article-processing charge (APC). Authors will not be required to make a final decision about access to their article until it has been accepted. Find out more about Transformative Journals

Please use the following link for uploading these materials:
[Redacted]

Best regards,

Kendra Donahue
Staff
Nature Cell Biology

On behalf of

Melina Casadio, PhD
Senior Editor, Nature Cell Biology
ORCID ID: <https://orcid.org/0000-0003-2389-2243>

Reviewer #1:

Remarks to the Author:

Thanks for the opportunity to carefully read all the comments. The authors have undertaken significant changes and improved the manuscript by including more data in the figure panels to accompany the schematics. and provided additional data throughout. I am happy the authors have addressed all my questions and made changes as appropriate.

Reviewer #2:

Remarks to the Author:

The authors have comprehensively and convincingly addressed all my comments and those of the other reviewers. I fully support the publication of the revised manuscript.

Reviewer #3:

Remarks to the Author:

The authors addressed all of my concerns and comments. I very much enjoyed reading through the revised paper and comments.

Final Decision Letter:

Dear Dr Schuh,

I am pleased to inform you that your manuscript, "The maintenance of oocytes in the mammalian ovary involves extreme protein longevity", has now been accepted for publication in Nature Cell Biology.

Thank you for sending us the final manuscript files to be processed for print and online production, and for returning the manuscript checklists and other forms. Your manuscript will now be passed to our production team who will be in contact with you if there are any questions with the production

quality of supplied figures and text.

Please note that *Nature Cell Biology* is a Transformative Journal (TJ). Authors may publish their research with us through the traditional subscription access route or make their paper immediately open access through payment of an article-processing charge (APC). Authors will not be required to make a final decision about access to their article until it has been accepted. Find out more about Transformative Journals

To assist our authors in disseminating their research to the broader community, our SharedIt initiative

provides you with a unique shareable link that will allow anyone (with or without a subscription) to read the published article. Recipients of the link with a subscription will also be able to download and print the PDF.

If you have not already done so, we strongly recommend that you upload the step-by-step protocols used in this manuscript to protocols.io (<https://protocols.io>), an open online resource that allows researchers to share their detailed experimental know-how. All uploaded protocols are made freely available and are assigned DOIs for ease of citation. Protocols and Nature Portfolio journal papers in which they are used can be linked to one another, and this link is clearly and prominently visible in the online versions of both. Authors who performed the specific experiments can act as primary authors for the Protocol as they will be best placed to share the methodology details, but the Corresponding Author of the present research paper should be included as one of the authors. By uploading your Protocols onto protocols.io, you are enabling researchers to more readily reproduce or adapt the methodology you use, as well as increasing the visibility of your protocols and papers. You can also establish a dedicated workspace to collect your lab Protocols. Further information can be found at <https://www.protocols.io/help/publish-articles>.

With kind regards,

Melina Casadio, PhD
Senior Editor, Nature Cell Biology
ORCID ID: <https://orcid.org/0000-0003-2389-2243>
